# Multilevel design and construction in nanomembrane rolling for three-dimensional angle-sensitive photodetection

Ziyu Zhang[1,6], Binmin Wu [1,6], Yang Wang[1], Tianjun Cai[1], Mingze Ma[1], Chunyu You[1], Chang Liu[1], Guobang Jiang [1], Yuhang Hu[1], Xing Li[1], Xiang-Zhong Chen[2,3,4], Enming Song[2,4], Jizhai Cui[1,4], Gaoshan Huang [1,3,4], Suwit Kiravittaya[5] & Yongfeng Mei [1,2,3,4] ✉

Releasing pre-strained two-dimensional nanomembranes to assemble on-chip three-dimensional devices is crucial for upcoming advanced electronic and optoelectronic applications. However, the release process is affected by many unclear factors, hindering the transition from laboratory to industrial applications. Here, we propose a quasistatic multilevel finite element modeling to assemble three-dimensional structures from two-dimensional nanomembranes and offer verification results by various bilayer nanomembranes. Take Si/Cr nanomembrane as an example, we confirm that the three-dimensional structural formation is governed by both the minimum energy state and the geometric constraints imposed by the edges of the sacrificial layer. Large-scale, high-yield fabrication of three-dimensional structures is achieved, and two distinct three-dimensional structures are assembled from the same precursor. Six types of three-dimensional Si/Cr photodetectors are then prepared to resolve the incident angle of light with a deep neural network model, opening up possibilities for the design and manufacturing methods of More-than-Moore-era devices.

Three-dimensional (3D) structure assembled by releasing pre-strain nanomembrane has become an essential technique for electronic[1–4], optical[5–8], and magnetic[9–11] applications. The self-assembly of the nanomembrane depends on the large deformation in ultrathin thickness and the out-plane strain gradient in a multilayer nanomembrane system. Generally, the strain gradient derives from thermal strain that is caused by temperature change, recovery of pre-stretched substrate, and lattice mismatch of epitaxial material. By tuning etching edges, nanomembrane systems, and deposition parameters, preparation of tubes[12–14], helices[15–17], buckles[18,19], and other structures[20–22] have been achieved for applications in photodetectors[8,23], field effect transistors[24,25], and microrobots[13,26]. However, the formation of nanomembrane is affected by multiple complex factors in the release process, such as etching trajectory, balance between chemical reaction and etchant diffusion, fluid behaviors of etchant, aspect ratio, etc. Unpredictable and uncontrollable incorrect formation challenges the uniformity and yield in device manufacturing.

Many analytical and numerical methods have been used for morphological prediction and design guidelines for pre-strained nanomembrane structures[27–32]. As early as 1925, Timoshenko has developed the curvature radius formula of the bi-metal thermostat, which is applicable to substrates of similar thickness to nanomembranes[33].

[1]Department of Materials Science & State Key Laboratory of Molecular Engineering of Polymer, Fudan University, Shanghai 200438, People's Republic of China. [2]Shanghai Frontiers Science Research Base of Intelligent Optoelectronics and Perception, Institute of Optoelectronics, Fudan University, Shanghai 200438, People's Republic of China. [3]Yiwu Research Institute of Fudan University, Yiwu 322000 Zhejiang, People's Republic of China. [4]International Institute of Intelligent Nanorobots and Nanosystems, Fudan University, Shanghai 200438, People's Republic of China. [5]Department of Electrical Engineering, Faculty of Engineering, Chulalongkorn University, Bangkok, Thailand. [6]These authors contributed equally: Ziyu Zhang, Binmin Wu. ✉e-mail: yfm@fudan.edu.cn

In the later development, Nikishkov[34] and Hsueh[35] have presented the analytical solutions of the curvature formula suitable for self-rolling nanomembranes. In addition, finite element modeling (FEM), as a numerical method, can intuitively simulate and predict the behavior of nanomembranes after release[28–31]. Huang et al. have demonstrated a quasistatic FEM model with moving boundaries to simulate the self-rolling process of nanomembranes[29]. Generally, the rolling behavior of completely released nanomembranes from long sides is regarded as a preference in minimal elastic energy[6,28,31,36]. Nevertheless, short-side rolling has been observed in many experimental results, which is interpreted as the dependence of rolling trajectory[6,28]. Idealizing the boundary movement of the sacrificial layer for specific structural formation or analysis in a local area lacks broad applicability and an accurate boundary condition model.

It is crucial to adopt a design approach that takes multiple factors into account to ensure efficient processing and optimal application of micro/nanoscale self-assembled devices. Here, we propose a quasi-static multilevel FEM that emphasizes the geometry and boundary condition of nanomembrane for 3D self-assembly structure. By considering the non-uniformity and geometric dependency of the etching process, we reproduce the anisotropy of boundary movement of the sacrificial layer. The multilevel FEM method is successfully applied in a wide range of material systems, nanomembrane thicknesses, pattern types, and pattern sizes, demonstrating excellent generalizability. We take Si/Cr rectangular nanomembrane as an example to study the mechanism and application of the self-assembly process in detail. An idealized model is used to simulate and calculate the elastic energy changes corresponding to different geometric characteristics quantitatively. The gradual shift observed in the rolling direction reflects the transition of minimal elastic energy and explains the change in 3D structural forming behavior with varying aspect ratios. Based on this FEM model, large-scale, high-yield, and high-uniformity 3D configurable structures are successfully achieved. In addition, we develop a range of 3D structure photodetectors for angle detection of incident light at an accuracy of 10° to demonstrate the potential in manufacturing electronic and photoelectronic devices.

## Results

### Release of pre-strained bilayer nanomembrane

We established a standard strain nanomembrane release model consisting of a rectangular nanomembrane with width $W$, length $L$, and in-plane strain gradient $\Delta\varepsilon$, sacrificial layer, and substrate, as shown in Fig. 1a. A fixed edge is deployed to ensure the on-chip self-assembly of nanomembrane, resulting in the morphological difference between opposite edge rolling and adjacent edge rolling. Once the sacrificial layer is etched, the pre-strained nanomembrane will release from the substrate and then self-assemble due to strain gradient, as shown in the right panel of Fig. 1b. During the release process, etching of the sacrificial layer can be modeled as a gradual movement of boundary, in which the velocity and direction of movement depend on the chemical reaction parameters at the solid-liquid interface, as shown in Fig. 1c. Geometric size of nanomembrane will lead to a diversity of etching velocity and direction at the solid-liquid interface, which can be intuitively represented by $\frac{v_W}{v_L}$, the velocity ratio of the velocity perpendicular to the length direction and width direction of a coordinate on nanomembrane, which is shown in the inset of Fig. 1d. The method

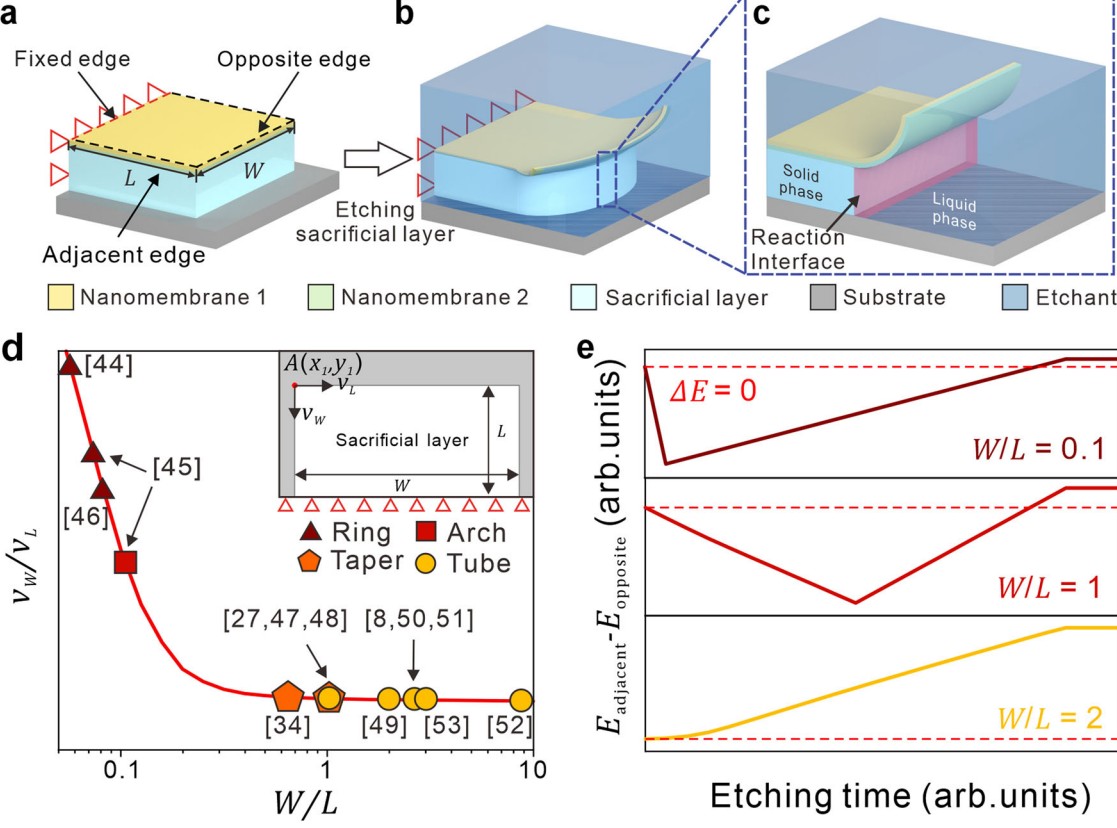

**Fig. 1 | Releasing of pre-strain rectangle bilayer nanomembranes. a** Basic model of pre-strain nanomembrane. **b** Wet release process of pre-strain bilayer nanomembrane with a fixed edge. **c** Schematic of sacrificial layer wet etching model: etchant diffusion from liquid phase to solid-liquid interface and chemical reaction between etchant and sacrificial layer in interface. **d** Self-assembled 3D structure types change respect to aspect ratio. Schematic of velocity ratio and aspect ratio (inset). **e** Elastic energy difference of nanomembranes that is released from adjacent and opposite edge for different aspect ratios. Source data are provided as a Source Data file.

of etching FEM is provided in Methods and Supplementary Fig. 1. In previous studies[20,32,37–48], the structures of nanomembranes exhibit a transition from ring to arch, taper, and finally tubes with an increase in aspect ratios. When the long side is much longer than the short side ($W \gg L$ or $W \ll L$), the etching from the long side will dominate the etching direction, and short-side etching is negligible due to the short distance between opposite edges. When the $W/L$ is close to 1, etching from both sides needs to be considered because the rolling direction may be competitive in similar geometric size (Fig. 1d). For example, Chalapat et al.[38] discovered the transition between ring and arch structures via fabrication of Ti/Al/Cr nanomembrane with increasing aspect ratios. The above phenomenon shows the potential of structural design with different aspect ratios. In pre-strained nanomembrane with various aspect ratios, there are differences in elastic energy after release from two directions. We set up a series of FEM models that release from different directions to simulate the change in elastic energy during etching. As shown in Fig. 1e, when the etching direction is fixed to adjacent and opposite edge, the gap in elastic energy from FEM calculation indicates that the distinct release direction may result in two types of 3D structures. $E_{\text{adjacent}}$ and $E_{\text{opposite}}$ represent elastic energy of adjacent and opposite edge release during etching, and $E_0$ is the elastic energy of unreleased nanomembrane. As for $W/L = 0.1$ and $W/L = 1$, the elastic energy decreases fast in the adjacent side release at the initial state, which will cause the adjacent side rolling in the same etching depth. Compared to the common tubular structure rolling from opposite edge, the obvious decrease from adjacent side indicates the possibility of adjacent side rolling and potential of multi-morphic 3D assembly from one pattern. In the situation of $W/L = 2$, the elastic energy of the opposite rolling is in a low energy state throughout the entire process, proving the change in the prior rolling direction. As mentioned above, etching velocity, etching direction, and aspect ratio are both important factors for 3D structural assembly, and they need to be considered in FEM modeling.

## Quasistatic multilevel finite element method modeling

In order to reproduce the formation process and find the mechanism of boundary condition during the release process, a multi-field coupling FEM model is utilized to simulate the etching process in the sacrificial layer via COMSOL Multiphysics software package[49–51]. In the etching FEM model, the bilayer is initially fixed with constraints to simulate the unreleased state. Considering the etchant flowing and boundary movement of the etching system, Fick's law, Navier-Stokes equations, and liquid-solid reaction model are introduced to establish the FEM model:

$$\begin{cases} \frac{\partial c}{\partial t} + \mathbf{u} \cdot \nabla c = \nabla \cdot (D \nabla c) \\ D \nabla c \cdot \mathbf{n} = -kc \\ \rho_{\text{Etchant}} \frac{\partial \mathbf{u}}{\partial t} + \rho_{\text{Etchant}} (\mathbf{u} \cdot \nabla) \mathbf{u} = \nabla \cdot \left[ -\rho_{\text{Etchant}} \mathbf{I} + \mu \left( \nabla \mathbf{u} + (\nabla \mathbf{u})^T \right) \right] \\ \upsilon = \frac{kcDM_{\text{Sacrificial}}}{\rho_{\text{Sacrificial}}} \end{cases} \quad (1)$$

where $c$ is etchant concentration, $\mathbf{u}$ is velocity of etchant flow, $D$ is diffusion coefficient, $k$ is reaction rate coefficient, $\mathbf{n}$ is normal vector, $\rho$ is density, $\mu$ is dynamic viscosity coefficient, $\upsilon$ is normal velocity of mesh, and $M$ is molar mass. The relationship between etchant concentration and reaction rate depends on the balance of diffusion and chemical reaction process, and the effect of each parameter is discussed (Supplementary Notes 1, 2, and Supplementary Figs. 2–5). As can be seen in Fig. 2a, since the FEM model is made up of a set of small units, the etching process of the sacrificial layer is realized by normal movement of unit meshes with a velocity $\upsilon$.

We perform a series of analytical steps in a time sequence and then realize the whole dynamic release process of bilayer nanomembranes. Geometrical features of boundary variation obtained from

previous simulations are applied to the feature division along the thickness direction of the Si/Cr bilayer in coordinates ($x_i$, $y_i$), $i = 1, 2, ...,$ node numbers, as shown in Fig. 2b. After geometric partitioning, boundaries under multiple discrete time points are divided into boundary conditions. In the elastic mechanical analysis step, the boundary conditions will be set as a series of constraints in chronological order to realize the continuous release of the nanomembrane (Fig. 2c).

Due to the asymmetric boundary conditions, complex practical release process, and large geometric deformation of the one-ended fixed bilayer nanomembrane structure, it is inaccurate to apply the large deflection equations of thin elastic plates (such as the Föppl-von Kármán equation[52]) for structural design. Considering only the release from the opposite edge, the 3D assembly of the nanomembranes will exhibit only tubular structures (Supplementary Note 3 and Supplementary Fig. 6). In contrast, the simulation results of the bilayer nanomembranes are guided by the multilevel design model during the release process, demonstrating better accuracy and successfully reproducing morphologies varying with width (Fig. 2d). Quasistatic FEM results prove that the etching trajectory of the sacrificial layer plays a crucial role in tuning the release of bilayer nanomembranes. In addition to the Si/Cr rectangular patterns, the multilevel FEM model has been designed for various types of patterns such as semicircular, triangular, and parallelogram patterns (Fig. 2d, Supplementary Note 4, Supplementary Figs. 7–14, and Supplementary Movies 1, 2), as well as in the high-frequency/low-frequency $SiN_x$ (LF/HF $SiN_x$) nanomembrane, NiTi nanomembrane, and $VO_2$/Cr nanomembrane systems, which demonstrate the potential for a wide range of applications (Fig. 2e-h, Supplementary Note 5, and Supplementary Figs. 15–17). Meanwhile, the method demonstrates cross-scale compatibility for models from the hundred nanometer scale to the hundred micrometer scale (Supplementary Note 6 and Supplementary Figs. 18–22). In this study, we choose the rectangular Si/Cr rectangular nanomembrane for detailed analysis and discussion of the self-assembly behavior due to their moderate parameter complexity and high geometrical symmetry.

## Experimental verification

To verify the design model, a series of silicon (Si)/chromium (Cr)/germanium (Ge) heterostructure with gradient aspect ratio was deposited by electron beam evaporation (Supplementary Note 7 and Supplementary Fig. 23). Si/Cr bilayer nanomembrane system is selected for experimentation, in which Cr can introduce large pre-strain and Si is the most widely used semiconductor materials. The lithography and reaction ion etching then define the one-end fixed rectangular sacrificial layer area by photoresist to produce a window for sacrificial layer etching. As shown in Fig. 3a, a series of one-end fixed Si/Cr bilayer nanomembranes with length $L = 40\,\mu m$ are selected as an example, and the width $W$ is increased from 2 μm to 80 μm with a step of 2 μm. The photoresist was removed via acetone immersion and supersonic cleaning to avoid surface residue of organic solvents. After rinsing the surface in deionized water and drying, the Ge sacrificial layer is etched to realize Si/Cr bilayer release. Ge sacrificial layer was removed by wet etching (30% $H_2O_2$, 75 °C). When Si/Cr bilayers in the etching area are released, they are transferred to acetone and dried in critical point dryer in liquid $CO_2$ to prevent morphological defects caused by conventional drying. When the length $L = 40\,\mu m$, there are obvious transition regions ($W = 8–10\,\mu m$, $16–18\,\mu m$, $28–30\,\mu m$, and $56–58\,\mu m$) and stable regions ($W = 2–6\,\mu m$, $12–14\,\mu m$, $20–26\,\mu m$, $32–54\,\mu m$, and $60–68\,\mu m$) in the distribution map of various morphology, which confirms that 3D assembly is stable and related to the aspect ratio of geometric shapes (Fig. 3a). The morphologies of the released nanomembrane exhibit a transitional change as the width increases: five types of 3D structures of ring, arch, helix, taper, and tube (Fig. 3b). For patterns in transition regions, the unstable structure formation

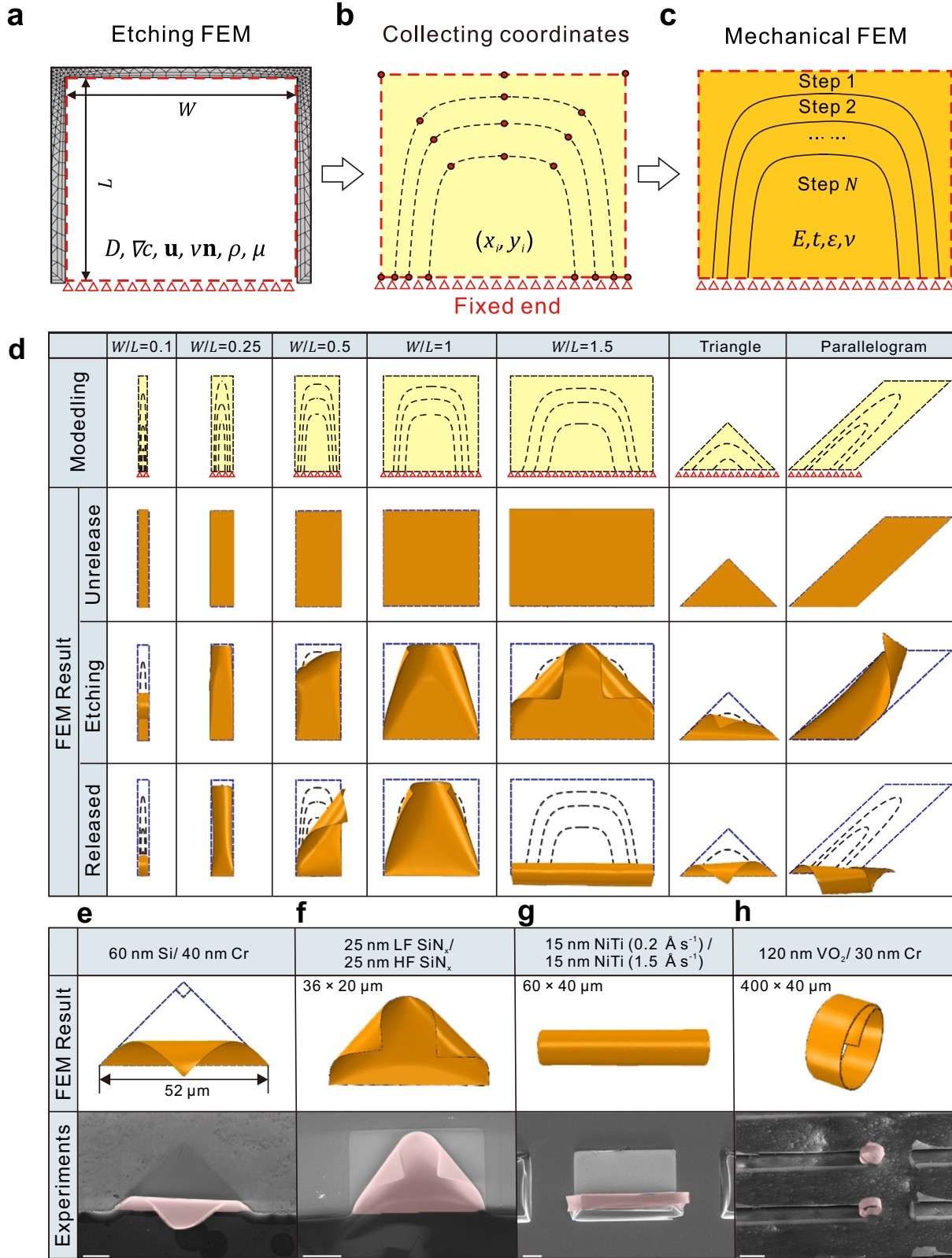

**Fig. 2 | Quasistatic multilevel model FEM simulation of bilayer pre-strain nanomembrane release.** The modeling process of multilevel quasistatic FEM: **a** simulating sacrificial layer boundary movement of the release process, **b** collecting coordinates of boundary in discrete time points, **c** importing coordinates into the dynamic simulation as a boundary condition. **d** Modeling and the result of quasistatic multilevel FEM modeling, in which moving boundaries are derived from coordinates in the reaction-diffusion model. Quasistatic multilevel FEM simulation and experimental results of **e** 60 nm Si/40 nm Cr triangle pattern, **f** 25 nm LF SiN$_x$/25 nm HF SiN$_x$ rectangle pattern, **g** 15 nm NiTi (0.2 Å s$^{-1}$)/15 nm NiTi (1.5 Å s$^{-1}$) rectangle pattern, and **h** 120 nm VO$_2$/30 nm Cr rectangle patterns. Scale bars of **e–g** are 10 μm, and scale bar of **h** is 100 μm. Pink areas in SEM images are self-assembled structures. Source data are provided as a Source Data file.

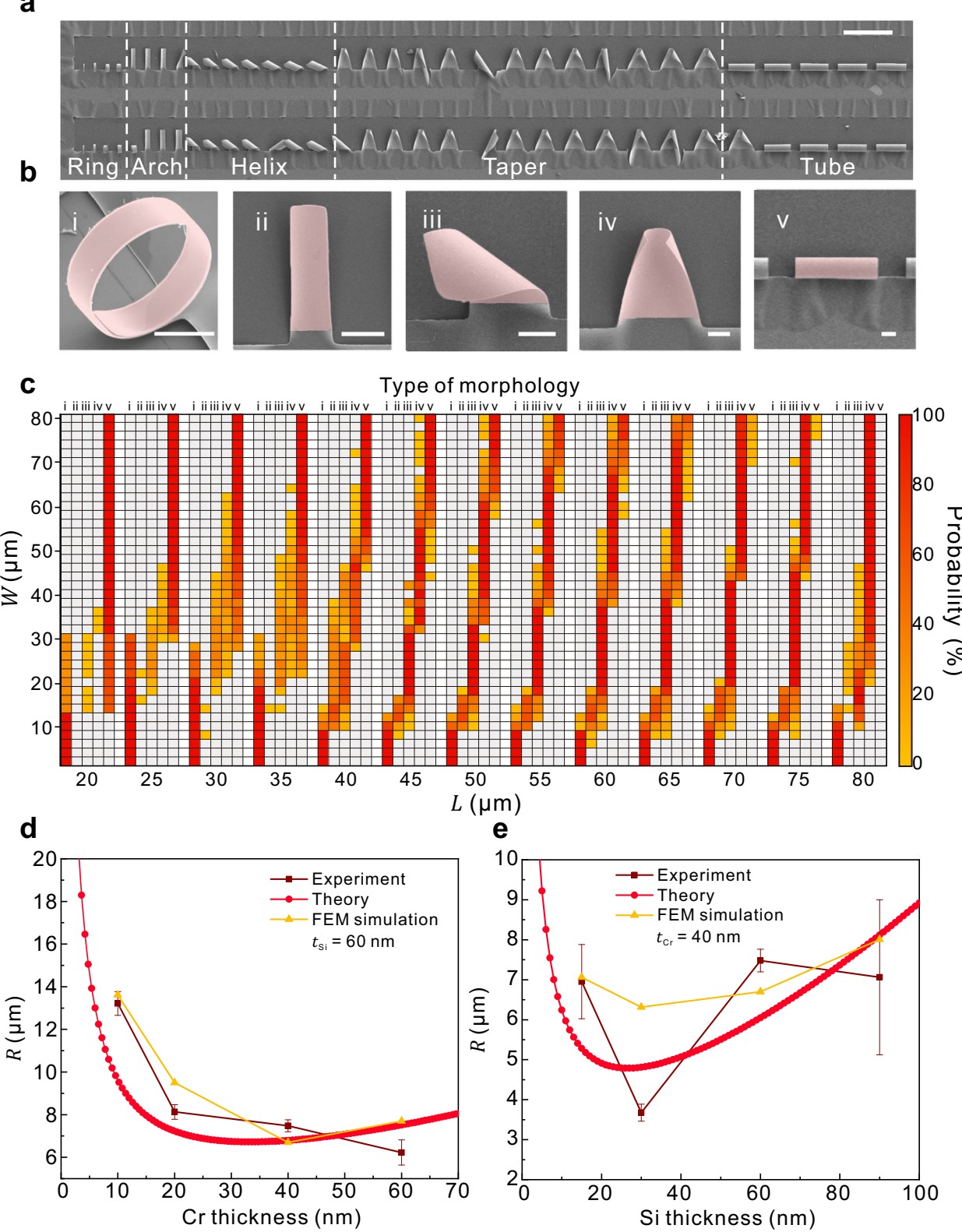

**Fig. 3 | 3D structures released from bilayer nanomembrane. a** SEM image of released Si/Cr bilayer nanomembrane with fixed length $L$ of 40 μm and various widths $W$ from 2 to 80 μm with step of 2 μm. Scale bars, 100 μm. **b** Five structures: i. ring, ii. arch, iii. helix, iv. taper, and v. tube. Scale bars, 10 μm. **c** Statistical table of the probability of morphological features in pattern with different sizes of $L$ from 20 to 80 μm with the step of 5 μm, and $W$ from 2 to 80 μm with the step of 2 μm, where the order numbers represent corresponding structures in **b**. **d** Experimental

result, theory calculation, and multilevel FEM simulation of radius of tubes with fixed Si layer thickness ($t_{Si}$ = 60 nm) and varied Cr layer thickness. **e** Experimental result, theory calculation, and multilevel FEM simulation of radius of tubes with fixed Cr layer thickness ($t_{Cr}$ = 40 nm) and varied Si layer thickness. The error bars in **d**, **e** are the standard deviation of radii in experiments. Pink areas in SEM images are self-assembled structures. Source data are provided as a Source Data file.

demonstrates the competition of two rolling directions and a relatively small gap in elastic energy. To systematically analyze the relationship between structural transition and aspect ratio, patterns with different lengths $L$ from 20 µm to 80 µm (step of 5 µm) were prepared. As shown in Fig. 3c, the probability distribution of five types of morphology and pattern sizes is discussed from a statistical point of view. With the increase in length, the transition width of each structure increases accordingly due to the variation in aspect ratio. For patterns with large lengths ($L = 80$ µm), the structure will lack the transition from taper to tube, but it can be fabricated by a design with larger width. The forming morphology of nanomembranes also follows the order of ring-arch-helix-taper-tube with increasing aspect ratio, corresponding to the structure distribution in Fig. 1d and Fig. 2d. Moreover, we designed Si/Cr systems containing Si nanomembranes (15 nm/30 nm/ 90 nm) and Cr nanomembranes (10 nm/20 nm/60 nm) of different thicknesses to analyze the initial strain and average strain of Si and Cr nanomembranes after release (Supplementary Note 8, Supplementary Figs. 24–30 and Supplementary Table 1). As shown in Fig. 3d,e, we collected the radii of tubular structures with the same Si thickness ($t_{Si} = 60$ nm) and different Cr thicknesses, and the same Cr thickness ($t_{Cr} = 40$ nm) and different Si thicknesses. With the increase of the thickness of Cr layer, the radii of assembled tubes with $t_{Si} = 60$ nm are $13.21 \pm 0.55$ µm ($t_{Cr} = 10$ nm), $8.13 \pm 0.34$ µm ($t_{Cr} = 20$ nm), $7.48 \pm 0.28$ µm ($t_{Cr} = 40$ nm), and $6.22 \pm 0.59$ µm ($t_{Cr} = 60$ nm), respectively. Meanwhile, with the increase of the thickness of Si layer, the radii of assembled tubes with $t_{Cr} = 60$ nm are $6.95 \pm 0.93$ µm ($t_{Si} = 15$ nm), $3.67 \pm 0.21$ µm ($t_{Si} = 30$ nm), $7.48 \pm 0.28$ µm ($t_{Si} = 60$ nm), and $7.06 \pm 1.93$ µm ($t_{Si} = 90$ nm), respectively. We used the above experiments and the force-moment balance formula to estimate the strain magnitude in the Si/Cr layer, that is

$$R = \frac{(E_{Si})^2 t_{Si}^4 + (E_{Cr})^2 t_{Cr}^4 + 2E_{Si}E_{Cr}t_{Si}t_{Cr}(2t_{Si}^2 + 2t_{Cr}^2 + 3t_{Si}t_{Cr})}{6E_{Si}E_{Cr}t_{Si}t_{Cr}(t_{Si} + t_{Cr})\Delta\varepsilon}, \quad (2)$$

where $E$ is Young's modules, $t$ is thickness of nanomembrane, and $\Delta\varepsilon = \varepsilon_{Cr} - \varepsilon_{Si}$ is the initial strain difference between Si and Cr layer. We substitute the thicknesses of nanomembranes and tube radius (Fig. 3d,e) into this formula for fitting, and the prestrain of the Cr layer obtained by fitting is $\varepsilon_{Cr} \sim 0.65\%$, while the pre-strain of the Si layer is determined to be $\varepsilon_{Si} \sim -0.55\%$. The corresponding stress gradient is approximately $\Delta\sigma = \varepsilon_{Cr}E_{Cr} - \varepsilon_{Si}E_{Si} \approx 2600$ MPa, which is consistent with the preset condition in our simulation ($\Delta\sigma = 2000$ MPa), providing an important guarantee for high-accuracy structural design. In addition, we used the thickness and pre-strain of nanomembranes as parameters to calculate the strain relationship in the Si/Cr nanomembrane. The average strain of Si and Cr nanomembranes are also in good agreement with the experiment, providing an effective method for strain characterization of multilayer nanomembrane systems (Supplementary Fig. 31).

## Elastic energy of released bilayer nanomembrane

The assembly of nanomembranes is well simulated. However, the essence of formation and the size dependence of different structures need to be explored. The optimized elastic energy calculation is derive from the Nikishkov's analytical solution model[34] (Supplementary Note 9 and Supplementary Figs. 32–34). We approximate the geometric parameters of the model as boundary conditions with symmetrical specific divide angles $\alpha$ to combine the effect of etching direction $\alpha = \arctan\frac{v_w}{v_l}$ and aspect ratio $W/L$ during release. Patterns with a length $L$ of 40 µm and a width $W$ of 10–80 µm are set with a series of direction angles for unilateral (Supplementary Note 10 and Supplementary Fig. 35) and bilateral (Fig. 4a) release models. The divided boundaries are parallel, and the divided area is classified as Region I. The constraint of the nanomembrane in the Region I will be removed inward from the boundary to simulate directional release.

When Region I is completely released, the nanomembrane in Region II starts to release. Finally, the elastic energy of the released nanomembrane is collected via FEM. The geometric characteristics of FEM results are provided to idealized analytical elastic energy equations to verify the results of simulations.

It can be observed in Fig. 4b that among the relative strain energy of different widths, we can find the direction angle $\alpha_{min}$ with the minimal relative elastic energy shifts to a larger direction angle with increasing width. The $\alpha_{min}$ of the bilateral-released model shifts from ~70° to ~45° when the width increases from 10 µm to 80 µm. This decreased transition of $\alpha_{min}$ indicates long-side rolling along the opposite edge. When the rolling direction decreases to 0°, the rolling will become the classic model of opposite rolling in Fig. 1d. Then, the $\alpha_{min}$ shifts back from ~70° to ~50° after the width decreases from 10 µm to 4 µm. The shift back of $\alpha_{min}$ is due to the dominant of long-side rolling from length direction, which is both restricted by high $\frac{v_w}{v_l}$ and a small aspect ratio $W/L \leq 0.25$. Therefore, the pattern of $W = 4$ µm with $\alpha_{min} = -50°$ assembles into a ring rather than rolling from adjacent edges. Generally, the existence of the $\alpha_{min}$ verifies the dependency between aspect ratio and etching direction and the relationship to the minimum energy theorem. For other $\alpha$ with higher relative elastic energy, the bilayer nanomembranes show reduced possibilities for rolling. The elastic energy calculation shows the same increasing and decreasing trend of the $\alpha_{min}$ shift. Fluctuations in relative elastic energy derived from estimation errors in selecting idealized regions.

Moreover, the transition of $\alpha$ contributes to the differences in the 3D structures of nanomembranes (Supplementary Note 11 and Supplementary Figs. 36–39). Taking $L = 40$ µm as an example, the probability of occurrence of various structures exhibits gradient change from 4 µm to 80 µm, which corresponds to the shift of $\alpha_{min}$ (Fig. 4c). For each structure, there are five corresponding areas of $\alpha_{min}$ to assemble: ring (30°–40°), arch (50°–80°), helix (65°–75°), taper (50°–65°), and tube (40°–55°). It is confirmed that the structure with the minimal relative elastic energy possesses a higher probability of rolling up along the $\alpha_{min}$ assembling into the corresponding structure (Supplementary Note 12 and Supplementary Fig. 40). As shown in Fig. 4d, the quasistatic FEM model simulates the change of $\alpha$ with etching time during the assembly process, and generally restores the relationship between $\alpha$ and etching time during the sacrificial layer etching process in the experiment (Fig. 4e, Supplementary Movie 3). When the etching finished, the distribution of $\alpha$ are as follows: ring (25°–35°), arch (70°–80°), helix (50°–65°), taper (60°–75°), and tube (40°–55°). The $\alpha$ angle obtained at the end of etching between model calculations and experimental results is highly consistent with the calculated $\alpha_{min}$ corresponding to each structure in Fig. 4c, providing an observation scheme for tracking intermediate states in the assembly process. We noticed that there is an obvious difference of the $\alpha$ between the multilevel model and the experiments in the early stage of etching, which is mainly due to the non-ideal release of sacrificial layer release and local non-uniform etching caused by insufficient accuracy in micro-nano processing (Supplementary Note 13 and Supplementary Fig. 41).

Regarding the Si/Cr bilayer with different widths, we anticipate that the $\alpha_{min}$ will also shift as the width increases with the instruction of FEM-based multilevel design model. With the guidance of the elastic energy map in Fig. 4c, we employed the multilevel design with a single stable energy state to prepare large-scale arrays with only one structure. As shown in Table 1, all five structure arrays (0.8 cm × 0.8 cm) exhibit high yield (Supplementary Note 14 and Supplementary Fig. 42), demonstrating good accuracy in designing 3D structures from self-rolling.

## Configurable structures with controlled etching edges

By introducing sequential etching in designed spatiotemporal order at targeted locations, diverse structural configurations can be achieved

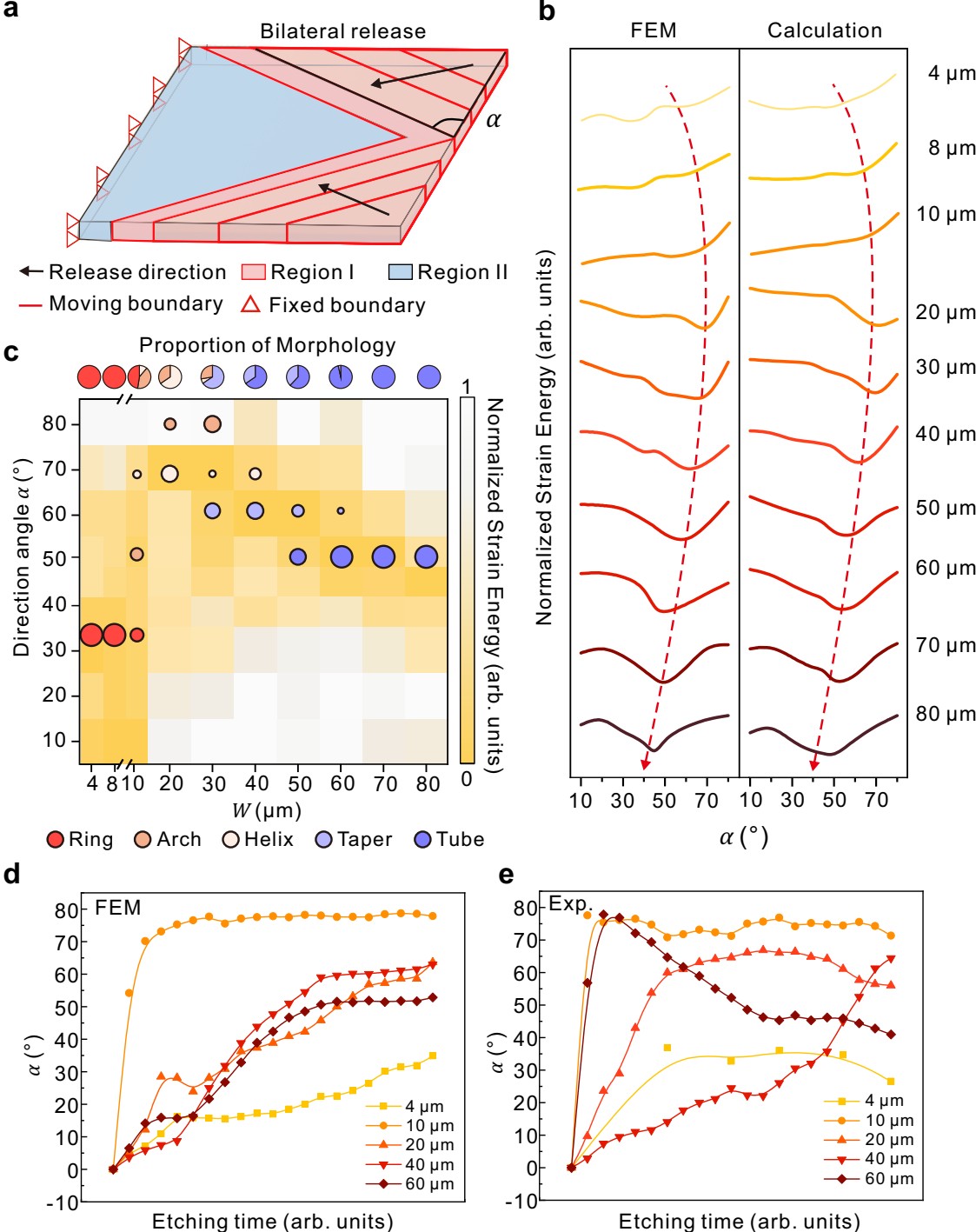

**Fig. 4 | Assembly mechanism and elastic energy of self-assembly nanomembrane. a** Diagram of idealized bilateral release models that consist of two regions released chronologically. **b** Relative elastic energy of idealized bilateral release models with different direction angles and pattern widths via FEM simulation. **c** Relationship between direction angle, distribution of various morphology, and relative strain energy. **d** Multilevel FEM simulations and **e** experimental results of relationship between direction angle $\alpha$ and etching time. Source data are provided as a Source Data file.

for identical patterns. The orientation of the patterns and adjustment of the etchant level position are utilized to control etching at specific locations, resulting in the formation of two distinct 3D structures. During the etching process, the solid-liquid interface rises slowly due to the capillarity between the nanomembrane and the etchant (Supplementary Movie 4). The first method of etching control is bottom etch, in which the fixed end is in contact with the etchant first. Then, the liquid level of the etchant will rise due to the capillary force in the etched pattern (Fig. 5a). In this situation, etching happens at the

adjacent edges first. (Fig. 5b). Then, another method for etching control is top etch, and the opposite edge is in contact with the etchant first, as shown in Fig. 5c. For etching starting from the opposite edge (Si/Cr top etch, Fig. 5d, Supplementary Note 15, and Supplementary Movie 5), the etchant interface prefers to contact the free end initially than adjacent edges. A quasistatic multilevel FEM model with selective release is designed to guide the 3D assembly (Fig. 5e). In the bottom etch, the boundary of the sacrificial layer will change from adjacent sides rather than the overall layer, artificially reducing the $\frac{v_W}{v_L}$.

**Table 1 | Yield of large-scale prepared different 3D structures**

| | $W/L$ (μm/μm) | Target structure (%) | Other structures (%) | Damaged (%) |
|---|---|---|---|---|
| Ring | 4/40 | 98.785 | 1.026 | 0.189 |
| Arch | 10/40 | 99.699 | 0.192 | 0.110 |
| Helix | 25/40 | 99.376 | 0.416 | 0.208 |
| Taper | 40/40 | 95.815 | 3.778 | 0.407 |
| Tube | 80/40 | 98.242 | 1.628 | 0.130 |

The formation of tubular structures benefits from the wrinkle in the partial release process by bottom etch. In local wrinkle structures, the bilayer tends to roll from the directions of wrinkling, when the assembly for other structures faces a higher energy barrier (Supplementary Fig. 43, Supplementary Movie 6). Compared to models in the normal etching process, the movement of the opposite edge in the top etch is larger at the beginning. More area will be released correspondingly in mechanical FEM modeling, leading to a larger $\alpha$ and $\frac{v_w}{v_l}$ in the local area (Supplementary Note 16 and Supplementary Fig. 44). Therefore, there is a significant variation in bilayer morphology in bottom etching and top etching. Selective etching introduces controllable regulation of etching direction and etching area, playing an important role in structural assembly. The successful preparation of the multi-morphic structure also proves the correctness of elastic energy analysis. By introducing extra boundary conditions, etching direction, and velocity ratio $\frac{v_w}{v_l}$ can also be simulated and realized. With proper control of surface tension, reconfiguration of different structure can be achieved (Supplementary Movie 7). The local etching method is also applicable to the polymorphic design of other patterns (Supplementary Note 17 and Supplementary Figs. 45–57). As shown in Fig. 5f, in parallelogram patterns, there is an additional parameter tilt angle that define the geometric feature of patterns. In such patterns, the bottom etch will result in the free-end of the patterns being released at the end of the etch, when the structure will tend to assemble along the shorter diagonal or the free edge of width. Conversely, when the top etching start at the free-end first, and the localized etching will allow the assembly to develop along a longer diagonal section to satisfy the lowest elastic energy[16]. We have successfully utilized inclined etching to assemble the same parallelogram patterns along different diagonals, thus forming two types of structures with significant differences (Fig. 5f, Supplementary Figs. 53–57). It is worth noting that the premise for the preparation of polymorphic microstructures is that there are multi-stable strain energy states when the pattern is assembled, and there is an enough elastic energy barrier between the stable states. Otherwise, the pattern will only assemble into one type of structures (Supplementary Figs. 46–52). To summarize, we employed the multilevel design method based on the FEM etching and elastic mechanical model to achieve the 3D assembly of pre-strained nanomembranes, offering fresh opportunities for prospective device design through strain engineering.

**Angle-sensitive detection of incident light**
The multilevel structural design for nanomembranes is also applicable to guide the fabrication of micro/nanoscale electronic devices. A wide range of incident light angle detection of tubular photodetector has been observed, indicating that anisotropic photodetection can be realized via structural design. Here, incident light angle detection can be achieved by combining the angular response difference of a series of photodetectors. As shown in Fig. 6a, a hemispherical omnidirectional incident light controller is designed to achieve a photodetector with incident light traversing the spherical surface at the selected angle ($\theta$, $\varphi$). The controller consists of a transparent PMMA hemispherical shell and an array of optical fiber interfaces, in which the laser can be incident at a specific angle through the interface connected to the corresponding coordinates of the spherical surface. The prepared Si/Cr photodetector is placed on a platform at the same height as the bottom of the incident light controller. Then the controller is calibrated to ensure that the projected coordinate on YZ plane of (90°,0°) input laser port is aligned with the center of photodetectors (Fig. 6a,b, Supplementary Note 20 and Supplementary Figs. 60–61). Then, the collected photodetection data will be imported into neural network to analysis the incident angle (Fig. 6c). The Si/Cr photodetectors are prepared by a similar method to the Si/Cr bilayer while Cr layer is defined as a 5 μm channel in the middle of the pattern to realize the photodetection of the Si layer (Supplementary Figs. 58, 59). Si/Cr photodetectors exhibit a maximum responsivity of 60 mA/W, response time of 100 ~ 700 μs, and external quantum efficiency of 7 ~ 12%, which can effectively respond to 520 nm incident light to achieve photodetection (Fig. 6d, e, and Supplementary Figs. 62–64). After measurements of photoresponse of light incident from different coordinates, the photocurrent of each coordinate will be normalized and classified by structure types. To facilitate visual data comparison, the normalization photocurrent of each structure will be displayed through the projection of light controller on YZ plane. To aid in identifying the position of incident light on the projection sphere, we established spherical coordinates denoted by $\theta$ and $\varphi$. Compared to the unreleased Si/Cr planar photodetector of Fig. 6f, the ring (Fig. 6g) and tube photodetectors exhibit a wider detection angle of high photocurrent[8,23], confirming that the 3D microstructure contributes to the anisotropic detection of incident light and enhancing the detection of $\varphi$ in area of $\theta = 70°$–$110°$. For the arch photodetector (Fig. 6h), the increasing trend of photocurrent from $\theta = -0°$ to $\theta = 90°$, which derives from the out-of-plane bending of the arch structure, exhibit angle detection feature in $\theta$ direction, and significantly distinguish the incident light from $\theta < 90°$ and $\theta > 90°$. Helix photodetector exhibits an inclined high photocurrent detection angle from $\varphi = -30°$ to $\varphi = 0°$ because of the asymmetric rolling behavior via instability during the release process helping the angle identifying whether $\varphi < 0°$ or $\varphi > 0°$(Fig. 6i). In the taper photodetector, obvious high photocurrent appears in and $\varphi = 130°$–$140°$, $40°$–$50°$, benefiting from multiple reflections within a specific range (Fig. 6j), focusing on the detection in above area. Compared to ring structure, the tube photodetector exhibit a better stability in wide angle detection, which contributes to the detection of $\varphi$ in area of $\theta = 50°$–$130°$ (Fig. 6k). The results above illustrate the sensitivity of photoelectric coupling in 3D configurations and can be extended and utilized for functionalizing 3D-assembled nanomembrane electronic devices.

Neural networks have made significant advances in handling complex environmental information and ambiguous inference rules[53–55]. Thus, based on the difference in incident angle detection between the above structures, the deep neural network (DNN) algorithm can establish the angle-sensitive detection model based on photodetectors with different structures. A data set containing 275 photocurrents for each channel from Fig. 6f–k is collected to establish the incident angle prediction model. After training initial DNN models, two sets of photocurrent data are input for additional validation. As shown in Fig. 6l, training sets from the longitudinal ($\varphi = 0°$) and latitudinal ($\theta = 40°$) direction exhibit an accuracy of 95% and 78%, respectively. In Fig. 6m, the relationship between DNN-predicted incident angle and experimental angle demonstrates an accuracy of 83% and 71% for incident angle detection in an angle resolution of 10° from the longitudinal and latitudinal directions, respectively. As can be seen in Fig. 6n, 83% of the weight of the corresponding angle ($\varphi = 0°$) exhibit >10 times compared to the weight other angles, indicating the effectiveness of incident light detection. Profited from high accuracy prediction, these 3D self-assemble photodetectors show potential in wearable devices, intelligent furniture, and intelligent driving systems.

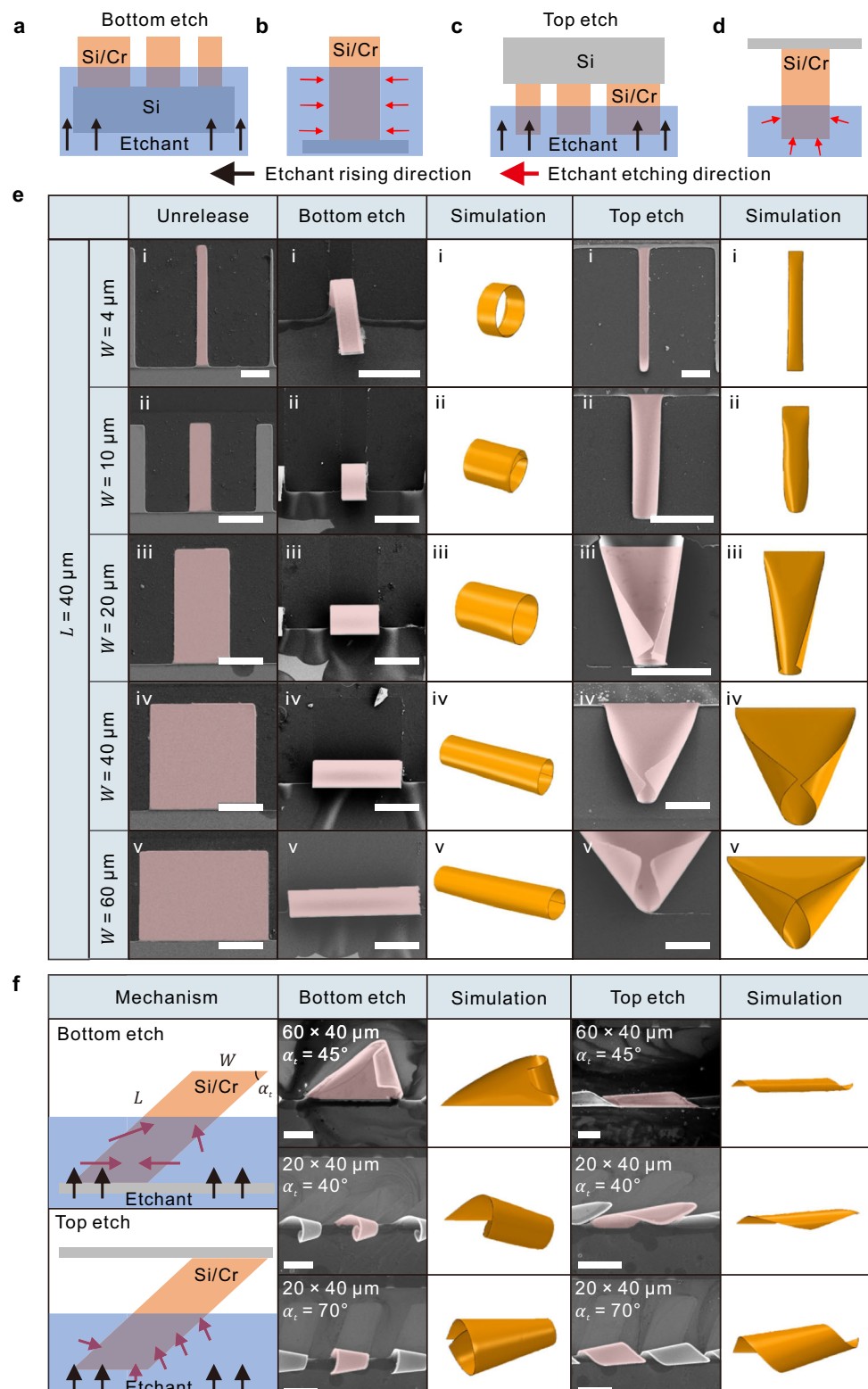

**Fig. 5 | Configurable structure assembly by controlling etching boundary.**
**a** Etchant rising and **b** etchant etching direction in adjacent side preferential etching. **c** Etchant rising and **d** etching direction in opposite side preferential etching. **e** Experimental and quasistatic multilevel FEM simulation results of Si/Cr bilayer with different sizes in unreleased condition, bottom etching condition, and top etching condition. **f** Mechanism, experimental results, and quasistatic FEM simulation results of multimorphic assembly in Si/Cr bilayer parallelogram patterns. Scale bars, 20 μm. Pink areas in SEM images are self-assembled structures. Source data are provided as a Source Data file.

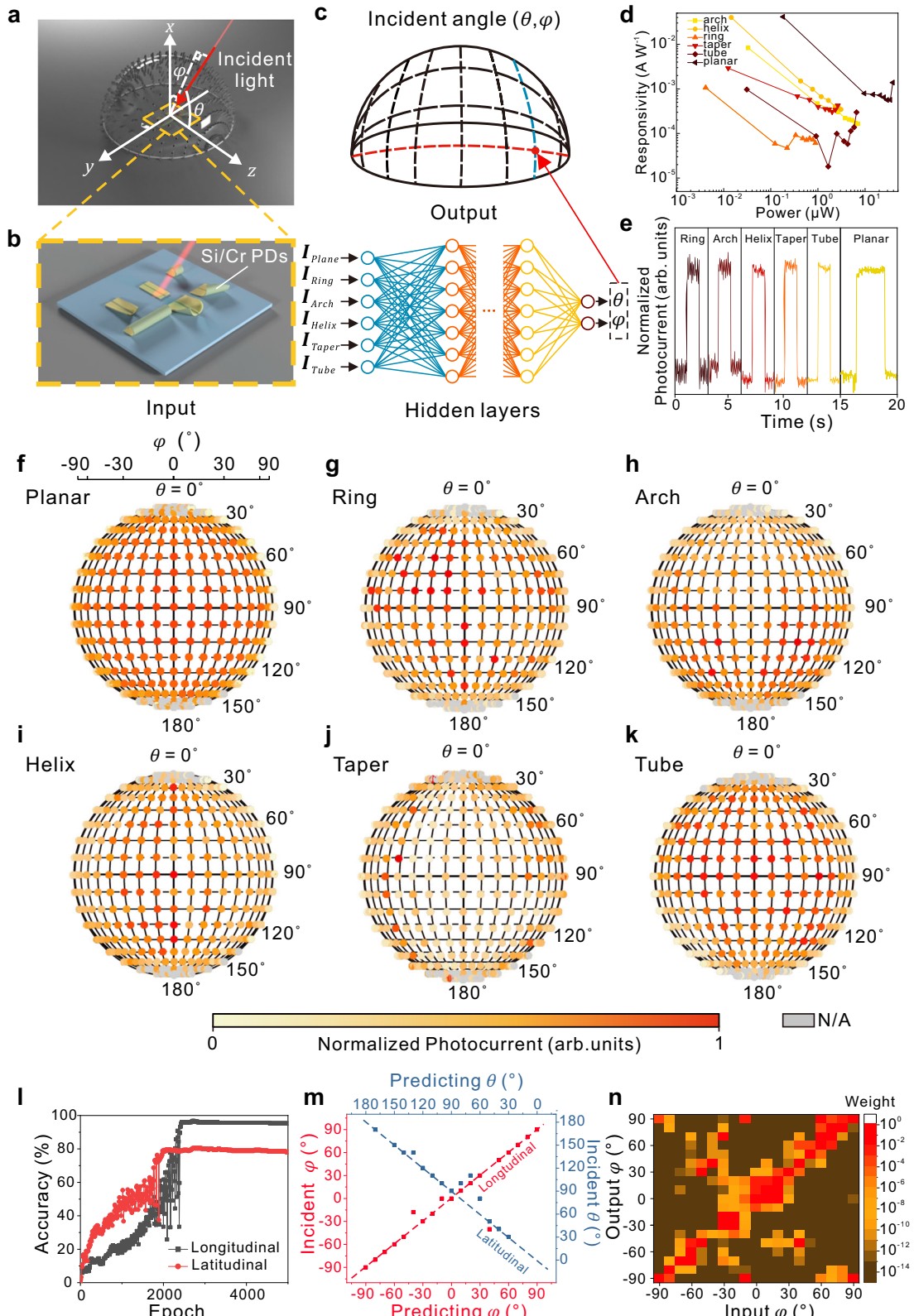

**Fig. 6 | DNN-assisted incident angle-sensitive photodetection. a** Schematic of the omnidirectional photoresponse measurement setup. A hemispherical PMMA light controller is aligned and placed on the photodetectors. **b** Si/Cr photodetectors at the spherical center of omnidirectional light controller. **c** Schematic of DNN-assisted analysis of the incident angle. **d** Relationship between responsivity and power of different photodetectors. **e** Current-Time relationship of different structures illuminated by 520 nm laser. XY plane projection of the photoresponse through the structure of **f** planar, **g** ring, **h** arch, **i** helix, **j** taper, and **k** tube photodetectors. **l** The epoch-dependent recognition accuracy of training set in longitudinal and latitudinal direction using DNN. **m** Angle identification of incident angle in longitudinal ($\varphi = 0°$) and latitudinal ($\theta = 40°$) direction. **n** Weight mapping of output angles and input angles of longitudinal direction ($\varphi = 0°$). Source data are provided as a Source Data file.

## Discussion

In summary, we have established a quasistatic multilevel FEM to simulate and predict the self-assembly 3D structure of pre-strain nanomembrane, solving the formation design problem under the coupling of complex etching trajectory and geometric pattern of 2D nanomembranes. Quasistatic multilevel FEM reproduces the dynamic release process by collecting coordinates of etching trajectory in chronological order and predict the transition of 3D structure varies with aspect ratio. This FEM method shows good applicability in a variety of material systems, precursor pattern types, pattern sizes, and nanomembrane thicknesses, which can also conduct preliminary geometric feature analysis of the self-assembly process and results. Si/Cr bilayer rectangular nanomembrane is adopted to verify the gradient change of structures. The analysis of elastic energy proves the correlation between aspect ratio, etching trajectory, and final morphologies. By controlling the spatiotemporal order of the etching process, diverse 3D structures can be realized in the same 2D pattern. The mechanism of multi-morphic construction can be explained by FEM modeling. Furthermore, as a proof-of-concept demonstration, six types of 3D nanomembranes are prepared, realizing incident light angle detection with the assistance of DNN. Our established quasistatic multilevel FEM lays a theoretical foundation for 2D nanomembrane self-assembly behavior, providing a construction paradigm for future optoelectronics, micro-electro-mechanical systems, and microrobots.

## Methods

### Finite element method simulation

Simulation of boundary movement of the sacrificial layer is computed by COMSOL Multiphysics 5.5. Deformation geometry, transport of diluted species, and lamina flow modules were applied to simulate fluid movement and etching process. To realize the reaction condition of $Ge/H_2O_2$ system, the default parameters were set as: $T = 343$ K, $D = 10^{-9}$ m$^2$ s$^{-1}$, $k_{343K} = 10^{-6}$, $c = 10$ mol L$^{-1}$, $\mu = 0.39$ Pa·s. Simulation of release process of Si/Cr bilayer is realized by ABAQUS CAE 2021. After the simulation of boundary movement, the mesh coordinates are collected. The whole etching process is sampled as $N \geq 80$ of time points at equal time distances. The coordinates of each time step will be set as the constraints of the geometric boundary of the corresponding time point on the bilayer model. Every boundary movement at the discrete time point will be regarded as a static analysis step[29,46]. In the subsequent analysis step, the geometric boundary constraint in the previous analysis step will be released to realize the continuous change of the sacrificial layer boundary with time. The static analysis step will be repeated cyclically until the last fixed boundary is removed. Linear elastic constitutive models were adopted for the Si and Cr layers. The Young's modulus of Si is 170 GPa, and the Poisson's ratio is 0.282[56] when the Young's modulus of Cr is 248 GPa, and the Poisson's ratio is 0.223[57]. The pre-stress difference between Si and Cr is set as −2000 MPa. Six-node 3D solid elements (C3D6) were used for bilayer models. Due to the large deformation during the release process, the geometric nonlinearity is considered. The effect of the steady/unsteady reaction-diffusion model in the $H_2O_2/Ge$ system is shown in Supplementary Note 21 and Supplementary Figs. 65, 66. For model with complex deformation and significant volume overlap, contact module is applied to ensure the accuracy of FEM simulations (Supplementary Note 22 and Supplementary Fig. 67).

### Fabrication of three-dimensional Si/Cr structures

The Ge/Si/Cr were grown by e-beam evaporation on 4-inch Si wafer with 500 nm $SiO_2$. Photoresist S1813 was spin-coated at 3000 rpm for 30 s, and etching windows were defined by the direct laser writing system. After development, the etching window was exposed, the Cr layer was wet etched using chromium etchant (HNO$_3$: cerium ammonium nitrate: H$_2$O = 1:3:16), and Si/Ge layer was then etched by reactive ion etching in condition of 18 sccm CHF$_3$ flow rate, 35 sccm SF$_6$ flow rate[58], 30 mT chamber pressure, and 50 W power. After reactive ion etching, the photoresist layer was removed in acetone via ultrasonic cleaning for 120 s. To release the Si/Cr bilayer from the Ge sacrificial layer, the samples were immersed in 30% $H_2O_2$ solution at 80 °C for 20−25 min. When the Si/Cr bilayer is released, critical point drying was utilized to remove acetone on the Si/Cr bilayer to prevent structural collapse caused by surface tension.

### Characterization of Si/Cr bilayer and photodetectors

Morphological characteristics of the released Si/Cr bilayer and photodetectors were characterized via SEM Zeiss Sigma 300. Optical images are captured via Zeiss Axiolab 5. The optoelectrical properties of Si/Cr photodetectors were performed by Keysight B2902B at temperature. Keysight 33500B waveform generator, Stanford Research Systems Model SR830 DSP lock-in amplifier, and Stanford Research Systems Model SR570 low-noise current preamplifier were utilized to extract photocurrent signals from current signals. TEM characterization and crystallographic analysis of the Si/Cr nanomembrane is performed by JEOL ARM200F (Supplementary Note 23 and Supplementary Figs. 68, 69).

### Setup of deep neural networks

To analyze the relationship between incident angle and photocurrent of photodetectors, we start with the hypothesis that the incident angle is two function f, g of the photocurrent ($I_{planar}$, $I_{ring}$, $I_{arch}$, $I_{helix}$, $I_{taper}$, $I_{tube}$), which can be written as:

$$\theta = f(I_{planar}, I_{ring}, I_{arch}, I_{helix}, I_{taper}, I_{tube}), \tag{3}$$

$$\varphi = g\left(I_{planar}, I_{ring}, I_{arch}, I_{helix}, I_{taper}, I_{tube}\right). \tag{4}$$

In the deep neural networks, the loss function and optimization algorithm are cross entropy and stochastic gradient descent, respectively. A multichannel data set contains 275 photocurrent data in each channel, which is collected from traversing measurements of various structures and introduced into a 7-layer hidden layer. Function f, g are independent of each other because they are orthogonal in the spherical coordinate system. Therefore, there is no correlation between the output of $\theta$ and $\varphi$.

## Data availability

All the data supporting the findings of this study are provided in the Supporting Information and Source Data file. Source data are provided with this paper.

## Code availability

The script supporting the findings of this study is provided in the Script file. The source code is available for download from CodeOcean at https://doi.org/10.24433/CO.5689289.v2. (ref. 59).

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

## Acknowledgements

This work is supported by the National Key Technologies R&D Program of China (2021YFA0715302 G.H. and 2021YFE0191800 Y.M.), the National Natural Science Foundation of China (62375054 Y.M. and 61975035 Y.M.), and the Science and Technology Commission of Shanghai Municipality (21142200200 Y.M. and 22ZR1405000 G.H.). Part of the experimental work was carried out in Fudan Nanofabrication Laboratory.

## Author contributions

These authors contributed equally: Z.Z., B.W. Y.M. conceived the project and designed experiments. Z.Z. and B.W. designed and conducted and data analysis; Z.Z., B.W., T.C., M.M. and X.L. conducted the experiments; Z.Z. and Y.W. conducted the finite element simulations; Z.Z., B.W., C.Y., G.J. T.C., and Y.H performed optoelectronic characterization. C.L. and Z.Z. performed the SEM characterization. Y.W. performed DNN-assisted incident angle recognition. Z.Z., B.W., G.H. and Y.M. co-wrote the manuscript. J.C., X.C., E.S., G.H. and S.K. discussed the results and commented on the manuscript.

## Competing interests

The authors declare no competing interest.
