## [Peer Review File · Nature Communications]

Multilevel design and construction in nanomembrane rolling for three-dimensional angle-sensitive photodetectionREVIEWER COMMENTS

Reviewer #1 (Remarks to the Author):

This manuscript reports the fabrication of self-assembled 3D structures of pre-strain nanomembranes based on the simulation using a quasistatic multilevel Finite Element Method (FEM). By resolving complex etching trajectory couplings and geometric patterns, the FEM tool effectively reproduces the dynamic release process and predicts structure transitions, demonstrating its utility in varied applications. In addition, they demonstrated diverse 3D structures from a single 2D pattern, and the successful incident light angle detection, indicates its practical viability for optoelectronics. Despite many reported results on 3D structures comprising membranes, this work introduces clear technical advancements through comprehensive simulations and demonstrates a specific application, offering valuable contributions to the related research field. Therefore, with necessary major revisions, this manuscript is recommended for publication, as it has the potential to significantly benefit and advance the field.

- In Figure 6, the process for fabricating a dome-shaped, 3D photodetector using Si/Cr bilayer membranes from the wafer is unclear. Specifically, the transformation of a 2D planar array into a 3D dome shape requires further elucidation. Also, the authors should provide a detailed explanation of the photoresponse mechanism of Si/Cr bilayers across different wavelengths. It is crucial to clarify whether a Si-Cr Schottky junction is utilized for photodetection, given the challenges in achieving high photo-efficiency without a pn junction. The authors are encouraged to include a description of the photoconversion efficiency of the fabricated Si/Cr photodetectors. Furthermore, the inclusion of movie files is recommended to effectively illustrate the functional capabilities of the 3D photodetection system.

- While the study primarily concentrates on Si/Cr bilayers, it is crucial to expand the material exploration for a more comprehensive assessment of the method's adaptability. Diversifying the materials under investigation would facilitate a more thorough evaluation of the approach's effectiveness across various compositions. This inclusive approach has the potential to yield insights into the fundamental principles, enriching scientific discussions and expanding the potential applications of the proposed method. Thus, it is strongly recommended to incorporate a broader range of materials, enhancing the overall depth and impact of the research findings.

Reviewer #2 (Remarks to the Author):

In this study, the authors propose a novel method utilizing quasistatic multilevel finite element modeling (FEM) to construct 3D structures from 2D nanomembranes. To validate their approach, they provide verification results using Si/Cr bilayer nanomembranes. The investigation reveals that the formation of 3D structures is influenced by both the minimum energy state and the geometric constraints imposed by the edges of the sacrificial layer. Furthermore, the authors present a compelling demonstration of large-scale, high-yield fabrication of these 3D structures, showcasing the potential practicality and efficiency of their proposed method. To explore the potential applications, the authors fabricate six variations of 3D Si/Cr photodetectors, which are capable of resolving the incident angle of light through the utilization of a DNN model. Overall, the authors' research holds great significance as it opens up exciting possibilities for the design and manufacturing methodologies of future optoelectronics and microrobots.

I think the work presented is of high quality and suitable for publication in Nature Communications.

I have just a few comments aimed at improving the paper.

1. There are multiple grammatical errors. Need to go through the manuscript and correct it.
2. It is well-known that multiple configurations obtained from the same precursor depend on the initial geometry and boundary conditions of 2D nanomembranes, but only rectangular shapes with different Width-Length ratios are considered. How about other shapes? Circular? The tilted strip?
3. What are the effects of the thickness of Si and Cr? I have not seen any discussion details of thickness and the related strain/stress gradient along thickness direction.
4. In the simulations of FEM, what is the setting details of interactions and contact conditions used in FEM? To what extent does this affect the simulation results?
5. Is there a size-correlated effect on the self-rolling behaviors of the initial 2D nanomembranes? Dose the conclusion still hold when the size of initial 2D nanomembranes being scaled up or down by a factor of 10?
6. Is it possible to achieve the real-time detection of the change of angle α during the etching process experimentally using high-speed imaging equipment? They can be compared with the FEM results and further validate the model parameters.

Reviewer #3 (Remarks to the Author):

This work presents a 3D assembly of 2D plane structures, guided by comprehensive FEA and well validated by extensive experiments. The remarkable agreement between FEA and experiments were obtained, regardless of dimensional sizes of films. Besides, exploration of assembled 3D structures with Si/Cr as photodetectors has also been explored to study the incidence angle of light with the help pf DNN, demonstrating an immediate application in a device level. Details of both FEA and experiments were well presented allowing the readers to follow and the results are reliable to be reproduced. Given the critical importance of 3D structures in design and nanomanufacturing of electronics and devices, particularly, structural materials made of traditional semiconductor material such as Si that was demonstrated in this work, the current multilevel FE approach-guided design and fabrication of 3D structures offer a powerful and low-cost means. I recommend the acceptance of this work after addressing the following minor issues.

- 1): the authors mentioned “the pre-strain nanomembrane”, I assume this pre-strain was induced by material lattice mismatch between two deposited layers and was not introduced by external stimuli or deposition process. Please clarify this concept.
- 2): During the releasing process, did the films remain a contact with etching solution or not? if not, does elastic energy during releasing need to consider the effect of liquid adhesion on the films, similar to the droplet evaporation-induced mechanical bending or folding of materials (doi.org/10.1039/C8SM00873F). When the geometric shape of films changes, such as from rectangular to circular or triangular one, did it affect the velocity of etchant flow or direction? if yes, how to unify the multilevel FE simulation boundary conditions?
- 3: How to determine whether the assembled 3D structures reached a final stable stage? And

with the quasi-static condition, could the local size of 3D structures such as local curvatures of taper or radius of tube during the etching process be predicted?

-----Responses to the reviewers' comments-----

=====

Reviewer #1

=====

This manuscript reports the fabrication of self-assembled 3D structures of pre-strain nanomembranes based on the simulation using a quasistatic multilevel Finite Element Method (FEM). By resolving complex etching trajectory couplings and geometric patterns, the FEM tool effectively reproduces the dynamic release process and predicts structure transitions, demonstrating its utility in varied applications. In addition, they demonstrated diverse 3D structures from a single 2D pattern, and the successful incident light angle detection, indicates its practical viability for optoelectronics. Despite many reported results on 3D structures comprising membranes, this work introduces clear technical advancements through comprehensive simulations and demonstrates a specific application, offering valuable contributions to the related research field.

Therefore, with necessary major revisions, this manuscript is recommended for publication, as it has the potential to significantly benefit and advance the field.

Our response:

Thank you very much for your suggestions and comments, which will bring significant help to the integrity and credibility of our work. Based on your suggestions, we have supplemented the experimental setup, mechanism, and device performance of the photodetection of 3D structures. At the same time, we conducted a detailed discussion on the design applicability of this model under different material systems, different patterns, and various nanomembrane thicknesses, proving that the method is applicable in fields such as photodetectors, inductance/capacitance devices, micro-robots, and micro-electromechanical systems.

**Comment:**

1: In Figure 6, the process for fabricating a dome-shaped, 3D photodetector using
Si/Cr bilayer membranes from the wafer is unclear. Specifically, the transformation of
a 2D planar array into a 3D dome shape requires further elucidation.

**Our response:**

Thanks for your comment. We apologize for the misunderstanding caused by
inadequate representation in our schematic and text. Re-Fig. 1 details how we set an
on-chip 3D structure photodetector array within a transparent hemispherical shell that
controls the angle of incident light. Using an on-chip 3D structure photodetector array
allows the incident light to be adjusted in 3D space by the semispherical shell,
ensuring the light beam is guided from the desired incident angle. To aid in
understanding, we have shown a magnified version of the on-chip 3D photodetectors
array in the diagram (Re-Fig. 1B), indicating that it is located at the $(90^\circ, 0^\circ)$
projection of the ZY plane of the semispherical shell. We also emphasized that so-
called 3D photodetection means the detection of incident light from various incident
angles in the 3D space by the on-chip photodetectors.

**Re-Fig. 1.** Revised schematic diagram of the 3D photodetection by Si/Cr
photodetectors. (A) Schematic of the omnidirectional photoresponse measurement
setup. A hemispherical PMMA light controller is aligned and placed on the
photodetectors. (B) Si/Cr photodetectors at the spherical center of the omnidirectional
light controller. (C) Schematic of DNN-assisted analysis of the incident angle. (D)
Relationship between responsivity and power of different photodetectors. (E) Current-

time relationship of different structures illuminated by 520 nm laser.

**Our modification:**

1. Fig. 6a-e has been revised as:

2. "The prepared Si/Cr photodetector is placed on a platform at the same height as the
 bottom of the incident light controller. Then the controller is calibrated to ensure that
 the projected coordinate on YZ plane of $(90^\circ, 0^\circ)$ input laser port is aligned with the
 center of photodetectors (Fig. 6a-b, Supplementary Fig. 63-64)." has been added in
 manuscript.

**Comment:**

2: Also, the authors should provide a detailed explanation of the photoresponse
 mechanism of Si/Cr bilayers across different wavelengths. It is crucial to clarify
 whether a Si-Cr Schottky junction is utilized for photodetection, given the challenges
 in achieving high photo-efficiency without a pn junction.

**Our response:**

Thank you very much for your valuable comments. The performance of
 photodetectors is an essential part of evaluating their application value, so it is
 necessary to conduct a comprehensive electrical performance characterization of them.

**Re-Fig. 2.** Electrical properties and photoelectric responsivity of Si/Cr photodetector.

(A) I-V curve and (B) $\log(I)$ - $\log(V)$ curve of arch-type photodetector. (C) $\ln(I/V)$ - \sqrt{V}

relationship curve of arch-type photodetector in the Poole-Frenkel emission region.

(D) Photoresponse of channel area and metal-semiconductor junction area of arch-

type photodetector. (E) I-V curves of arch-type photodetectors under 520 nm laser
 irradiation with different powers. (F) Responsivity-power relationship diagram of
 photodetectors with different structures. (G) Responsivity-wavelength relationship of
 arch-type photodetector. (H) Simulated absorbance of Ge/Si/Cr in planar SiO₂ (35
 85 nm)/Ge (50 nm)/Si (60 nm)/Cr (40 nm) multilayer.

 We characterized the photodetection properties of the prepared Si/Cr photodetector,
 as shown in Re-Fig. 2. We conducted an I-V test on the arch-type Si/Cr photodetector,
 which showed Schottky contact-dominated carrier transport behavior in the
 atmospheric environment (Re-Fig. 2A). According to the log(I)~log(V) curve, it can
 be seen that the device mainly exhibits three types of transmission behavior at 0~10 V,
 with slopes of 0.61, 0.99, and 1.02 respectively (Re-Fig. 2B). Among them, the slope
 is 0.61 in the region of 0.05~0.35 V. This is because when the voltage is small,
 carriers will be easily trapped by surface defects before being transported to the
 electrode, causing the $\frac{d \log(I)}{d \log(V)}$ less than 1. In the region of 0.35~1.35 V, the $\frac{d \log(I)}{d \log(V)}$ is
 0.99, and the current-voltage shows a linear relationship, but it does not indicate that
 Si/Cr exhibits ohmic contact. This is because the device exhibits the typical Poole-
 Frenkel (P-F) emission effect in Region III (1.35-10 V), which is only present in
 Schottky contacts. The effect mainly originates from the external electric field
 emitting the trapped electrons and holes to the continuous state related to the intrinsic
 dislocation of the Schottky barrier layer, and further directional movement of carrier
 generates current. When carriers in P-F emission enter the semiconductor material
 side from the Schottky contact metal material layer, they need to pass through the trap
 energy level in the potential barrier rather than through direct transition. At the same
 time, P-F emission mainly occurs inside the semiconductor material rather than at the
 metal-semiconductor interface. Its current density can be expressed as:

$$J = CE_b \exp \left[-\frac{q(\phi_B - \sqrt{qE_b/\pi\epsilon_0\epsilon_s})}{k_B T} \right],$$

 in which ϕ_B is the voltage barrier that electrons must cross when moving from one
 atom to another in a material in a zero electric field, k_B is Boltzmann's constant, ϵ_s is
 the relative dielectric constant of the barrier layer, T is the temperature, E_b is the
 external electric field strength, which is a constant. Taking the natural logarithm of
 both sides of this formula gives:

$$\ln (J/E_b) = \frac{q}{kT} \sqrt{\frac{qE_b}{\pi\epsilon_0\epsilon_s}} - \frac{q\phi_B}{kT} + \ln C.$$

There is

$$\ln (J/E_b) \propto \sqrt{E_b},$$

and combine the linear relationship of

$$J = \sigma E.$$

Multiply both ends by the cross-sectional area of the device A , we have

$$I = JA = \sigma AE = \sigma A \frac{V}{L} = k\sigma V,$$

where k is a constant. The current-voltage relationship and the current density-electric
 field intensity relationship can be linearly converted, so we can observe the
 relationship between $\ln (I/V) - \sqrt{V}$ to analyze the P-F emission in the device (Adv.
 Electron. Mater. 2018, 4, 1700567). As shown in Re-Fig. 2C, the $\ln (I/V) - \sqrt{V}$
 curve in Region III shows excellent linearity, confirming that the device is a Schottky
 contact. Therefore, the linear region in Region II is mainly related to non-ideal factors
 in the device. Due to the large series resistance, the exponential current characteristics
 have not yet been demonstrated in this voltage region. At this time, the Schottky
 junction I-V characteristic equation will be written as (S. M. Sze, Physics
 Semiconductor Devices, 4th ed. Wiley, New York, 2021.):

$$I = I_0 \left[\exp \left(\frac{q(V-IR_s)}{nkT} \right) - 1 \right],$$

$$I_0 = AA^*T^2 \exp (-q\phi_b/kT),$$

when $V \sim IR_s$ exists, the exponential term of the above formula can be expanded as:

$$I \sim I_0 \cdot \left(\frac{q(V-IR_s)}{nkT} + 1 \right) - 1 = I_0 \cdot \frac{q(V-IR_s)}{nkT},$$

then $I = \frac{V}{\frac{nkT}{I_0q} + R_s} = c_0V$, where c_0 is a constant. Taking the logarithm of both sides gives

$\log(I) = \log(V) + \log(c_0)$, in which the slope is $\frac{d \log(I)}{d \log(V)} = 1$. Therefore, the larger

series resistance may be the reason for the Region II slope of 1. Due to the lack of C-
 V testing and variable temperature testing platforms, we are currently unable to
 measure the barrier height of the Si-Cr Schottky contact. We hope that corresponding
 characterization methods can be introduced in subsequent work for further analysis.

Since the voltage in Region I is small, a large number of carriers cannot be moved
 into the electrode area for collection, and the voltage in Region III is large, which will
 bring a large dark current and noise that affects photodetection. Therefore, we chose

the operating voltage of the photodetector to be 1 V.

Then, we analyzed the mechanism of the photodetector. The photodetection of this
device is mainly based on the resistance change caused by the generation of carriers in
the silicon nanomembrane under light rather than the built-in electric potential field of
the Schottky junction. We conducted photoresponse tests on the channel area and the
Si/Cr Schottky contact area. It can be seen that only the silicon channel area shows an
obvious response to light, while the Si-Cr Schottky contact area exhibits no response
to light, which means that the Schottky junction is not mainly involved in the
photodetection process, and photo carriers mainly originate from the silicon
nanomembrane channel (Re-Fig. 2D). Although commercial photodetectors are
mainly photovoltaic devices based on p-n junctions, photoconductive photodetectors
are also widely used in the direction of photosensitive elements. In the follow-up
work, we aim to further optimize device design, develop CMOS technology, and
prepare microstructured photodetectors with higher performance.

The thickness of our photodetector is at the nanometer scale, which will exhibit a
weak response in mid-infrared wavelength. Therefore, we focused on studying the
responsivity-wavelength relationship in the visible spectrum. We tested the
responsivity-wavelength relationship of the arch-type Si/Cr photodetector, as shown
in Re-Fig. 2G. It can be seen that the detector can effectively detect the visible light
range, and its responsivity is 20~60 mA/W in the wavelength range of 400-750 nm.
The responsivity shows multiple peaks, which is different from bulk Si devices. The
reflection at the silicon oxide/silicon substrate interface will cause periodic
fluctuations in the absorption of the Si/Cr detector. We simulated the SiO₂ (35 nm)/Ge
(50 nm)/Si (60 nm)/Cr (40 nm) system and found that the absorption rate of Ge/Si/Cr
is similar to our device responsivity wavelength relationship, confirming that the
nanomembrane system has a certain phase length interference effect on specific
wavelength incident light (Re-Fig. 2H). Due to the local rolling of the arch structure,
its response band will differ from that of the planar multilayer nanomembrane system.

Re-Fig. 3. Response time of (A) ring, (B) arch, (C) helix, (D) taper, (E) tube, and (F) planar Si/Cr photodetectors.

For various 3D structured Si/Cr photodetectors, the relationship between the
responsivity and power is as shown in the Re-Fig. 2F. The Si/Cr 3D photodetectors
can detect incident light under weaker light, thanks to their 3D structures. The
responsivity of photodetectors can reach up to 40 mA/W, which is slightly lower than
commercial silicon-based photodetectors (Re-Fig. 2E). However, various types of
Si/Cr photodetectors, including planar structures, exhibit a decrease in responsivity as
the power density of the incident light source increases. This may be related to the
recombination of photogenerated charge carriers or the enhanced scattering rate as the

carrier concentration increases at higher optical power densities. We tested the
responsivity-wavelength relationship of the arch-type Si/Cr photodetector, as shown
in the Re-Fig. 2F. It can be seen that the detector can effectively detect the visible
light range, and its responsivity will gradually increase as the wavelength increases.
At the same time, we tested the responsivity of the Si/Cr photodetector, as shown in
the figure (Re-Fig. 4). The ring, arch, helix, taper, tube, and planar structure
photodetectors showed rise times of 120, 138, 389, 758, and 192 μs respectively and
decay times of 10, 688, 180, 624, and 197 μs , respectively. The response time of
photodetectors is slower than commercial silicon-based detectors and is primarily
limited by the lower carrier mobility and presence of impurities in amorphous silicon.
It is worth noting that in nanomembrane, the thickness of the semiconductor material
is very thin so that the surface defects will dominate the performance of the device,
and the nanomembrane released from the sacrificial layer will be contaminated by
impurities on both the front and back sides. The above disadvantages will further
affect the response time of the photodetectors. Therefore, surface engineering of self-
assembled structures will become an important research direction in our subsequent
work to improve the performance of 3D photodetectors.

**Our modification:**

- 1. Re-Fig. 2 has been added as Supplementary Fig. 60.
- 2. Re-Fig. 3 has been added as Supplementary Fig. 61.
- 3. The answer has been added as "19. Photodetection mechanism and performance of
Si/Cr photodetectors" in Supporting Information.
- 4. "Si/Cr photodetectors exhibit a maximum responsivity of 60 mA/W, response time
of 100~700 μs , and external quantum efficiency of 7~12%, which can effectively
respond to 520 nm incident light to achieve photodetection (Supplementary Fig. 60-
61)." has been added in manuscript.

**Comment:**

3: *The authors are encouraged to include a description of the photoconversion*
*efficiency of the fabricated Si/Cr photodetectors. Furthermore, the inclusion of movie*
*files is recommended to effectively illustrate the functional capabilities of the 3D*
*photodetection system.*

**Our response:**

Thank you again for your suggestion. For the photodetectors, their photoconversion
efficiency is usually characterized by external quantum efficiency (EQE). We
obtained the EQE of the photodetectors through the responsivity calculation. The
calculation formula is as follows:

$$EQE = R_{\lambda} \cdot \frac{h\nu}{e} \times 100\%$$

**Re-Fig. 4.** EQE-wavelength relationship of the arch-type photodetector.

It fluctuates in the range of 7~12% (Re-Fig. 4), which can effectively detect incident
light, but there is still room for improvement. The low responsivity and quantum
efficiency of photodetectors are mainly caused by the following reasons. First, the
silicon nanomembrane that constitutes the photodetector is thinner and absorbs fewer
carriers generated by incident light, resulting in a smaller photocurrent. Secondly, the
silicon nanomembrane grown by electron beam evaporation is amorphous, and wet
etching during the self-assembly process will introduce additional contamination at
the nanomembrane interface. The amorphous structure and a large number of surface
states will significantly affect the migration rate of photogenerated carriers from the
channel to the electrode and their recombination before they are collected by the
electrode, thus reducing the responsivity and prolonging the response time. Our
research on this type of 3D structured photodetector mainly focuses on its angle
detection and identification, and there is a lot of room for improvement in
performance optimization. We have compiled some performance of visible light band
detector devices reported in commercial and previous studies (Re-Table 1).

Re-Table 1. Comparison of photodetection performance of this work with other photodetectors.

Material	Detection mechanism	Detection range	Responsivity	Photoconversion efficiency	Response time/response frequency	Source
Si (60 nm)	Photodiode	400-750 nm	52.5 mA/W @ 530 nm	12.2% @ 530 nm (EQE)	138 μs (rise)/68 μs (decay)	This work
Si(bulk)	Photovoltaic	350-1100 nm	640 mA/W @ 980 nm	82.2% @ 980 nm (EQE)	10 ns (rise)/10 ns (decay)	THORLABS FDS100-CAL
Si/Graphene	Photovoltaic	1300-1600 nm	30 mA/W @ 1550 nm	10% @ 1550 nm (IQE)	~18 GHz	Nat. Photonics. 2013, 7, 892
Si (bulk)	Photodiode	400-700 nm	~4 A/W @ 540 nm	~900% @ 540 nm	30 ms	JCHL GL48569 photoresistor
Silicon (300 nm)	Photovoltaic	400-800 nm	80 mA/W @ 600 nm	-	9 GHz	IEEE Electron Device Lett. 2022, 43, 1077.
Si(bulk): Ag	Photodiode	400-1200 nm	1.71 A/W @ 800 nm	266% @ 800 nm (EQE)	12.5 μs (rise)/15.9 μs (decay)	Adv. Opt. Mater. 2018, 6, 201700638
CdTe nanoparticles	Photodiode	365-800 nm	4 mA/W (visible wavelength)	-	134 μs	J. Optics 2023, 1. doi.org/10.1007/s12596-023-01165-2
β-Ga ₂ O ₃	Photodiode	200-	61.3 A/W	30000%	3 ms (rise)/35	Appl. Phys.

onducti	300	@254 nm	@254	ms (decay)	Lett. 2023,
ve	nm		nm		123, 041103.

In the follow-up research work, we also hope to improve the detection performance of
the 3D photodetector further through dry etching, passivation layer coating, and high-
quality thin nanomembrane epitaxy. In addition, the free-standing self-rolled
microstructure has excellent potential in long-wave infrared detection. The
freestanding structure can be realized by using a simple one-step self-rolling process,
thereby avoiding the complicated process flow in the traditional bolometer
manufacturing process (Rogalski, A. *Infrared detectors*. CRC press, (2000).).

In order to facilitate readers' understanding, we have supplemented the process of
setting up the test platform. As shown in the Re-Fig. 5, the laser is driven by DC
power supply 1, in which the signal generator generates a square wave signal to
modulate the laser to achieve the emission of optical signals of a specified frequency.
Currently, the Si/Cr photodetector driven by another DC power source detects the
optical signal emitted by the laser and generates the current signal. The current signal
will be input into the lock-in amplifier and the oscilloscope respectively through the
preamplifier, where the lock-in amplifier is used to record the photocurrent intensity,
and the oscilloscope is used to record the photoresponse waveform and measure the
response time. First, we fix the 3D photodetector to be tested on the optical platform,
which has been connected to the PCB circuit board through the bonding machine, and
the PCB circuit board is connected to the semiconductor test unit (Re-Fig. 6A-C). The
optical image of the omnidirectional light controller is shown in Re-Fig. 6D. It
consists of fixed grooves distributed according to spherical coordinates and a
detachable SMA optical fiber interface. The fixed grooves are distributed at intervals
of 10° according to longitude and latitude, and they interface with the SMA optical
fiber. After the combination, the laser can be connected to perform photoelectric
testing on the sample in the center of the spherical shell.

Then, we place the incident light controller on the optical platform and perform XY
plane alignment and Z-axis alignment, respectively, to ensure that the incident light
(90° , 0°) orientation is located in the center of the photodetectors on the PCB board
and align the height of the z-axis at the bottom of the controller with the
photodetectors. Then, connect the 520 nm laser to the required SMA port and use a
semiconductor measurement unit to provide voltage to the sample for photodetection

of the incident light.

**Re-Fig. 5.** Schematic diagram of 3D structure Si/Cr photodetector testing equipment.

**Re-Fig. 6.** Testing Setup for 3D structured photodetector. (A) Positioning of device
and controller support platform. (B) Connection of devices, test boxes, and test units.
(C) Test unit connection and schematic diagram. The (D) XY plane of the incident
light direction controller is aligned with the (e-f) Z axis. (G) Schematic diagram of
photodetection testing process.

**Our modification:**

1. Re-Fig. 4-6 has been added as Supplementary Fig. 62-64.

- 2. Re-Table 1 has been added as Supplementary Table 2.
- 3. The answer has been added as a part of "19. Photodetection mechanism and
performance of Si/Cr photodetectors" and "20. Setup of incident light directional
detection" in Supporting Information.
- 4. "Si/Cr photodetectors exhibit a maximum responsivity of 60 mA/W, response time
of 100~700 μ s, and external quantum efficiency of 7~12%, which can effectively
respond to 520 nm incident light to achieve photodetection (Supplementary Fig. 60-
61)." has been added in manuscript.
- 5. "The prepared Si/Cr photodetector is placed on a platform at the same height as the
bottom of the incident light controller. Then the controller is calibrated to ensure that
the projected coordinate on YZ plane of (90°, 0°) input laser port is aligned with the
center of photodetectors (Fig. 6a-b, Supplementary Fig. 63-64)." has been added in
manuscript.

**Comment:**

4: *While the study primarily concentrates on Si/Cr bilayers, it is crucial to expand the*
*material exploration for a more comprehensive assessment of the method's*
*adaptability. Diversifying the materials under investigation would facilitate a more*
*thorough evaluation of the approach's effectiveness across various compositions. This*
*inclusive approach has the potential to yield insights into the fundamental principles,*
*enriching scientific discussions and expanding the potential applications of the*
*proposed method. Thus, it is strongly recommended to incorporate a broader range of*
*materials, enhancing the overall depth and impact of the research findings.*

**Our response:**

Thank you very much for your insightful comments. For the multilevel FEM model,
its universality to different material systems will be important for the future
development of electronic devices and microrobots. Therefore, we selected three other
representative material systems in micro-nano devices for feasibility verification: low-
frequency/high-frequency silicon nitride (HF/LF SiN_x) for optical resonators and
capacitors (Nat. Electron. 2018, 1, 305; Nanotechno. 2019, 30, 364001.), NiTi
nanomembranes for microrobots (Adv. Mater. Technol. 2019, 4, 1900583.), and
VO₂/Cr nanomembrane for bolometers and actuators (Sci. Adv. 2023, 9, eadi7805;

Nat. Commun. 2022, 13, 7819.). These three systems include several on-chip and off-
 chip applications realized by micro-nano 3D structures, and there are also great
 differences in strain states. The experimental results are also consistent with the
 simulation of the FEM model. We supplement the above results into the manuscript
 and Supporting Information, which confirms the universality of the model and further
 broadens the application field of the method.

 **Re-Fig. 7.** SEM images of self-assembled SiN_x nanomembrane and FEM results of
 the multilevel design model. (A-B) Panorama SEM images of self-assembled silicon
 nitride nanomembranes. (C) 4×20 μm², (D) 10×20 μm², (E) 20×20 μm², (F) 24×20
 μm², (G) 36×20 μm², (H) 60×20 μm², SEM images of self-assembled SiN_x
 nanomembrane. FEM simulation results of self-assembled SiN_x nanomembranes with
 sizes of (I) 4×20 μm², (J) 10×20 μm², (K) 20×20 μm², (L) 24×20 μm², (M) 36×20 μm²,
 (N) 60×20 μm².

First, we will demonstrate self-assembly based on HF/LF SiN_x nanomembranes (Re-
Fig. 7). We take this system into the multilevel FEM model for analysis, and the result
shows a transition of ring-taper-tube structure. As can be seen from the SEM images,
the system based on dual-frequency silicon nitride exhibits a ring-helix-taper-tube
structure with size from 2×20 μm² to 80×20 μm² (2-20/22-28/30-44/46 -80 μm).
Among them, the structural characteristics of the helix are mainly reflected in the
asymmetry of the rolling process. We obtained experiment results that are highly
consistent with the FEM model, verifying that the structure of self-assembled HF/LF
SiN_x is controlled by the boundary and size of the sacrificial layer at the same time.
The multilevel model provides an effective means for high-precision design and
potential performance control of microstructure inductors and optical microcavity
devices.

**Re-Fig. 8.** SEM images of self-assembled NiTi alloy nanomembrane and FEM results
of multilayer design model. (A) Panorama SEM image of self-assembled NiTi
nanomembranes. SEM images of self-assembled nanomembranes with sizes of (B)
30×40 μm², (C) 38×40 μm², (D) 42×40 μm², (E) 60×40 μm², and (F) 72×40 μm².
FEM simulation results of self-assembled nanomembranes with sizes of (G) 4×40
μm², (H) 10×40 μm², (I) 20×40 μm², (J) 40×40 μm², and (K) 60×40 μm².

Subsequently, we demonstrated the preparation of self-assembled structures of NiTi
alloy nanomembranes (Re-Fig. 8). The usual preparation method of NiTi alloy
microstructure is based on oblique deposition of photoresist sacrificial layer. Among
them, grazing angle deposition can provide a fixed end and a free end for the
photoresist through the shadow effect (Sci. Adv. 2018, 4(4), eaap8203). We use the
difference in deposition rate during electron beam evaporation of NiTi
nanomembranes to introduce strain differences and use the interlayer strain gradient
to achieve self-assembly of the nanomembranes. Since the etching rate of photoresist
in acetone is extremely fast and it is an etching model dominated by the diffusion
process, the boundary conditions of the sacrificial layer will be similar to the Si/Cr
system. In the FEM model, it is worth noting that under large strains (2~3%, 2.5% for
FEM model, Adv. Mater. Technol. 2019, 4, 1900583.), even if the boundary
conditions cause obvious long-side rolling at the lateral sides. When the patterns are
fully released, they still roll unidirectionally from the short end due to the deformation
limit of the fixed end, resulting in most of the self-assembly, and the results are all
unidirectional tubes. In the experiment, the structures we obtained are all ring-tube
structures, and few helix-type structures existed, which verified the applicability of
this model to situations where one end is dominant under diffusion-reaction
equilibrium. Introducing this multilevel design method will facilitate the mass
production of microrobots and the refined design of actuators.

**Re-Fig. 9.** SEM images of self-assembled VO₂/Cr nanomembrane and FEM results of
the multilayer design model. (A-B) SEM images of 40×400 μm² self-assembled

VO₂/Cr nanomembranes. (C) SEM image of 200×200 μm² self-assembled VO₂/Cr
nanomembrane. FEM simulation results of self-assembled nanomembranes with sizes
of (D-E) 40×400 μm² and (F) 200×200 μm².

Finally, we used VO₂, a type of heat-sensitive functional material, combined with the
Cr layer for verification (Re-Fig. 9). We studied the self-assembled 120 nm VO₂/30
386 nm Cr system and conducted structural design and prediction of the self-assembled
VO₂ nanomembrane patterns of 200×200 μm² and 50×500 μm². Due to the large
thickness of vanadium oxide, the radii of its ring and tube will increase accordingly.
However, patterns with an aspect ratio = 1:1 still exhibit a tube structure. This is
because the VO₂ strain is relatively large (0.7~1%, 0.8% for FEM) (Nat. Commun.
2022, 13, 7819.), which can still cause a significant strain gradient even when the Cr
layer is thin, allowing the taper to overcome both ends. The competitive energy
barriers of multidirectional rolling in the etching process are finally realized to
assemble the tubular structure. Among them, the 200×200 μm² self-assembled
nanomembrane exhibits the shape of taper and tube, most of which are tube-type
structures.

We have also previously studied the Gaussian-protected polymorphic preparation of
VO₂/Cr nanomembranes (Nat. Commun. 2021, 12, 509.), and we hope that this model
can be used to predict the switching of strain states after self-assembly.

In addition, our model can also design and predict different material thickness ratios
and different pattern designs under the same material system. For the Si/Cr system,
we designed multiple groups of Si/Cr nanomembranes with different thicknesses.
Based on this, we predicted the Si/Cr self-assembly system under different strain
states and successfully predicted the radius change of tubular structures under
different strain difference states. (Re-Fig. 10). We designed Si/Cr systems with
different parameters corresponding to Si nanomembranes (15/30/90 nm) and Cr
nanomembranes (10/20/60 nm) of different thicknesses. We number the above
samples as follows:

**Re-Table 2.** Si/Cr nanomembrane sample numbers and corresponding parameters.

Sample number	Si thickness (nm)	Cr thickness (nm)
60	10
60	20

60	60
15	40
30	40
90	40
Standard	60	40

**Re-Fig. 10.** Diameter and strain distribution of Si/Cr self-rolled microtubes under
 different nanomembrane thicknesses. (A) FEM simulation, theoretical, and
 experimental results of the relationship between Si/Cr microtube radius and Cr
 nanomembrane thickness. (B) FEM simulation, theoretical, and experimental results
 of the relationship between Si/Cr microtube radius and Si nanomembrane thickness.

The error bars are the standard deviation of radii in experiments.

As can be seen from Re-Fig. 10, the radii of the Si/Cr nanomembrane sample under
 different thickness combinations show a high degree of consistency in the FEM model,
 theory, and experiment, indicating that the model has the ability to conduct
 preliminary quantitative analysis of geometric characteristics.

**Re-Fig. 11.** Schematic diagram of the etching process of different shapes. Geometric
 parameters of (A) semicircle, (B) isosceles right triangle, and (C) parallelogram strip.

**Re-Fig. 12.** SEM images of self-assembled triangles and FEM results of the
 multilevel design model. (A-B) Panorama SEM images of self-assembled triangles.
 SEM images of self-assembled triangles with sizes of (C) $D = 24 \mu\text{m}$, (D) $D = 52 \mu\text{m}$,
 (E) $D = 100 \mu\text{m}$, and (F) $D = 110 \mu\text{m}$. Triangular FEM simulation results with sizes of
 (G) $D = 24 \mu\text{m}$, (H) $D = 52 \mu\text{m}$, (I) $D = 100 \mu\text{m}$, and (J) $D = 110 \mu\text{m}$.

**Re-Fig. 13.** SEM images of self-assembled semicircle and FEM results of the
 multilevel design model. (A) Panorama SEM images of a self-assembled semicircle.
 SEM images of self-assembled semicircles with sizes of (B) $D = 20 \mu\text{m}$, (C) $D = 28$
 440 μm , (D) $D = 54 \mu\text{m}$, and (E) $D = 70 \mu\text{m}$. FEM results of self-assembled semicircles
 with sizes of (F) $D = 20 \mu\text{m}$, (G) $D = 28 \mu\text{m}$, (H) $D = 54 \mu\text{m}$, and (I) $D = 70 \mu\text{m}$ of
 self-assembled semicircular FEM simulation results.

 **Re-Fig. 14.** SEM images of self-assembled parallelogram strips with different widths
 and FEM results of the multilevel design model. (A-B) Panorama SEM images of
 self-assembled parallelogram strip nanomembranes with different widths. SEM
 images of self-assembled parallelogram strip nanomembrane with sizes of (C) 4×40
 449 μm^2 , $\alpha_t = 45^\circ$, (D) $16 \times 40 \mu\text{m}^2$, $\alpha_t = 45^\circ$, (E) $30 \times 40 \mu\text{m}^2$, $\alpha_t = 45^\circ$, and (F) 80×40
 450 μm^2 , $\alpha_t = 45^\circ$. FEM simulation results of self-assembled parallelogram strip nanomembrane
 with sizes of (G) $4 \times 40 \mu\text{m}^2$, $\alpha_t = 45^\circ$, (H) $16 \times 40 \mu\text{m}^2$, $\alpha_t = 45^\circ$, (I) $30 \times 40 \mu\text{m}^2$, $\alpha_t = 45^\circ$,
 and (J) $80 \times 40 \mu\text{m}^2$, $\alpha_t = 45^\circ$.

**Re-Fig. 15.** SEM diagram of self-assembled parallelogram strips. (A) Panorama SEM
 images of self-assembled parallelogram strips. SEM images of parallelogram strips
 with sizes of (B-I) $\alpha_t = 10^\circ$ - 80° . FEM results of parallelogram strips with sizes of (J-Q)
 $\alpha_t = 10^\circ$ - 80° .

For different pattern designs, we tried triangles, semicircles, and various
 parallelograms (Re-Fig. 11-15). In triangles and semicircles, the multilevel model
 only gets tubular structures. In parallelograms with different angles and aspect ratios,
 the model is also designed by designing corresponding etching paths, in which the
 asymmetry in pattern leads to the assembly of helix structure, and the experimental
 results once again verified the feasibility of this design and process.

In summary, our model demonstrates broad applicability to multiple material systems,
 deposition processes, and pattern designs. We believe that the multilevel FEM method
 will be valuable for future electronic devices, microelectromechanical systems,
 microrobots, and other electronic devices. The design and the prediction provide a

high-precision, low-cost method to provide 3D integration solutions for next-
generation devices.

**Our modification:**

1. Re-Fig. 7-9 has been added as Supplementary Fig. 15-17.

2. Re-Table 2 has been added as Supplementary Table 1.

3. Re-Fig. 11 has been added as Supplementary Fig. 7.

4. Re-Fig. 12-15 has been added as Supplementary Fig. 11-14.

5. The answer has been added as a part of "4. FEM modeling for different patterns",
"5. FEM modeling for various material systems" and "8. FEM modeling for different
Si/Cr thickness" in Supporting Information.

6. "In addition to the Si/Cr rectangular patterns, the multilevel FEM model has been
designed for various types of patterns such as semicircular, triangular, and
parallelogram strips (Fig. 2d, Supplementary Fig. 7-14, and Movie S1 and S2), as well
as in the high-frequency/low-frequency SiN_x nanomembrane, NiTi nanomembrane,
and VO₂/Cr nanomembrane systems, which demonstrate the potential for a wide
range of applications (Supplementary Fig. 15-17)." has been added in manuscript.

7. "In this study, we choose the rectangular Si/Cr rectangular nanomembrane for
detailed analysis and discussion of the self-assembly behavior due to their moderate
parameter complexity and high geometrical symmetry." has been added in manuscript.

8. Fig. 2d-h of manuscript has been revised and added as follow:

=====

**Reviewer #2**

=====

*In this study, the authors propose a novel method utilizing quasistatic multilevel finite*
*element modeling (FEM) to construct 3D structures from 2D nanomembranes. To*
*validate their approach, they provide verification results using Si/Cr bilayer*
*nanomembranes. The investigation reveals that the formation of 3D structures is*
*influenced by both the minimum energy state and the geometric constraints imposed*
*by the edges of the sacrificial layer. Furthermore, the authors present a compelling*
*demonstration of large-scale, high-yield fabrication of these 3D structures,*
*showcasing the potential practicality and efficiency of their proposed method. To*
*explore the potential applications, the authors fabricate six variations of 3D Si/Cr*
*photodetectors, which are capable of resolving the incident angle of light through the*
*utilization of a DNN model. Overall, the authors' research holds great significance as*
*it opens up exciting possibilities for the design and manufacturing methodologies of*
*future optoelectronics and microrobots.*

*I think the work presented is of high quality and suitable for publication in Nature*
*Communications. I have just a few comments aimed at improving the manuscript.*

**Our response:**

Thank you very much for your recognition of our work. The manuscript will be
carefully checked to eliminate grammatical errors. We conducted a detailed
discussion on the design and mechanism of the FEM in nanomembrane systems with
various patterns, thicknesses, and sizes, confirming the broad applicability of this
analysis method to self-assembly in various situations. We also reaffirmed the control
of etching spatiotemporal order in other patterns to achieve multi-morphic assembly.
At the same time, we demonstrate the analysis of tube radius in FEM results, theory,
and experimental results, which show a high degree of consistency and provide data
for detailed strain state analysis and discussion. Additionally, we greatly appreciate

your insights on the FEM contact module and analysis of nanomembrane dynamic
assembly. We have demonstrated the feasibility of applying the contact module to the
model to improve the simulation accuracy of complex structures and to track the
assembly process using quasistatic methods. We are also looking forward to further
research and improvement of the multilevel method in subsequent work.

**Comment:**

*1: There are multiple grammatical errors. Need to go through the manuscript and*
*correct it.*

**Our response:**

Thank you very much for your comments. We apologize for the grammatical
problems and unsmooth sentences in the text. We have thoroughly checked the
manuscript and Supporting Information to minimize grammatical errors and resolve
spelling issues.

**Our modification:**

1. In Page 2, Lines 23, the word “releasing” have been revised to “**release**”.

2. In Page 2, Lines 24, the word “nanomembrane” have been revised to
“**nanomembranes**”.

3. In Page 2, Lines 24-26, the sentence “Here, we propose a quasistatic multilevel
finite element modeling (FEM) to assemble 3D structures from 2D nanomembrane
and offer verification results by Si/Cr bilayer nanomembranes.” have been revised to
“Here, we propose a quasistatic multilevel finite element modeling (FEM) to
assemble 3D structures from 2D **nanomembranes** and offer verification results by
**various** bilayer nanomembranes.”.

4. In Page 2, Lines 26-29, the sentence “We confirm that the 3D structural formation
is governed by both the minimum energy state and the geometric constraints imposed
by the edges of sacrificial layer.” have been revised to “**Take Si/Cr nanomembrane as**
**an example, we** confirm that the 3D structural formation is governed by both the
minimum energy state and the geometric constraints imposed by the edges of **the**

sacrificial layer.”.

5. In Page 3, Lines 42, the word “and/or” have been revised to “**and**”.

6. In Page 3, Lines 61, the word “considered as preference” have been revised to
“**regarded as a preference**”.

7. In Page 3, Lines 62, the word “dependence” have been revised to “**the dependence**”.

8. In Page 3, Lines 64, the word “analysis on” have been revised to “**analysis in**”.

9. In Page 4, Lines 68, the word “geometry” have been revised to “**the geometry**”.

10. In Page 4, Lines 71, the word “sacrificial layer” have been revised to “**the**
**sacrificial layer**”.

11. In Page 4, Lines 80, the words “high uniformity” have been revised to “**high-**
**uniformity**”.

12. In Page 4, Lines 92, the word “substrate” has been revised to “**the substrate**”.

13. In Page 4, Lines 94, the words “as gradual movement” have been revised to “**as a**
**gradual movement**”.

14. In Page 5, Lines 97, the words “at solid-liquid interface” have been revised to “**at**
**the solid-liquid interface**”.

15. In Page 5, Lines 102, the words “with increase in aspect ratio” have been revised
to “with **an** increase in aspect **ratios**”.

16. In Page 5, Lines 104, the words “etching direction” have been revised to “**the**
**etching direction**”.

17. In Page 5, Lines 105, the words “both side” have been revised to “both **sides**”.

18. In Page 5, Lines 123, the words “prior rolling direction” have been revised to “**the**
**prior rolling direction**”.

19. In Page 6, Lines 131, the words “etching system” have been revised to “**the**
**etching system**”.

20. In Page 6, Lines 132, the words “FEM model” have been revised to “**the FEM**
**model**”.

21. In Page 7, Lines 153, the words “of nanomembrane” have been revised to “of **the**
**nanomembranes**”.

22. In Page 7, Lines 154, the words “Si/Cr bilayer” have been revised to “**the bilayer**
**nanomembranes**”.

23. In Page 7, Lines 156, the word “quasistatic” has been revised to “**Quasistatic**”.

24. In Page 7, Lines 158, the word “key” has been revised to “**crucial**”.

25. In Page 7, Lines 175, the words “one end” have been revised to “**one-end**”.

26. In Page 8, Lines 178, the word “is” have been revised to “**are**”.

27. In Page 8, Lines 194-195, the sentence “As can be seen in Fig. 3c, the probability
distribution of five types of morphology and pattern sizes is discussed from the
statistical view.” has been revised to “**As shown in Fig. 3c, the probability distribution**
**of five types of morphology and pattern sizes is discussed from a statistical point of**
**view.**”.

28. In Page 8, Lines 200-201, the sentence “The forming morphology of
nanomembranes also follows the order of ring-arch-helix-taper-tube with increasing
aspect ratio, and they are corresponding to the structure distribution in Fig. 1d and Fig.
2d.” has been revised to “The forming morphology of nanomembranes also follows
the order of ring-arch-helix-taper-tube with increasing aspect ratio, **corresponding to**
**the structure distribution** in Fig. 1d and Fig. 2d.”.

29. In Page 9, Lines 228, the words “, however” have been revised to “**. However**”.

30. In Page 10, Lines 232, the words “combine effect” have been revised to “combine
**the effect**”.

31. In Page 10, Lines 233, the words “with length” have been revised to “with **a**
length”.

32. In Page 10, Lines 235, the words “release model” have been revised to “release
**models**”.

33. In Page 10, Lines 236-238, the words “region” have been revised to “**Region**”.

34. In Page 10, Lines 240-241, the words “results of simulation” have been revised to
“**the results of simulations**”.

35. In Page 10, Lines 244, the words “bilateral-released model” have been revised to
“**the bilateral-released model**”.

36. In Page 10, Lines 246, the words “opposite edge” have been revised to “**the**
opposite edge”.

37. In Page 10, Lines 247, the words “classic model” have been revised to “**the** classic
model”.

38. In Page 10, Lines 248, the words “after width” have been revised to “after **the**
width”.

39. In Page 10, Lines 251, the words “into ring” have been revised to “into **a** ring”.

40. In Page 10, Lines 258, the words “3D structures” have been revised to “**the** 3D

structures”.

41. In Page 11, Lines 261, the words “is corresponding” have been revised to
“corresponds”.

42. In Page 11, Lines 264, the words “to roll” have been revised to “of rolling”.

43. In Page 11, Lines 280, the words “of elastic energy” have been revised to “of the
elastic energy”.

44. In Page 12, Lines 295, the words “is contact with etchant” have been revised to “is
in contact with the etchant”.

45. In Page 12, Lines 299, the words “boundary” have been revised to “the boundary”.

46. In Page 12, Lines 300, the word “forming” has been revised to “formation”.

47. In Page 12, Lines 304-307, the sentences “Compared to models in normal etching
process, the movement of opposite edge in top etch is larger at the beginning, and
more area is released correspondingly in mechanical FEM modeling, leading to a
larger α and $\frac{v_W}{v_L}$ in local area (Supplementary Fig. 19).” have been revised to
“Compared to models in the normal etching process, the movement of the opposite
edge in the top etch is larger at the beginning. More area will be released
correspondingly in mechanical FEM modeling, leading to a larger α and $\frac{v_W}{v_L}$ in the
local area (Supplementary Fig. 44).”.

48. In Page 13, Lines 335, the words “while Cr layer” have been revised to “, while
the Cr layer”.

49. In Page 14, Lines 375, the words “contains” have been revised to “containing”.

50. In Page 14, Lines 376, the words “the training of” have been revised to “training”.

51. In Page 14, Lines 378, the words “a accuracy” have been revised to “an accuracy”.

52. In Page 14, Lines 383 the words “corresponding angle” have been revised to “the
corresponding angle”.

53. In Page 15, Lines 392, the words “quasistatic” have been revised to “Quasistatic”.

54. In Page 15, Lines 397, the words “Si/Cr bilayer nanomembrane” have been
revised to “Si/Cr bilayer rectangular nanomembrane”.

55. In Page 15, Lines 399, the words “etching process” have been revised to “the
etching process”.

56. In Page 15, Lines 399, the word “nanomembrane” has been revised to
“nanomembranes”.

57. In Page 15, Lines 403, the word “assistant” has been revised to “assistance”.

58. In Page 16, Lines 410, the words “sacrificial layer” have been revised to “**the**
sacrificial layer”.

59. In Page 16, Lines 415, the words “simulation” have been revised to “**the**
simulation”.

60. In Page 16, Lines 423, the word “released” has been revised to “**removed**”.

61. In Page 16, Lines 424, the word “Si and Cr layer” has been revised to “**the Si and**
**Cr layers**”.

62. In Page 17, Lines 435, the words “direct laser writing” have been revised to “**the**
direct laser writing”.

63. In Page 17, Lines 441, the words “Ge sacrificial layer” have been revised to “**the**
Ge sacrificial layer”.

64. In Page 17, Lines 443, the words “Si/Cr bilayer” have been revised to “**the Si/Cr**
bilayer”.

65. In Page 17, Lines 446, the words “released Si/Cr bilayer” have been revised to
“**the released Si/Cr bilayer**”.

66. In Page 17, Lines 446, the words “released Si/Cr bilayer” have been revised to
“**the released Si/Cr bilayer**”.

67. In Page 18, Lines 460-462, the sentence “A multichannel data set contains 275
photocurrent each channel is collected from traversing measurement of various
structures and introduced into a 7-layer hidden layer.” has been revised to A
multichannel data set contains 275 photocurrent **data in** each channel, **which** is
collected from traversing **measurements** of various structures and introduced into a 7-
layer hidden layer.”.

68. In Page 24, Lines 642, the sentences “Elastic energy difference of nanomembranes
releasing from adjacent edge and opposite edge for different aspect ratio.” have been
revised to “Elastic energy difference of nanomembranes releasing from adjacent **and**
**opposite edge for different aspect ratios.**”.

**Comment:**

*2: It is well-known that multiple configurations obtained from the same precursor*
*depend on the initial geometry and boundary conditions of 2D nanomembranes, but*
*only rectangular shapes with different Width-Length ratios are considered. How*
*about other shapes? Circular? The tilted strip?*

**Our response:**

Thank you again for your suggestion. The rectangles we used in the manuscript as a
demonstration are due to their moderate symmetry and variable control complexity, so
the relationship between the elastic energy of the system and the assembled structure
can be obtained more directly in the multilevel design method. For circular or
semicircular patterns, the parameters are mainly controlled by the radius. The
relationship between the self-assembly behavior lacks the aspect ratio as a parameter
to diversify the structure type. Therefore, it is difficult to obtain enough information
to analyze the forming mechanism. For the tilted parallelogram strip, we will
introduce length L , width W , and tilt angle α_t to analyze the assembly process. The tilt
angle as an additional parameter will increase the data magnitude by one dimension,
which may be helpful for detailed analysis of the relationship between the self-
assembly and the result, but it will increase the cost of comprehensive simulation and
analysis. Here, we exhibit some multilevel design examples and experimental results
with patterns of semicircle, triangle, and parallelogram strips to verify the universality
of the multilevel model.

Since the circular pattern has a short connection area with the fixed end, it is easy to
tear and fall off from the Si fixed end during the etching process. Therefore, we used a
semicircular pattern with a longer connection area for research.

**Re-Fig. 16.** Schematic diagram of the etching process of different shapes. The red
dotted line in the figure is the moving boundary of the sacrificial layer, and the black
arrow is the etching direction. Geometric parameters of (A) semicircle, (B) isosceles
right triangle, (C) parallelogram strip. Schematic diagram of the sacrificial layer
boundary changes of (D) semicircle, (E) isosceles right triangle, and (F) parallelogram
strip that are completely immersed in the etchant. Schematic diagram of the boundary
changes of the sacrificial layer during top etching of (G) semicircle, (H) isosceles
right triangle, and (I) parallelogram strip. Schematic diagram of the boundary changes
of the sacrificial layer during bottom etching of (J) semicircle, (K) isosceles right
triangle, and (L) parallelogram strip.

As shown in Re-Fig. 16, we have set models for semicircles with different radii R ,
isosceles right triangles with different diagonal lengths D , parallelogram strips with
the same included angle α_t and different aspect ratios $\frac{W}{L}$, and parallelogram strips with
the same length-to-width ratio at different included angles. FEM analysis and
experiments were conducted, confirming the broad applicability of the model (an
example of the sacrificial layer etching animation of the above pattern is attached to

the Movie S1 and S2). For the top etching (Re-Fig. 16G-I) and bottom etching (Re-
Fig. 16J-L) processes, we used the same method as modeling rectangular patterns,
that is, adding a preliminary release area for the top etching and stepwise etching from
both sides to the top for the bottom etching.

**Re-Fig. 17.** SEM images and FEM results of the immersive-etched self-assembled
semicircle of the multilevel design model. (A) Panorama SEM images of immersive-
etched self-assembled semicircles. SEM images of immersive-etched self-assembled
semicircle with sizes of (B) $D = 20 \mu\text{m}$, (C) $D = 28 \mu\text{m}$, (D) $D = 54 \mu\text{m}$, and (E) $D =$
$70 \mu\text{m}$. SEM images of immersive-etched self-assembled semicircle with sizes of (F)
$D = 20 \mu\text{m}$, (G) $D = 28 \mu\text{m}$, (H) $D = 54 \mu\text{m}$, and (I) $D = 70 \mu\text{m}$.

**Re-Fig. 18.** SEM images and FEM results of the bottom-etched self-assembled
semicircle of the multilevel design model. (A) Panorama SEM images of the bottom-

etched self-assembled semicircle. SEM images of bottom-etched self-assembled
 semicircles with sizes of (B) $D = 20\ \mu\text{m}$, (C) $D = 28\ \mu\text{m}$, (D) $D = 54\ \mu\text{m}$, and (E) $D =$
 $70\ \mu\text{m}$. TEM results of bottom-etched self-assembled semicircles with sizes of (F) $D =$
 $20\ \mu\text{m}$, (G) $D = 28\ \mu\text{m}$, (H) $D = 54\ \mu\text{m}$, and (I) $D = 70\ \mu\text{m}$.

**Re-Fig. 19.** SEM images and FEM results of the top-etched self-assembled semicircle
 of the multilevel design model. (A) Panorama SEM images of the top-etched self-
 assembled semicircle. SEM images of top-etched self-assembled semicircles with
 sizes of (B) $D = 20\ \mu\text{m}$, (C) $D = 28\ \mu\text{m}$, (D) $D = 54\ \mu\text{m}$, and (E) $D = 70\ \mu\text{m}$. TEM
 results of top-etched self-assembled semicircles with sizes of (F) $D = 20\ \mu\text{m}$, (G) $D =$
 $28\ \mu\text{m}$, (H) $D = 54\ \mu\text{m}$, and (I) $D = 70\ \mu\text{m}$.

**Re-Fig. 20.** SEM images and FEM results of the immersive-etched self-assembled
triangle. (A-B) Panorama SEM images of immersive-etched self-assembled triangles.
SEM images of immersive-etched self-assembled triangles with sizes of (C) $D = 24$
758 μm , (D) $D = 52 \mu\text{m}$, (E) $D = 100 \mu\text{m}$, and (F) $D = 110 \mu\text{m}$. FEM simulation results of
759 immersive-etched self-assembled triangles with sizes of (G) $D = 24 \mu\text{m}$, (H) $D = 52$
760 μm , (I) $D = 100 \mu\text{m}$, and (J) $D = 110 \mu\text{m}$.

Re-Fig. 21. SEM images and FEM results of the bottom-etched self-assembled triangle. (A-B) Panorama SEM images of bottom-etched self-assembled triangles.

SEM images of bottom-etched self-assembled triangles with sizes of (C) $D = 24 \mu\text{m}$, (D) $D = 52 \mu\text{m}$, (E) $D = 100 \mu\text{m}$, and (F) $D = 110 \mu\text{m}$. FEM simulation results of the bottom-etched triangle with sizes of (G) $D = 24 \mu\text{m}$, (H) $D = 52 \mu\text{m}$, (I) $D = 100 \mu\text{m}$, and (J) $D = 110 \mu\text{m}$.

Re-Fig. 22. SEM images and FEM results of the top-etched self-assembled triangle.

(A) Panorama SEM images of the top-etched self-assembled triangle. SEM images of
top-etched self-assembled triangles with sizes of (B) $D = 24 \mu\text{m}$, (C) $D = 52 \mu\text{m}$, (D)
$D = 100 \mu\text{m}$, and (E) $D = 110 \mu\text{m}$. FEM simulation results of the top-etched triangle
with sizes of (F) $D = 24 \mu\text{m}$, (G) $D = 52 \mu\text{m}$, (H) $D = 100 \mu\text{m}$, and (I) $D = 110 \mu\text{m}$.

**Re-Fig. 23.** SEM images and FEM results of immersive-etched self-assembled
triangles of different heights. (A) Panorama SEM images of immersive-etched self-
assembled triangles of different heights. (B) SEM images and (C) FEM simulation
results of immersive-etched self-assembled triangles with $D = 20 \mu\text{m}$ and triangle
bottom height $H = 80 \mu\text{m}$.

**Re-Fig. 24.** SEM images and FEM results of bottom-etched self-assembled triangles

with different heights. (A) Panorama SEM images of self-assembled triangles with
different heights etched on the bottom. (B) SEM images and (C) FEM simulation
results of bottom-etched self-assembled triangles with $D = 20 \mu\text{m}$ and $H = 80 \mu\text{m}$.

**Re-Fig. 25.** SEM images of self-assembled triangles with different heights etched on
the top and FEM results of the multilevel design model. (A) Panorama SEM images
of self-assembled triangles with different heights etched on the top. (B) SEM images
and (C) FEM simulation results of top-etched self-assembled triangles with $D = 20$
790 μm and $H = 80 \mu\text{m}$.

For semicircles (Re-Fig. 17-19) and isosceles right triangles (Re-Fig. 20-22), we only
obtained tubular structures under immersive-etched conditions in both FEM
simulations and experiments, because the nanomembranes tended to roll in one
direction. It is related to the fact that the etching path always tends to form local long-
end rolls at the free end. It is difficult to control multiple etching directions during the
etching process, and thus, it is hard for semicircles and triangles to form multi-stable
microstructures. For acute-angled isosceles triangles, various types of etching also
show similar results (Re-Fig. 23-25). Among them, the helix morphology formed
under immersive etching exhibits more asymmetry due to its asymmetry in all
directions etching. An unstable local state is more likely to exist in the isotropic
etching path, causing asymmetric rolling in the initial etching process.

**Re-Fig. 29.** SEM images and FEM results of top-etched parallelogram strips with
different widths. (A) Panorama SEM images of top-etched self-assembled
parallelogram strip nanomembranes with different widths. SEM images of self-
assembled parallelogram strip top-etched nanomembrane with sizes of (C) $4 \times 40 \mu\text{m}^2$,
$\alpha_t = 45^\circ$, (D) $16 \times 40 \mu\text{m}^2$, $\alpha_t = 45^\circ$, (E) $30 \times 40 \mu\text{m}^2$, $\alpha_t = 45^\circ$, and (F) $80 \times 40 \mu\text{m}^2$, $\alpha_t =$
45° , $\alpha_t = 45^\circ$. FEM simulation results of self-assembled top-etched parallelogram strip
nanomembrane with sizes of (G) $4 \times 40 \mu\text{m}^2$, $\alpha_t = 45^\circ$, (H) $16 \times 40 \mu\text{m}^2$, $\alpha_t = 45^\circ$, (I)
$30 \times 40 \mu\text{m}^2$, $\alpha_t = 45^\circ$, and (J) $80 \times 40 \mu\text{m}^2$, $\alpha_t = 45^\circ$.

We believe that polymorphic assembly can also be realized in other forms, but the
premise is that there are at least two low-strain elastic energy states in the assembly
structures to form a polymorphic state (Science 2011, 333, 1726.). Meanwhile, there
should be a large enough energy barrier for switching between to ensure that the states
are stable (Biomed. Mater. Eng. 2014, 24, 557; J. Appl. Mech. Trans. ASME 2011, 78,
0110021.). Taking a parallelogram pattern as an example, we assume the ideal etching
direction is in different etching directions β . When passing through a fixed point in
the strip, if an assembling structure wants to switch from the longer diagonal side
rolling to the shorter diagonal side rolling, a higher strain energy state will exist when
the rolling direction is changed, hindering the transition between the two stable states
(Re-Fig. 26). Therefore, only patterns with multi-stable state and a sufficiently large
strain energy barrier (such as parallelogram strips and rectangles with moderate tilt
angles) can realize multiple structural designs under the same precursor. For
semicircles and isosceles right triangles, the length of the free end is short within the
characteristic size of 2 to $80 \mu\text{m}$, making it difficult to form a potential barrier that
effectively separates the two forms, so there is no multi-stable structure.

A parallelogram with the same angle but a changed length-to-width ratio will roll
along different diagonals due to the difference in spatial order during oblique etching
(Re-Fig. 27-29). When etching occurs from the bottom, the wide edge at the top will
be fixed during the initial etching. When finally released, it will tend to exhibit a
unidirectional roll perpendicular to the shorter diagonal, thus to form various types of
tubes and rings (Re-Fig. 28). When etching occurs from the top, the rapid expansion
of the initial etching boundary in the obtuse angle area at the top will cause self-

assembly to occur along the shorter diagonal direction, thus forming a large number
of helix structures. The switching between the two forms will produce high elastic
energy states in other rolling directions as potential barriers, thereby achieving two
types of stable structures with apparent differences. We noticed that some Cr
nanomembrane are affected by photolithography accuracy and local penetration of
chromium etchant during micro-nano processing. Therefore, the ends of the
parallelogram strips will be inconsistent with the design model. At the same time,
some simulations of parallelogram strips have problems that the nanomembrane
moves behind the substrate in the experiments (Re-Fig. 28I). Although their
morphology is consistent with the experiment, this means that for more accurate
model predictions, the contact between the self-assembled nanomembrane and the
substrate needs to be considered. When applying an improved model in the future, the
multilevel FEM method can be further refined to improve the feasibility of the
application.

**Re-Fig. 30.** SEM images and FEM results of immersive-etched self-assembled
parallelogram strips with different tilt angles. (A) Panorama SEM images of
immersion etching of self-assembled parallelogram strips at different tilt angles. SEM
images of immersive-etched self-assembled parallelogram strips with angles of (B-I)

$\alpha_t = 10^\circ-80^\circ$. FEM results of immersive-etched self-assembled parallelogram strips
 with angles of (J-Q) $\alpha_t = 10^\circ-80^\circ$.

 **Re-Fig. 31.** SEM images and FEM results of the bottom-etched self-assembled
 parallelogram strips with different tilt angles. (A) Panorama SEM images of
 immersion etching of self-assembled parallelogram strips at different tilt angles. SEM
 images of bottom-etched self-assembled parallelogram strips with angles of (B-I) $\alpha_t =$
 $10^\circ-80^\circ$. FEM results of bottom-etched self-assembled parallelogram strips with
 angles of (J-Q) $\alpha_t = 10^\circ-80^\circ$.

by the two etching methods is not high enough to form different types of structures. It
gradually approaches the helix structures in the released rectangular shapes. However,
it is worth noting that the difference in etching methods still affects the rolling
direction in the initial phase, causing the pattern to form a helix structure with
opposite chirality.

**Our modification:**

- 1. Re-Fig. 16 has been added as Supplementary Fig. 45.
- 2. Re-Fig. 17 has been added as Supplementary Fig. 13.
- 3. Re-Fig. 18-19 has been added as Supplementary Fig. 46-47.
- 4. Re-Fig. 20 has been added as Supplementary Fig. 12.
- 5. Re-Fig. 21-26 has been added as Supplementary Fig. 48-53.
- 6. Re-Fig. 27 has been added as Supplementary Fig. 14.
- 7. Re-Fig. 28-29 has been added as Supplementary Fig. 54-55.
- 8. Re-Fig. 30 has been added as Supplementary Fig. 15.
- 9. Re-Fig. 31-32 has been added as Supplementary Fig. 56-57.
- 10. The answer has been added as a part of "4. FEM modeling for different patterns"
and "17. Design and fabrication of multimorphic structures via controlled local
etching" in Supporting Information.
- 11. "In addition to the Si/Cr rectangular patterns, the multilevel FEM model has been
designed for various types of patterns such as semicircular, triangular, and
parallelogram strips (Fig. 2d, Supplementary Fig. 7-14, and Movie S1 and S2)" has
been added in manuscript.
- 12. Fig. 2d-h of manuscript has been revised and added as follow:

13. "With proper control of surface tension, reconfiguration of different structure can
 be achieved (Movie S7). The local etching method is also applicable to the
 polymorphic design of other patterns (Supplementary Fig. 45-57). We have
 successfully utilized inclined etching to assemble the same parallelogram patterns
 along different diagonals, thus forming two types of structures with significant
 differences (Supplementary Fig. 53-57). It is worth noting that the premise for the
 preparation of polymorphic microstructures is that there are multi-stable strain energy
 states when the pattern is assembled, and there is an enough elastic energy barrier
 between the stable states. Otherwise, the pattern will only assemble into one type of
 structures (Supplementary Fig. 46-52)." has been added in manuscript.

14. Fig. 5f of manuscript has been added as follow:

Sample number	Si thickness (nm)	Cr thickness (nm)
60	10
60	20
60	60
15	40
30	40
90	40
Standard	60	40

We then conducted self-assembly experiments on samples from the above parameters
and studied the relationship between morphology-size distribution and tube radius-
thickness distribution.

**Re-Fig. 33.** SEM images and FEM results of Si (60 nm)/Cr (10 nm) double-layer
rectangular nanomembrane (sample #1). (A-B) Panorama SEM images of self-
assembled bilayer rectangular nanomembranes. SEM images of double-layer
rectangular nanomembranes with different sizes of (C) $4 \times 40 \mu\text{m}^2$, (D) $10 \times 40 \mu\text{m}^2$, (E)
$20 \times 40 \mu\text{m}^2$, (F) $40 \times 40 \mu\text{m}^2$, and (G) $60 \times 40 \mu\text{m}^2$. FEM simulation results of double-

layer rectangular nanomembranes with different sizes of (H) $4 \times 40 \mu\text{m}^2$, (I) $10 \times 40 \mu\text{m}^2$,
(J) $20 \times 40 \mu\text{m}^2$, (K) $40 \times 40 \mu\text{m}^2$, and (L) $60 \times 40 \mu\text{m}^2$.

**Re-Fig. 34.** SEM images and multilayer design model FEM results of Si (60 nm)/Cr
(20 nm) double-layer rectangular nanomembrane (sample #2). (A-B) Panorama SEM
images of self-assembled bilayer rectangular nanomembranes. SEM images of
double-layer rectangular nanomembranes with different sizes of (C) $4 \times 40 \mu\text{m}^2$, (D)
$10 \times 40 \mu\text{m}^2$, (E) $20 \times 40 \mu\text{m}^2$, (F) $40 \times 40 \mu\text{m}^2$, and (G) $60 \times 40 \mu\text{m}^2$. FEM simulation
results of double-layer rectangular nanomembranes with different sizes of (H) 4×40
983 μm^2 , (I) $10 \times 40 \mu\text{m}^2$, (J) $20 \times 40 \mu\text{m}^2$, (K) $40 \times 40 \mu\text{m}^2$, and (L) $60 \times 40 \mu\text{m}^2$.

Re-Fig. 35. SEM images and multilayer design model FEM results of Si (60 nm)/Cr (60 nm) double-layer rectangular nanomembrane (sample #3). (A-B) Panorama SEM images of self-assembled bilayer rectangular nanomembranes. SEM images of double-layer rectangular nanomembranes with different sizes of (C) $4 \times 40 \mu\text{m}^2$, (D) $10 \times 40 \mu\text{m}^2$, (E) $20 \times 40 \mu\text{m}^2$, (F) $40 \times 40 \mu\text{m}^2$, and (G) $60 \times 40 \mu\text{m}^2$. FEM simulation results of double-layer rectangular nanomembranes with different sizes of (H) $4 \times 40 \mu\text{m}^2$, (I) $10 \times 40 \mu\text{m}^2$, (J) $20 \times 40 \mu\text{m}^2$, (K) $40 \times 40 \mu\text{m}^2$, and (L) $60 \times 40 \mu\text{m}^2$.

 **Re-Fig. 36.** SEM images and multilayer design model FEM results of Si (15 nm)/Cr
 (40 nm) double-layer rectangular nanomembrane (sample #4). (A-B) Panorama SEM
 images of self-assembled bilayer rectangular nanomembranes. SEM images of
 double-layer rectangular nanomembranes with different sizes of (C) 4×40 µm², (D)
 10×40 µm², (E) 20×40 µm², (F) 40×40 µm², and (G) 60×40 µm². FEM simulation
 results of double-layer rectangular nanomembranes with different sizes of (H) 4×40
 µm², (I) 10×40 µm², (J) 20×40 µm², (K) 40×40 µm², and (L) 60×40 µm².

Re-Fig. 37. SEM images and multilayer design model FEM results of Si (30 nm)/Cr (40 nm) double-layer rectangular nanomembrane (sample #5). (A-B) Panorama SEM images of self-assembled bilayer rectangular nanomembranes. SEM images of double-layer rectangular nanomembranes with different sizes of (C) 4×40 μm², (D) 10×40 μm², (E) 20×40 μm², (F) 40×40 μm², and (G) 60×40 μm². FEM simulation results of double-layer rectangular nanomembranes with different sizes of (H) 4×40 μm², (I) 10×40 μm², (J) 20×40 μm², (K) 40×40 μm², (L) 60×40 μm².

Re-Fig. 38. SEM images and multilayer design model FEM results of Si (90 nm)/Cr

(40 nm) double-layer rectangular nanomembrane (sample #6). (A-B) Panorama SEM
 images of self-assembled bilayer rectangular nanomembranes. SEM images of
 double-layer rectangular nanomembranes with different sizes of (C) 4×40 μm^2 , (D)
 10×40 μm^2 , (E) 20×40 μm^2 , (F) 40×40 μm^2 , and (G) 60×40 μm^2 . FEM simulation
 results of double-layer rectangular nanomembranes with different sizes of (H) 4×40
 1019 μm^2 , (I) 10×40 μm^2 , (J) 20×40 μm^2 , (K) 40×40 μm^2 , (L) 60×40 μm^2 .

**Re-Fig. 39.** Relationship between structures and widths of the samples.

First, we observed the ring-arch-helix-taper-tube structural transition in various types
 of self-assembled Si/Cr nanomembranes by multilevel models and experiments (Re-
 Fig. 33-38). However, according to the structure -width relationship diagram (Re-Fig.
 39), it can be seen that there are obvious differences in the transition regions of each
 structure. For example, in sample #3, the taper-tube transition occurs at a narrower
 width $W = 36\sim 38 \mu\text{m}$, and in sample #4, there is almost no tube structure. As the
 thickness of the Si nanomembrane continues to increase, the Si/Cr nanomembrane
 will be prone to forming helix and arch structures (samples #4-#6). It indicates that
 the structural distribution is significantly related to the system strain during the self-
 assembly of nanomembranes. Therefore, we started with the radius of various tubular
 structures to study the strain state of the Si/Cr bilayer nanomembrane.

**Re-Fig. 40.** Diameter and strain distribution of Si/Cr self-rolled microtubes with
 different nanomembrane thicknesses. (A) FEM simulation, theoretical, and
 experimental results of the relationship between Si/Cr microtube radius and Cr
 nanomembrane thickness. (B) FEM simulation, theoretical, and experimental results
 of the relationship between Si/Cr microtube radius and Si nanomembrane thickness.
 (C) Dependence of the average strain of the Cr layer on the pre-strain of the Cr layer
 and the thickness of the Cr layer in Si/Cr microtubes. (D) Dependence of the average
 strain of the Si layer on the pre-strain of the Si layer and the thickness of the Si layer
 in Si/Cr microtubes. The error bars are the standard deviation of radii in experiments.

We used the above experiments and the force-moment balance (J. Appl. Phys. 2001,
 91, 9652; J. Appl. Phys. 2003, 94, 5333.) formula to estimate the strain magnitude in
 the Si/Cr layer, that is

$$R = \frac{(E_{\text{Si}})^2 t_{\text{Si}}^4 + (E_{\text{Cr}})^2 t_{\text{Cr}}^4 + 2E_{\text{Si}}E_{\text{Cr}}t_{\text{Si}}t_{\text{Cr}}(2t_{\text{Si}}^2 + 2t_{\text{Cr}}^2 + 3t_{\text{Si}}t_{\text{Cr}})}{6E_{\text{Si}}E_{\text{Cr}}t_{\text{Si}}t_{\text{Cr}}(t_{\text{Si}} + t_{\text{Cr}})\Delta\epsilon},$$

where E is Young's modulus, t is thickness of nanomembrane, and $\Delta\epsilon = \epsilon_{\text{Cr}} - \epsilon_{\text{Si}}$ is
 1050 the initial strain difference between Si and Cr layer. We substitute the set thickness

and tube radius distributions into this formula for fitting. The relationship between the
 tube radius and the thickness of the nanomembrane we obtained is shown in Re-Fig.
 40A, B, in which the prestrain of the Cr layer obtained by fitting is $\varepsilon_{Cr} \sim 0.65\%$, the
 pre-strain of the Si layer is approximately $\varepsilon_{Si} \sim -0.55\%$. The corresponding stress
 gradient is approximately $\Delta\sigma = \varepsilon_{Cr}E_{Cr} - \varepsilon_{Si}E_{Si} \approx 2600 \text{ MPa}$, which is consistent
 with the preset conditions in our simulation ($\Delta\sigma = 2000 \text{ MPa}$), providing an
 important guarantee for high-accuracy structural design. We can see that as the
 thicknesses of Si and Cr nanomembranes continue to increase, the Si/Cr tube radius
 shows a decreasing-increasing trend. As shown in Re-Fig. 40A, when $t_{Si} = 60 \text{ nm}$, as
 the Cr thickness continues to increase, the radii of the assembled tubes are 13.21 ± 0.55
 1061 μm (sample #1), $8.13 \pm 0.34 \mu\text{m}$ (sample #2), $7.48 \pm 0.28 \mu\text{m}$ (standard sample), and
 1062 $6.22 \pm 0.59 \mu\text{m}$ (sample #3). As shown in Re-Fig. 40B, when $t_{Cr} = 40 \text{ nm}$, as the Si
 thickness continues to increase, the radii of the assembled tubes are $6.95 \pm 0.93 \mu\text{m}$
 (sample #4), $3.67 \pm 0.21 \mu\text{m}$ (sample #5), $7.48 \pm 0.28 \mu\text{m}$ (standard sample), and
 $7.06 \pm 1.93 \mu\text{m}$ (sample #6). With the increase in thickness, maintaining the same
 strain layer requires a more significant strain difference between the two materials.
 Therefore, when the strain difference between the two materials is constant, a large
 layer thickness will always lead to an increase in the tube radius. We also obtained the
 strain distribution in the thickness direction by simulating the same prestrained Si/Cr
 system with different thicknesses.

In addition, the radius-thickness relationship of each sample obtained by the finite
 element design model also shows good consistency with experiments and formula
 calculations. For tubular models of various thicknesses, the multilayer design models
 all show larger tube radii than the theoretical models. This is because the elastic
 mechanics model used in the modeling process is a plane stress model, and for wider
 strip self-assembly, its mechanical behavior will be closer to the plane strain model,
 and the model can be modified as

$$1079 \quad R = \frac{(E'_{Si})^2 t_{Si}^4 + (E'_{Cr})^2 t_{Cr}^4 + 2E'_{Si}E'_{Cr}t_{Si}t_{Cr}(2t_{Si}^2 + 2t_{Cr}^2 + 3t_{Si}t_{Cr})}{6E'_{Si}E'_{Cr}t_{Si}t_{Cr}(t_{Si} + t_{Cr})\Delta\varepsilon'}$$

where $E'_i = E_i/(1 - \nu_i^2)$, $\Delta\varepsilon' = (1 + \nu_i)\Delta\varepsilon$. The correction of Young's modulus and
 strain gradient will improve the accuracy of the model in a wider range of situations.
 We studied the influence of the thickness and initial strain of the other layer on the

average strain in the nanomembrane layer when the parameters of the Si layer and the
 Cr layer are fixed, respectively. Among them, the average strain is calculated by the
 following formula:

$$\underline{\varepsilon} = \frac{\int_{t_{i-1}}^{t_i} \varepsilon(z) dz}{\sum_{i=1}^2 t_i}$$

It is used to characterize the strain state and strain size of the entire nanomembrane.
 First, we chose $t_{Si} = 60 \text{ nm}$, $\varepsilon_{Si} = -0.55\%$ to quantified the effect of t_{Cr} and ε_{Cr}
 (Re-Fig. 40C). It can be seen that when the top layer is Cr, its overall strain state
 mainly manifests as tensile strain, which is consistent with the mechanical behavior
 observed in the experiment. When the Si/Cr nanomembrane is released from the
 substrate, the Cr nanomembrane in the stretched state will be released and will be
 subject to relative compressive strain. In contrast, the Si nanomembrane will be
 subject to more tensile strain. As the thickness of the Cr nanomembrane continues to
 increase, its own strain will dominate the overall strain state of the nanomembrane.
 For each sample in our experiment, as the Cr thickness increases from 10 to 60 nm,
 the average strain shows a small attenuation from 0.9% to 0.5% and maintains the
 tensile strain state. It is worth noting that the increase in average strain relative to
 prestrain mainly originates from the neutral axis position during rolling $y_b =$

$\frac{\sum_{i=1}^n E_i t_i (y_i + y_{i-1})}{2 \sum_{i=1}^n E_i t_i}$, $y_i = \sum_{k=1}^i t_k$ The change in the overall strain will also lead to a

gradual increase in the tube radius as the nanomembrane thickness further increases.

Subsequently, we chose $t_{Cr} = 40 \text{ nm}$, $\varepsilon_{Cr} = 0.65\%$ to quantify the effect of t_{Si} and
 ε_{Si} (Re-Fig. 40D). The average strain relationship will show multiple rising and
 falling trends due to the interaction between the tube radius and the neutral axis, the
 neutral axis, and the thickness of each layer. When $t_{Si} = 40 \sim 100 \text{ nm}$, the abnormal
 prestrain-average strain region that appears mainly comes from the fact that when the
 neutral axis is located near the Si layer, the bending moments in the layer cancel each
 other out. When the thickness of the Si layer further increases, the strain within the Si
 layer is still dominated by its pre-strained state. In the experiment, as the Si
 nanomembrane thickness increased from 15 to 90 nm, the average strain of each
 sample remained at a compressive strain state of -0.5%~ -0.6%, which is highly
 consistent with the experimental verification and model design of Si/Cr double-layer
 nanomembranes.

In summary, we systematically discussed the stress-strain relationship in Si/Cr

nanomembranes and showed good consistency between experiments, theories, and
simulations. We proved that this method can also be applied to detailed design of the
same material system, further expanding its application. In future designs, we can also
introduce strain layer changes, such as local ion implantation (Phys. Rev. Mater. 2021,
5, 124603.) to further refine the structure design.

**Our modification:**

- 1. Re-Table 3 has been added as Supplementary Table 1.
- 2. Re-Fig. 33-39 have been added as Supplementary Fig. 24-30.
- 3. Re-Fig. 40A, B have been added as Fig. 3 d, e.
- 4. Re-Fig. 40C, D have been added as Supplementary Fig. 31.
- 5. The answer has been added as "FEM modeling and strain analysis for different
Si/Cr thickness" in Supporting Information.
- 6. Moreover, we designed Si/Cr systems with different parameters correspond to Si
nanomembranes (15/30/90 nm) and Cr nanomembranes (10/20/60 nm) of different
thicknesses to analyze the initial strain and average strain of Si and Cr
nanomembranes after release (Supplementary Table 1 and Supplementary Fig. 24-30).
As shown in Fig. 3d, e, we collected the radii of tubular structures with the same Si
thickness ($t_{Cr} = 40$ nm) and different Cr thicknesses, and the same Cr thickness
($t_{Si} = 60$ nm) and different Si thicknesses. With the increase of the thickness of Cr
layer, the radii of assembled tubes with $t_{Si} = 60$ nm are 13.21 ± 0.55 μm ($t_{Cr} =$
10 nm), 8.13 ± 0.34 μm ($t_{Cr} = 20$ nm), 7.48 ± 0.28 μm ($t_{Cr} = 40$ nm), 6.22 ± 0.59
1136 μm ($t_{Cr} = 60$ nm), respectively. Meanwhile, with the increase of the thickness of Si
layer, the radii of assembled tubes with $t_{Cr} = 60$ nm are 6.95 ± 0.93 μm ($t_{Si} = 15$ nm),
3.67 ± 0.21 μm ($t_{Si} = 30$ nm), 7.48 ± 0.28 μm ($t_{Si} = 60$ nm), and 7.06 ± 1.93 μm ($t_{Si} =$
90 nm), respectively. We used above experiments and the force-moment balance
formula to estimate the strain magnitude in the Si/Cr layer, that is

$$R = \frac{(E_{Si})^2 t_{Si}^4 + (E_{Cr})^2 t_{Cr}^4 + 2E_{Si} E_{Cr} t_{Si} t_{Cr} (2t_{Si}^2 + 2t_{Cr}^2 + 3t_{Si} t_{Cr})}{6E_{Si} E_{Cr} t_{Si} t_{Cr} (t_{Si} + t_{Cr}) \Delta \varepsilon},$$

where E is Young's modules, t is thickness of nanomembrane, and $\Delta \varepsilon = \varepsilon_{Cr} - \varepsilon_{Si}$ is
the initial strain difference between Si and Cr layer. We substitute the set thickness
and tube radius into this formula for fitting. The relationship between the tube radius

and the thickness of the nanomembrane we obtained is shown in the Supplementary
 Fig. 31A, B, in which the prestrain of the Cr layer ε_{Cr} obtained by fitting is $\sim 0.65\%$,
 the pre-strain of the Si layer ε_{Si} is approximately $\sim -0.55\%$, the corresponding
 stress gradient is approximately $\Delta\sigma = \varepsilon_{Cr}E_{Cr} - \varepsilon_{Si}E_{Si} \approx 2600$ MPa, which is
 consistent with the preset conditions in our simulation ($\Delta\sigma = 2000$ MPa),
 providing an important guarantee for high-accuracy structural design. In addition, we
 used the thickness and pre-strain of nanomembranes as parameters to calculate the
 strain relationship in the Si/Cr nanomembrane. The average strain of Si and Cr
 nanomembranes are also in good agreement with the experiment, providing an
 effective method for strain characterization of multilayer nanomembrane systems
 (Supplementary Fig. 31)." has been added in manuscript.

**Comment:**

*4: In the simulations of FEM, what is the setting details of interactions and contact*
 *conditions used in FEM? To what extent does this affect the simulation results?*

**Our response:**

Thank you for your insightful question. In this work, our elastic mechanical model
 uses surface contact constraints for the combination of Si/Cr bilayer to realize the
 connection between the nanomembranes. For the release of each time step, we use
 hinged ($U_1 = U_2 = U_3 = 0$, no displacement) boundary conditions for the unreleased
 part and a completely fixed boundary condition ($U_1 = U_2 = U_3 = 0, UR_1 = UR_2 = UR_3 =$
 0 , no displacement, no rotation) is used at the permanently fixed end to ensure that it
 meets the experimental conditions. It is worth noting that we do not define contact,
 self-contact, and contact control for the nanomembranes of the elastic mechanics
 module at the initial development of the model. The inapplicability of contact and
 self-contact in this model stems from the nonlinear mechanics in large deformation,
 many time steps, and the complex boundary conditions of point cloud construction
 between each time step. But in some cases, contact conditions for the elastic model
 are essential for accurate structure design and prediction.

**Re-Fig. 41.** Schematic diagram of contact module in the multilevel design model. (A),
 (C) FEM simulation results of the multilevel model in the case of single-turn rolling
 without the contact module. (B), (D) FEM simulation results of the multilevel model
 in the case of single-turn rolling when applying the contact module. (E) FEM
 simulation results of the multilevel model in the case of multi-turn rolling without
 applying the contact module. (F) FEM simulation results of the multilevel model in
 the case of multi-turn rolling when applying the contact module. (G) SEM image of
 multi-turn rolling Si/Cr nanomembrane.

Here, we demonstrate some examples of the contact module in the multilevel FEM

model adopted in this study. In the unidirectional rolling process of the traditional
FEM model, when the contact module is not applied to the model, the self-assembled
nanomembrane will partially overlap (Re-Fig. 41A, C). This phenomenon has less
impact on self-assembly models with a single turn or less than a single turn because
the self-contact area of such models is smaller. However, the lack of a contact module
will cause extremely inaccurate boundary condition results for the analysis of the
multi-turn rolling model and further affect the subsequent stress-strain analysis that
may be required. The large-scale Si/Cr nanomembrane self-assembly model contains
a large number of model self-contacts during the deformation process. As shown in
the Re-Fig. 41E, when the contact module is not introduced into the model, it will
cause serious distortion of the model, with multiple tubes of self-assembly in different
directions, and it is difficult to stabilize. Here, we apply the general contact module to
the same model and define the tangential contact mechanism as a penalty function
whose form is

$$1200 \quad \tau = \mu\sigma,$$

where τ is the tangential friction force, σ is the normal stress, and the friction
coefficient is defined as the dynamic friction coefficient of the Cr-Cr layer $\mu = 0.34$.
The normal contact mechanism is hard contact. As a result, it can be seen that there is
no volume overlap of the nanomembranes in the calculation results (Re-Fig. 41B, D).
At the same time, its multi-turn self-assembled structure will quickly stabilize due to
the existence of the contact volume, reflecting more realistic experimental results.
After applying the contact module, we found that the self-contact problem of the
model is significantly improved (Re-Fig. 41F), but due to the need to calculate the
contact between a large number of small-sized meshes during the quasistatic release
of the component, a large amount of computing power is consumed, which will
significantly extend the simulation time. Based on the above situation, when the self-
assembly process is free from self-contact or only has single-axis curvature single-
turn rolling, we mainly use non-contact module operations. When the self-assembly
process contains complex deformation or multi-turn rolling, we will consider the
contact module to ensure the accuracy of the results.
In the future, we can further analyze the reasons for the failure during the preparation
of conventional micro-nano structures (such as blocking of displacement and local
wrinkles) by utilizing contact modules.

**Our modification:**

- 1. Re-Fig. 41 has been added as Supplementary Fig. 67.
- 2. The answer has been added as "22. Contact module in FEM model" in Supporting
Information.
- 3. "For model with complex deformation and significant volume overlap, contact
module is applied to ensure the accuracy of FEM simulations (Supplementary Fig.
67)." has been added in method of manuscript.

**Comment:**

5: Is there a size-correlated effect on the self-rolling behaviors of the initial 2D
nanomembranes? Does the conclusion still hold when the size of initial 2D
nanomembranes being scaled up or down by a factor of 10?

**Our response:**

Thank you very much for your question. For 2D nanomembranes, the size effects
involved in shrinking and amplification are quite different. For the two-dimensional
patterns involved in this manuscript with a size of 2 to 80 μm , we will discuss the two
cases separately. When the overall size is reduced by 10 times, the minimum size will
be close to 200 nm, while the thickness of the nanomembrane system is about 100 nm,
which means that the self-rolling theory and the plane strain/plane stress theory in the
simulation will no longer apply. In this condition, the strain perpendicular to the
thickness direction will not be negligible (Landau, L. D., Lifshitz, E. M., Atkin, R.,
Fox, N. *The theory of elasticity. Physics of Continuous Media*, (CRC Press), pp 167-
178. (2020)), that is,

$$\varepsilon_z = \frac{1}{E}(\sigma_z - \mu(\sigma_x + \sigma_y)) \neq 0.$$

When the width is equal to the thickness ($W = 0.4 \mu\text{m}$, $t_{Si} + t_{Cr} = 0.1 \mu\text{m}$), this
will lead to a significant increase in the radius of curvature (Re-Fig. 42, 43). At the
same time, the length of the cantilever beam will be significantly shortened, and its
large deformation behavior will also be significantly weakened. It can be seen from
the FEM that the patterns with aspect ratio $\frac{W}{L}$ from 0.1 to 2 all exhibit uniaxial
bending, and their small deformation and a small distance between the fixed end make
it difficult to form multi-directional rolling during the release process.

**Re-Fig. 42.** FEM simulation results of double-layer rectangular nanomembranes with
 pattern sizes of (A) $0.4 \times 4 \mu\text{m}^2$, (B) $1 \times 4 \mu\text{m}^2$, (C) $2 \times 4 \mu\text{m}^2$, (D) $4 \times 4 \mu\text{m}^2$, and (E) 6×4
 1255 μm^2 .

**Re-Fig. 43.** Tube radius ratio between the size of $(0.4-6) \times 4 \mu\text{m}^2$ (R) and $60 \times 40 \mu\text{m}^2$
 (R_0).

At the same time, the surface tension and liquid flow in the nano-scale structure will
 be significantly enhanced, resulting in a significant deterioration in the forming
 stability and difficulty to control the rolling direction (ACS nano 2009, 3, 1663-1668.),
 which will also introduce additional considerations for the simulation. Moreover, the
 sharp reduction in sizes makes the structure preparation a new problem because the
 conventional photolithography process will make it difficult to process sub-micron-
 scale patterns, and other processes (such as EBL) will further increase the process
 time and cost.

**Re-Fig. 44.** SEM images and multilayer design model FEM results of a double-layer
 rectangular nanomembrane with a size factor of 0.5 times. (A-B) Panorama SEM
 images of self-assembled bilayer rectangular nanomembranes. SEM images double-
 layer rectangular nanomembranes with sizes of (C) $2 \times 20 \mu\text{m}^2$, (D) $5 \times 20 \mu\text{m}^2$, (E)
 $10 \times 20 \mu\text{m}^2$, (F) $20 \times 20 \mu\text{m}^2$, and (G) $30 \times 20 \mu\text{m}^2$. FEM simulation results of double-
 layer rectangular nanomembranes with sizes of (H) $2 \times 20 \mu\text{m}^2$, (I) $5 \times 20 \mu\text{m}^2$, (J)
 $10 \times 20 \mu\text{m}^2$, (K) $20 \times 20 \mu\text{m}^2$, and (L) $30 \times 20 \mu\text{m}^2$.

When the size is scaled to 0.5 times, their pattern sizes are considerably increased
 compared to 0.1 times, but the length of the released cantilever beam is still short,
 which cannot make the self-assembled Si/Cr nanomembrane roll in a stable state from
 multiple directions. Therefore, it will be challenging to fabricate a taper-type structure
 at this scale (Re-Fig. 44).

**Re-Fig. 45.** SEM images and multilayer design model FEM results of a double-layer
 rectangular nanomembrane with a size factor of 5 times. (A-B) Panorama SEM
 images of self-assembled bilayer rectangular nanomembranes. SEM images of
 double-layer rectangular nanomembranes with sizes of (C) $20 \times 200 \mu\text{m}^2$, (D) 50×200
 1287 μm^2 , (E) $100 \times 200 \mu\text{m}^2$, (F) $200 \times 200 \mu\text{m}^2$, and (G) $300 \times 200 \mu\text{m}^2$. FEM simulation
 results of double-layer rectangular nanomembranes with sizes of (H) $20 \times 200 \mu\text{m}^2$, (I)
 $50 \times 200 \mu\text{m}^2$, (J) $100 \times 200 \mu\text{m}^2$, (K) $200 \times 200 \mu\text{m}^2$, and (L) $300 \times 200 \mu\text{m}^2$.

**Re-Fig. 46.** SEM images and multilayer design model FEM results of double-layer
 rectangular nanomembranes with a size factor of 10 times. (A-F) SEM images of
 double-layer rectangular nanomembranes with a size factor of 10 times. FEM
 simulation results of double-layer rectangular nanomembranes with sizes of (G)
 $40 \times 40 \mu\text{m}^2$, (H) $100 \times 400 \mu\text{m}^2$, (I) $200 \times 400 \mu\text{m}^2$, (J) $400 \times 400 \mu\text{m}^2$, and (K) 600×400
 1296 μm^2 .

According to the simulation of the multilevel design model, when the pattern size is
 enlarged 5-10 times, the patterns will tend to roll unidirectionally in multiple
 directions from each edge, eventually forming a complex 3D structure composed of
 various tubular structures and local bending (Re-Fig. 45-46). For the FEM simulation
 of large-size patterns, we did not use the contact module. When considering the
 contact module, the mesh needs to be as small as possible to avoid the problem of the
 nanomembrane being unable to roll due to collisions of self-rolling meshes, which
 will significantly extend the calculation time to decades of days. Moreover, we only
 used the incomplete etching time points in the FEM simulation to show the
 corresponding insufficient etching structure in the SEM. Only when the width is small

(20~40 μm), the nanomembrane tends to form a multi-turn rolled structure. Since the
cantilever beams are too long, the multi-turn rolled structure will be difficult to
maintain a tight structure, thus forming a structure between helix and ring (Re-Fig.
46A). In the experiment, we observed multi-turn rolled structures and long-side
unidirectional rolled structures in the smaller width area, which also means that the
disturbance during the release process of the sacrificial layer at large sizes will have a
more significant impact on the self-assembly process. At the same time, for broader
nanomembrane patterns, the larger bending moment caused by the long edge rolling
during the rolling process will cause the Si/Cr nanomembrane and the edge of the Si
nanomembrane to tear, thus further affecting the yield.

In summary, the size effect will significantly affect the self-assembly process of pre-
strained nanomembranes. At smaller sizes, we need to consider the failure of plane
elasticity theory and the weakening of large deformation effects, limiting the diversity
of nanomembrane structure assembly. When the size is larger, we need to consider the
occurrence of local rolling in multiple directions and possible problems such as
tearing and breakage of nanomembrane during experiments, which usually result in
lower yields. Although the distribution of microstructures will be affected by size
effects, our multilevel model still performs good applicability for structural design
and prediction in the scale range of hundreds of nanometers to hundreds of
micrometers.

**Our modification:**

- 1. Re-Fig. 42-46 have been added as Supplementary Fig. 18-22.
- 2. The answer has been added as "6. FEM applicability for different sizes" in
Supporting Information.
- 3. "Meanwhile, the method demonstrates cross-scale compatibility for models from
the hundred nanometer scale to the hundred micrometer scale (Supplementary Fig.
18-22)." has been added in manuscript.

**Comment:**

6: *Is it possible to achieve the real-time detection of the change of angle α during*
*the etching process experimentally using high-speed imaging equipment? They can be*
*compared with the FEM results and further validate the model parameters.*

**Our response:**

Thank you very much for your question. We apologize that we haven't purchased
high-speed imaging equipment, but since the etching process takes 15-30 minutes, we
can use a conventional microscope to record and read the data. The etching process
will be supplemented as a video file (Movie S3). At the same time, we will also
exhibit the process of rolling angle α changing with time during the quasistatic release
process of various structures in the multilevel model.

**Re-Fig. 47.** Schematic diagram of the relationship between α -etching time and non-
ideal situations in FEM and experiments. (A) The relationship between the α -etching
time of various structures measured in the multilevel FEM model. (B) The
relationship between the α -etching time of various structures measured in the Si/Cr
release experiment. (C-E) Ideal etching boundary as envisioned in the FEM model. (f-
1355 h) The local anisotropic release process and optical images exist in the experiment.

As shown in the Re-Fig. 47A, we obtained the trend of α changing with etching time
for various structures in the FEM. They all exhibit a trend of gradually increasing
with the etching process. At the end of etching, the obtained α are $\sim 35^\circ$ (4 μm , ring),
$\sim 78^\circ$ (10 μm , arch), $\sim 64^\circ$ (20 μm , helix), $\sim 63^\circ$ (40 μm , taper), $\sim 53^\circ$ (60 μm , tube),
which is also close to the α angle corresponding to the lowest elastic energy we
obtained from the energy calculation, which are $30^\circ\sim 40^\circ$ (ring), $50^\circ\sim 80^\circ$ (arch),

65°~75° (helix), 50°~65° (taper), 40°~55°(tube). We noticed that at $W = 20 \mu\text{m}$, the
angle-time relationship fluctuated in the early stages of etching, which is caused by a
sudden change in the rolling direction when the helix structure switched from an
initial bidirectional rolling to a unidirectional rolling. Through the sacrificial layer
etching video (Movie S3), we also obtained the α -time curves of various structures
above. The α obtained at the end of the etching are $\sim 26^\circ$ (4 μm , ring), $\sim 71^\circ$ (10 μm ,
arch), $\sim 56^\circ$ (20 μm , helix), $\sim 64^\circ$ (40 μm , taper), and $\sim 40^\circ$ (60 μm , tube), which are
also in the corresponding lowest energy α region (Re-Fig. 47B). It is worth noting that
although the α angle at the end of etching is consistent with that predicted by the
multilevel model, the α -time relationship during the process is different from the
model in the initial stage of etching, which is mainly composed of two reasons. First
of all, in the experiments, the etched area is not always released uniformly with the
etching time. For example, the assembly of the 40 μm area into the taper shape will
mainly occur at the end of etching (Movie S3). At the same time, the multilevel model
stipulates the time required for each quasistatic step is the same (Re-Fig. 47C-E).
Secondly, in the experiment, we observed the phenomenon of α first increasing and
then decreasing during the initial etching of the $W = 60 \mu\text{m}$ pattern. This is due to the
preferential local rolling on both sides of the rectangle during the actual experiment
(Re-Fig. 47F-H). It will lead to a change in the dominant rolling direction during
initial etching (Nano Lett. 2011, 11, 2280; Nano Lett. 2010, 10, 3927). As etching
continued, experiments and multilevel designs finally gave similar results, but it also
shows that there is room for improvement in the prediction accuracy of the model. In
the future, we will also fully consider more boundary conditions that may exist in the
early stage of etching (such as slight pattern deviation and local asymmetric effects),
laying the foundation for the broader application of the model.

**Our modification:**

- 1. Re-Fig. 47A, B have been added as Fig. 4d, e.
- 2. Re-Fig. 47C-H have been added as Supplementary Fig. 41.
- 3. Video of etching in Re-Fig. 47B has been added as Movie S3.
- 4. The answer has been added as "13. Tracking of self-assembly process of Si/Cr
nanomembrane" in Supporting Information.
- 5. "As shown in Fig. 4d, the quasistatic FEM model simulates the change of α with

etching time during the assembly process, and generally restores the relationship
between α and etching time during the sacrificial layer etching process in the
experiment (Fig. 4e, Movie S3). When the etching finished, the distribution of α are
as follows: ring ($25^\circ - 35^\circ$), arch ($70^\circ - 80^\circ$), helix ($50^\circ - 65^\circ$), taper ($60^\circ - 75^\circ$), and
tube ($40^\circ - 55^\circ$). The α angle obtained at the end of etching between model
calculations and experimental results is highly consistent with the calculated α_{min}
corresponding to each structure in Fig. 4c, providing an observation scheme for
tracking intermediate states in the assembly process. We noticed that there is an
obvious difference of the α between the multilevel model and the experiments in the
early stage of etching, which is mainly due to the non-ideal release of sacrificial layer
and local non-uniform etching caused by insufficient accuracy in micro-nano
processing (Supplementary Fig. 41)." has been added in method of manuscript.

=====

**Reviewer #3**

=====

*This work presents a 3D assembly of 2D plane structures guided by comprehensive*
*FEM and well-validated by extensive experiments. The remarkable agreement*
*between FEM and experiments were obtained, regardless of dimensional sizes of films.*
*Besides, exploration of assembled 3D structures with Si/Cr as photodetectors has also*
*been explored to study the incidence angle of light with the help pf DNN,*
*demonstrating an immediate application in a device level. Details of both FEM and*
*experiments were well presented allowing the readers to follow and the results are*
*reliable to be reproduced. Given the critical importance of 3D structures in design*
*and nanomanufacturing of electronics and devices, particularly, structural materials*
*made of traditional semiconductor material such as Si that was demonstrated in this*
*work, the current multilevel FE approach-guided design and fabrication of 3D*
*structures offer a powerful and low-cost means. I recommend the acceptance of this*
*work after addressing the following minor issues.*

**Our response:**

Thank you very much for your review. We discussed the crystallization state and
interface characteristics of Si/Cr nanomembranes based on your suggestions,
improving the credibility of the work. At the same time, your comments provide us
with inspiration for reconfigurable microstructure electronic device assembly. Finally,
we discuss the self-assembly of nanomembranes with different patterns and
thicknesses in a multilevel FEM model, demonstrating the applicability of this method
in complex precursors and fine structures.

**Comment:**

*1: the authors mentioned "the pre-strain nanomembrane", I assume this pre-strain is*
*induced by material lattice mismatch between two deposited layers and is not*

*introduced by external stimuli or deposition process. Please clarify this concept.*

**Our response:**

Thank you very much for your question. The lattice mismatch between the two phases
is closely related to the lattice constants of the two materials/phases, and the lattice
mismatch usually requires that the mismatch between the lattice constants be less than
5% before the complete grain boundary congruence can occur. When the mismatch
degree is 5% -25%, a semi-coherent phase boundary is formed, and when the
mismatch degree is greater than 25%, a non-coherent interface is formed. According
to the lattice constants of Si and Cr, the lattice constant of Si is 5.43Å, and the lattice
constant of Cr is 2.884Å, and the mismatch degree of the two is as high as 88.3%,
which means that it is almost impossible to establish a coherent phase interface at the
Si/Cr interface, so there is no pre-strain introduced by the lattice mismatch.

In addition, according to theoretical calculations, if the self-assembly with the strain
caused by the lattice mismatch as the pre-strain, their tube radius will only be tens of
nanometers, which is seriously inconsistent with the experiment. Under the
assumption of plane strain, the strain gradient corresponding to lattice mismatch is
introduced into the Si/Cr nanomembrane system for calculation. When the strain
gradient considering lattice mismatch is located throughout the entire nanomembrane
layer, the radius of the tube obtained by rolling is: (J. Appl. Phys. 2001, 91, 9652; J.
Appl. Phys. 2003, 94, 5333.):

$$1461 \quad R = \frac{(E'_{Si})^2 t_{Si}^4 + (E'_{Cr})^2 t_{Cr}^4 + 2E'_{Si}E'_{Cr}t_{Si}t_{Cr}(2t_{Si}^2 + 2t_{Cr}^2 + 3t_{Si}t_{Cr})}{6E'_{Si}E'_{Cr}t_{Si}t_{Cr}(t_{Si} + t_{Cr})\Delta\varepsilon} = 227.4 \text{ nm},$$

in which E'_i is Young's modulus, t_i is the thickness of each layer of nanomembrane
$t_{Si} = 60 \text{ nm}$, $t_{Cr} = 40 \text{ nm}$, $\Delta\varepsilon = 88.3\%$ is the lattice mismatch caused by epitaxy of
the Si/Cr nanomembrane layer. When considering that the actual lattice mismatch is
mainly located near the interface, otherwise the dislocation will lead the strain
relaxation (Appl. Phys. Lett. 1988, 52, 380.). We assume that the epitaxy layer is
about ten atomic layers, and the radius of Si/Cr nanomembrane is only:

$$1468 \quad R = \frac{2\sum_{i=1}^4 E'_i t_i [z_i^2 + z_{i-1} z_i + z_{i-1}^2 - 3z_b(z_i + z_{i-1} - z_b)]}{3\sum_{i=1}^4 E'_i t_i (z_i + z_{i-1} - 2z_b)(\varepsilon_{const} - \varepsilon_i^0)} = 474.1 \text{ nm}.$$

Among them, $i=1,2,3,4$ is strain-free Si, lattice-mismatched Si layer, lattice-
mismatched Cr layer, and strain-free Cr nanomembrane layer, respectively. $t_1 + t_2 =$
60 nm , $t_3 + t_4 = 40 \text{ nm}$, $t_2 = 10 \times 5.43 \text{ \AA} = 5.43 \text{ nm}$, $t_4 = 10 \times 2.884 \text{ \AA} =$

2.884 nm, $\varepsilon_3^0 = 0.883$.

$$\varepsilon_{const} = -\frac{\sum_{i=1}^2 E_i' t_i \varepsilon_i^0}{\sum_{i=1}^2 E_i' t_i} = 0.174$$
$$z_b = \frac{\sum_{i=1}^2 E_i' t_i (z_i + z_{i-1})}{2\sum_{i=1}^2 E_i' t_i} = 95 \text{ nm}$$

The tube radius measured in the experiment is about 5~6 μm . Therefore, when
considering that the strain of the Si/Cr nanomembrane originates from epitaxy, the
theoretical results will also deviate seriously from the experiment, indicating that the
strain gradient between the Si/Cr nanomembrane interface does not rely on lattice
mismatch to form a strain gradient.

Moreover, we analyzed the samples by transmission electron microscope (TEM), and
it can be seen that there is no coherent interface at the Si/Cr interface, but an obvious
non-coherent phase boundary. The above evidence exhibits that the prestrain of Si and
Cr bilayer nanomembrane is mainly caused by the temperature difference during
deposition and the difference in thermal expansion coefficient of the bilayer
nanomembranes after cooling, rather than lattice mismatch (Nano Lett. 2018, 18,
3688.; Nanoscale, 2011, 3, 96.).

In order to confirm that the strain gradient between Si/Cr double-layer
nanomembranes does not originate from lattice mismatch, we performed focused ion
beam (FIB) cutting on the sample and used high-angle annular dark field scanning
transmission electron microscopy (HAADF-STEM) and TEM to examine the cross
section of the sample. Morphological characterization is performed. As can be seen
from Re-Fig. 48A, the sample is composed of Si/SiO₂/Ge nanomembrane/Si
nanomembrane/Cr nanomembrane from bottom to top. The carbon protective layer
and Pt protective layer are sprayed on the Cr nanomembrane to protect the cross
section of FIB cutting samples. To verify the elemental composition of each material,
energy dispersive X-ray spectroscopy is used to conduct elemental analysis of the
interface. The distribution diagram of each element is shown in Re-Fig. 48B-G, in
which each element shows high signal intensity in the corresponding single-element
layer, confirming the elemental composition of each layer in the cross section. From
the analysis of Re-Fig. 48B and C, it can be seen that there is a silicon oxide layer
with a thickness of approximately 35 nm above the silicon substrate, which is

obtained by thermal oxidation of the silicon wafer substrate. In addition, Re-Fig. 48C
 shows stronger intensity in the Cr nanomembrane layer. This is because the higher
 temperature during electron beam evaporation will oxidize the evaporated Cr and the
 remaining oxygen in the cavity. But in general, chromium nanomembranes have less
 oxygen adsorption content, which can be seen from optical images and composition
 analysis. Since silicon and germanium nanomembranes are also deposited by electron
 beam evaporation, their EDX images show a small amount of oxygen distribution. As
 shown in Re-Fig. 48G, the entire nanomembrane system shows a small amount of
 carbon element distribution. This is due to the slight accumulation of carbon-
 containing substances and carbon protective layers in the sample during processing
 and electron beam bombardment. The samples are plasma cleaned, but contamination
 by carbonaceous species is not completely avoided. This cross-sectional analysis
 shows the clear interface between the Ge/Si/Cr three-layer nanomembrane, which can
 be further analyzed by selected area electron diffraction (SAED) techniques to
 analyze the interface and crystallization state of each material.

**Re-Fig. 48.** Characterization of the cross-sectional structure and element distribution
 of the nanomembrane sample. (A) HAADF-STEM image of the cross-section of the
 sample. EDX images of the distribution of (B) Si, (C) O, (D) Pt, (E) Cr, (F) Ge, and
 (G) C elements in the cross section of the sample.

As shown in Re-Fig. 49A-D, we used high-resolution transmission electron
 microscopy (HRTEM) to characterize the morphology of each layer and interfaces. It
 can be seen that there are obvious interfaces between each layer. In addition, we
 performed SAED analysis on each layer. Re-Fig. 49A shows the interface diagram
 between the silicon substrate and silicon oxide. The images show the orderly

arrangement of atoms in the silicon substrate without grain boundary. At the same
time, Re-Fig. 49E shows the SAED image of the silicon substrate. It can be seen that
clear diffraction spots in the direction of the [011] zone axis, in which the silicon (200)
and (11 $\bar{1}$) crystal plane proves that the substrate is single crystal silicon. In Re-Fig.
49B and Re-Fig. 49C, HRTEM images of the germanium nanomembrane and silicon
nanomembrane do not show regular atomic arrangement. In the SAED images of Re-
Fig. 49F and Re-Fig. 49G, there are no sharp diffraction rings corresponding to
polycrystalline materials or discrete diffraction spots corresponding to single crystal
materials, but diffused diffraction rings, confirming that the germanium
nanomembrane and silicon nanomembrane deposited by electron beam evaporation
are amorphous nanomembranes. Previous studies have reported growing
polycrystalline silicon and polycrystalline germanium thin nanomembranes through
electron beams, but it is worth noting that the substrate usually needs to be heated to
400-600 °C. (Appl. Phys. Lett. 2014, 104, 242102.; J. Microelectromech. Syst. 2015,
24, 1951.), which provides sufficient energy for the movement of atoms to the liquid-
solid interface during the growth of silicon and germanium crystals. The electron
beam evaporation in this study is deposited on the substrate at room temperature,
which causes the atoms to cool immediately after nucleation on the substrate and
become unable to continue growing, eventually forming amorphous materials. Re-Fig.
49D shows the HRTEM image of the Si/Cr interface, in which the Si nanomembrane
still shows a disordered atomic arrangement, and the atoms in the Cr nanomembrane
show obvious grain boundary structures and grains. From the SAED characterization
of Re-Fig. 49H, it can be seen that the Cr nanomembrane exists in a polycrystalline
form, which exhibits obvious diffraction rings, and the diffraction ring distance in the
inverted space is highly consistent with the theoretical value. (William, M. H.
*Handbook of Chemistry and Physics 95th edn* (CRC Press, 2014).). Compared with Si
and Ge, which are semiconductor materials, the growth of Cr grains faces a relatively
small thermodynamic energy barrier, and their thermal conductivity is also good,
which will facilitate the growth during electron beam evaporation and ultimately form
a polycrystalline Cr nanomembrane.

In summary, we can conclude that the strain gradient between Si/Cr nanomembranes
is not due to lattice mismatch because the Si nanomembranes are amorphous
nanomembranes, which do not have an ordered lattice structure, and the Cr

nanomembranes are polycrystalline. The lattice constants of the two materials are
 very different, and they cannot achieve coherent crystal planes. Therefore, we believe
 that the strain gradient of Si/Cr nanomembranes originates from nanomembrane
 cooling after electron beam evaporation and the difference in thermal strain caused by
 different thermal expansion coefficients during cooling process after deposition.

 **Re-Fig. 49.** Characterization of the interface morphology of the nanomembrane
 sample substrate and analysis of the crystal form of each layer of nanomembrane.
 HRTEM images of (A) silicon substrate/silicon oxide, (B) silicon oxide/germanium
 nanomembrane, (C) germanium nanomembrane/silicon nanomembrane, (D) silicon
 nanomembrane/chromium nanomembrane interface. SAED images of (E) silicon
 substrate, (F) germanium nanomembrane, (G) silicon nanomembrane, and (H)
 chromium nanomembrane. The inset is the corresponding TEM images of the
 corresponding SAED area.

**Our modification:**

- 1. Re-Fig. 48-49 have been added as Supplementary Fig. 68-69.
 2. The answer has been added as "23.TEM characterization and crystallographic
 analysis of the Si/Cr nanomembrane" in Supporting Information.
 3. "TEM characterization and crystallographic analysis of the Si/Cr nanomembrane is
 performed by JEOL ARM200F (Supplementary Fig. 68-69)." has been added in
 manuscript.

**Comment:**

2: During the releasing process, did the nanomembranes remain a contact with
etchant or not? if not, does elastic energy during releasing need to consider the effect
of liquid adhesion on the nanomembranes, similar to the droplet evaporation-induced
mechanical bending or folding of materials (doi.org/10.1039/C8SM00873F). When
the geometric shape of nanomembranes changes, such as from rectangular to circular
or triangular one, did it affect the velocity of etchant flow or direction? if yes, how to
unify the multilevel FE simulation boundary conditions?

**Our response:**

Thank you again for your insightful question. During the release process shown in Fig.
2, the nanomembrane is always located in the etchant without additional liquid-
atmosphere or solid-atmosphere interfaces. For the local immersion etching in Fig. 5,
we found in experiments that there are differences between surface energy and layer
thickness between the ion-etched lower substrate channel and the silicon/chromium
nanomembrane. Therefore, the etchant liquid level will be slightly higher during the
etching process due to capillary action and the silicon/germanium nanomembrane.
But it is worth noting that the location where the etching occurs can be seen as
completely submerged in the liquid, as shown in Movie S4. At this time, since the
liquid covers the silicon/chromium nanomembrane, even if there is a local sacrificial
layer that is etched and there is an etchant between the nanomembrane and the
substrate, the surface tension of the nanomembrane will be compensated by the liquid
on both sides of the nanomembrane. Therefore, the impact of the difference on self-
assembly will be negligible. During the etching process, surface tension will mainly
act on the external gas-liquid-solid interface layer, which will be most obvious in the
channel formed by the silicon/chromium nanomembrane and the silicon oxide
substrate. Because the interface energy between hydrogen peroxide and water is
similar ($\gamma_{H_2O_2} = 74.0 \text{ mN/m}$, $\gamma_{H_2O} = 72.7 \text{ mN/m}$, $T = 293\text{K}$), we bring the surface
tension into an ideal condition, that is, when the liquid surface curvature radius is
equal to the width of the silicon oxide trench:

$$\sigma = \frac{\gamma}{R} \approx 7.3 \times 10^4 \text{ Pa},$$

where $R = 10 \text{ }\mu\text{m}$. However, when this stress acts on the Si/Cr nanomembrane, it will
only be applied to a very small area, that is, the interface position in the thickness
direction of the nanomembrane. At the same time, this pressure will only cause about

$3.6 \times 10^{-5}\%$ strain, and therefore, will have little impact on nanomembrane assembly
 (Re-Fig. 50).

 **Re-Fig. 50.** FEM results of (A) no surface tension and 10^5 Pa stress on the plane when
 etching Si/Cr nanomembrane.

Some studies have reported the capillary force model in U-shaped rectangular grooves
 (Sci. Rep. 2020, 10, 19709.). However, the depth of our groove is extremely small,
 and the surface tension outside the groove cannot be ignored, so this model is not used.
 The liquid level difference must be measured if this model is used. In the experiment,
 we observed that the liquid level will be integrated (Movie S4), which is
 approximately $10 \mu\text{m}$ higher than the silicon/chromium nanomembrane. At the same
 time, the groove width is $W = 20 \mu\text{m}$, and the depth is $D = 35 \text{ nm}$. The formula for
 calculating the liquid surface curvature in the U-shaped groove is:

$$h_{rec} = \frac{\sigma[(2D+W)\cos\theta - W]}{\rho g DW} = 10 \mu\text{m}.$$

the surface tension is calculated as $\sigma \sim 6.3 \times 10^{-9} \text{ Pa}$, which seriously deviates from
 the magnitude of stress that should be achieved at this scale. In the Movie S4, we can
 also observe that the self-assembled structure is always immersed under the liquid
 surface, which verifies this theory. At the same time, we will put the sample into
 supercritical point dryer after etching to avoid the impact of surface tension on the
 structure when the liquid is dried. Therefore, the effect of surface tension on the
 structure will be negligible during the conventional assembly process.

The structure control via surface tension also inspires us to utilize the droplet interface
 to enable morphological switching of the assembled micro-nano structures, which will
 bring an effective trigger method for reconfigurable micro-nano devices. In this case,
 the droplets will generate significant surface tension due to the natural drying process
 to regulate the nanomembrane structure. At this time, a large stress will occur in the

gap of the germanium sacrificial layer of about 60 nm between the nanomembrane
 and the substrate. At this time, the stress is approximately:

$$\sigma = \frac{\gamma}{R} \approx 1.2 \times 10^6 \text{ Pa}$$

In this condition, the droplets almost cover the bottom of the entire nanomembrane
 pattern, and the tension generated by the nanomembrane will have enough contact
 area to control the structure and shape of the nanomembrane. As the droplets
 gradually dry, the thickness of the liquid gap will further decrease, and greater stress
 will be generated to drive the nanomembrane to realize morphological reconstruction.
 This principle is similar to surface tension origami reported in previous studies (Soft
 Matter 2018, 14, 5968-5976; Nano Lett. 2019, 19, 6221-6226; Phys. Rev. Lett. 2007,
 98, 156103). In this experiment, a fixed end is introduced, and the switchable
 morphological modulation of ring-planar-ring is successfully realized. Compared with
 the complete immersion of liquid in the previous local etching process, the
 reconfiguration of nanomembranes mainly lies in the force generated by the surface
 tension on one side of the nanomembrane when the droplet dries (Re-Fig. 51). The
 method of using liquid to effectively control the morphology of microstructure is
 currently not possible through FEM due to the fast and massive load of surface
 tension. Overall, we think this is a very promising method, and we will study its
 modeling and application in the future.

 **Re-Fig. 51.** Demonstration of using liquid surface tension to control microstructure
 assembly morphology. (A-C) Surface tension is used to attract the nanomembrane on
 one side of the bottom layer to achieve the flattening process. (D) Si/Cr
 nanomembrane photodetector after flattening. (E-G) The surface tension of the liquid

is used to adsorb the top nanomembrane to restore the flat structure to a rolled-up
microstructure.

**Our modification:**

1. Video of reconfiguration in Re-Fig. 51 has been added as Movie S7.

2. "With proper control of surface tension, reconfiguration of different structures can
be achieved (Movie S7)." has been added in manuscript.

**Comment:**

3: *How to determine whether the assembled 3D structures reached a final stable*
*stage? And with the quasi-static condition, could the local size of 3D structures such*
*as local curvatures of taper or radius of tube during the etching process be predicted?*

**Our response:**

When the geometry of the nanomembrane changes, its boundary conditions will affect
the concentration distribution of the etchant, but the effect on the flow velocity and
concentration will be minimal, because the concentration distribution of the etchant in
the hydrostatic state mainly depends on its diffusion coefficient and concentration
layer. Moreover, the direction of the etchant flux will always follow the change of the
concentration gradient, which also reflects the broad applicability of multilevel FEM.

**Re-Fig. 52.** Effect of etching pattern on concentration and etchant flux distribution.
 (A) Schematic diagram of semicircular pattern parameters. (B) Schematic diagram of
 triangle pattern parameters. (C) Schematic diagram of parallelogram strip parameters.
 Concentration and etchant flux vector distribution during the etching process of (D)
 triangle, (E) semicircle, (F) parallelogram, and (G) rectangular strip.

First, we studied the concentration distribution and etchant flow direction of different
 forms of sacrificial layers. As shown in the Re-Fig. 52, the concentration gradient and
 etchant flow direction during etching of triangles, semicircles, parallelogram strips,
 and rectangles are all perpendicular to the sacrificial layer boundary, and the etchant
 concentration decreases as the distance between the sacrificial layer interface
 decreases. It shows that mass transfer mainly occurs at the solid-liquid interface (Re-
 Fig. 52).

**Re-Fig. 53.** Effect of pattern geometry on etching concentration and morphology. (A)

Initial setting of etching pattern. (B) The etching morphology changes and

concentration distribution when the pattern size is reduced to 0.1 times. (C) The

etching morphology changes and concentration distribution for the original pattern.

(D) The etching morphology changes and concentration distribution when the pattern

size is reduced to 0.1 times when the pattern size is expanded 10 times. The

morphological changes and concentration distribution of the sacrificial layer at the

same absolute time (E) when the pattern size is reduced to 0.1 times, (F) the original

pattern, and (G) the pattern size is expanded 10 times.

Subsequently, we analyzed the etchant concentration at the solid-liquid interface, as

shown in the Re-Fig. 54. It can be seen that for various sacrificial layer patterns, the
 concentration of the etchant at the solid-liquid interface is extremely low compared
 with the concentration of the etchant at the initial condition (10 mol/L) (Re-Fig. 54D),
 indicating that etching near the interface. The liquid has reacted completely. This data
 also shows that the chemical reaction rate in the H₂O₂/Ge system is greater than the
 diffusion rate, and it is a diffusion-controlled solid-liquid reaction system. For various
 sacrificial layer patterns, the area where the concentration gradient of the etchant
 exists is about 50-70 μm (Re-Fig. 52). This diffusion distance is only affected by the
 ratio of chemical reaction rate and diffusion rate. When the reaction rate is slow, the
 amount of material consumed will be less than the diffusion rate, resulting in a shorter
 concentration gradient area, as shown in Supplementary Fig. 4A. In summary,
 although the shapes of the sacrificial layer will affect the local distribution of
 concentration, it will always follow the principle of maximizing the flux of the etchant
 along the normal direction of the solid-liquid interface. Its overall distribution only
 depends on the ratio of chemical reaction to diffusion rate, proving broad applicability
 of this multi-layer FEM model.

**Re-Fig. 54.** Flow rate and concentration distribution of sacrificial layer etching during

the actual etching process. (A) Schematic diagram of the flow velocity field near the
 bubble during its rise. (B) Flow velocity changes at the measurement point in an ideal
 stationary fluid environment. (C) Changes in flow velocity at the measure point in the
 presence of bubble disturbance. (D) Concentration-time relationship of measure
 points during the etching process.

 **Re-Fig. 55.** The flow velocity distribution of (A) triangle, (B) semicircle, (C)
 parallelogram, (D) rectangular strip and corresponding coordinates of measure points
 during the etching and bubble rising process in Re-Fig. 54.

 Secondly, we have used FEM to demonstrate the impact on etching when there is a
 high-speed external flow field, which will not occur in the usual wet etching process
 (Supplementary Fig. 3A). We first analyzed the flow velocity field during etching in
 a completely stationary liquid, where the reference point is shown as the red dot in
 Re-Fig. 55. FEM results show that the total flow rate is negligible during completely
 stationary etching (Re-Fig. 54B). However, in the actual process, the generation and
 disappearance of bubbles and other disturbances will make this situation difficult to
 be achieved. Therefore, we established a transient model based on the changes in the
 liquid flow velocity field when the bubbles rise to observe the parts that have the most
 obvious impact on the model in actual experiments. Taking the rising rate of water

bubbles in a water tank as an example, we established a piecewise function to
simulate the influence of the flow field near the nanomembrane during the rising
process of bubbles (Re-Fig. 54A). Therefore, the flow velocity field defined is as
shown in the function, and the field is 200 μm from the reference point boundary
(Heat and Mass Transf. 2017, 53, 2885; Minerals, 2023, 13, 1130.):

$$1756 \quad \mathbf{v}_{\text{flow}} = \begin{cases} 0.2\vec{x} + 0.2\vec{y} \text{ m/s}, & 0\text{s} \leq t \leq 0.02 \text{ s} \\ 0 & 0.02\text{s} \leq t \leq 1 \text{ s} \end{cases},$$

where \vec{x} and \vec{y} are unit vector in x and y directions. The first section of the function is
used to simulate the flow velocity field when the bubble rises, and the second section
simulates the flow velocity field when the bubble disappears. At this time, we can see
that when the bubbles rise, the flow velocity near the sacrificial layer is only 1 cm/s.
As the bubbles disappear, the flow velocity near the sacrificial layer decreases rapidly.
The flow velocity only takes 0.05-0.1 s to drop to the corresponding level, which will
only cause minor influence on self-assembly. Therefore, the multilevel FEM model is
applicable to sacrificial layers of various shapes (Re-Fig. 54C). The etching process of
the sacrificial layer usually lasts for tens of minutes, and the number of oxygen
bubbles generated is small, so the impact of the flow rate on the self-assembly will be
negligible.

Re-Fig. 56. SEM images of self-assembled triangle nanomembranes and FEM results of the multilevel design model. (A-B) Panorama SEM images of self-assembled triangle nanomembranes. SEM images of self-assembled triangle nanomembranes with sizes of (C) $D = 24 \mu\text{m}$, (D) $D = 52 \mu\text{m}$, (E) $D = 100 \mu\text{m}$, and (F) $D = 110 \mu\text{m}$. FEM simulation results of triangle nanomembranes with sizes of (G) $D = 24 \mu\text{m}$, (H) $D = 52 \mu\text{m}$, (I) $D = 100 \mu\text{m}$, and (J) $D = 110 \mu\text{m}$.

**Re-Fig. 57.** SEM images of the self-assembled semicircle nanomembranes and FEM
 results of the multilevel design model. (A) Panorama SEM images of a self-assembled
 semicircle. SEM images of self-assembled semicircle nanomembranes with sizes of
 (B) $D = 20 \mu\text{m}$, (C) $D = 28 \mu\text{m}$, (D) $D = 54 \mu\text{m}$, (E) $D = 70 \mu\text{m}$. FEM results of self-
 assembled semicircle nanomembranes with sizes of (F) $D = 20 \mu\text{m}$, (G) $D = 28 \mu\text{m}$,
 (H) $D = 54 \mu\text{m}$, (I) $D = 70 \mu\text{m}$ of self-assembled semicircular FEM simulation results.
 To verify the broad applicability of the multilevel FEM system, we designed and
 simulated other patterns in the Si/Cr nanomembrane system. For semicircles and
 isosceles right triangles (Re-Fig. 56,57), we only obtain a type of tubular structure
 under conventional etching conditions, which is related to the tendency of the long-
 end rolling.

 **Re-Fig. 58.** SEM images of self-assembled parallelogram strips with different widths
 and FEM results of the multilevel design model. (A-B) Panorama SEM images of
 self-assembled parallelogram strip nanomembranes with different widths. SEM
 images of self-assembled parallelogram strip nanomembranes with sizes of (C) 4×40

1793 μm^2 , $\alpha_t = 45^\circ$, (D) $16 \times 40 \mu\text{m}^2$, $\alpha_t = 45^\circ$, (E) $30 \times 40 \mu\text{m}^2$, $\alpha_t = 45^\circ$, and (F) $80 \times 40 \mu\text{m}^2$,
 $\alpha_t = 45^\circ$. FEM simulation results of self-assembled parallelogram strip
 nanomembranes with sizes of (G) $4 \times 40 \mu\text{m}^2$, $\alpha_t = 45^\circ$, (H) $16 \times 40 \mu\text{m}^2$, $\alpha_t = 45^\circ$, (I)
 $30 \times 40 \mu\text{m}^2$, $\alpha_t = 45^\circ$, and (J) $80 \times 40 \mu\text{m}^2$, $\alpha_t = 45^\circ$.

For parallelogram strips with the same included angle and different length-to-width
 ratios, the rolled-up structures show a ring-helix-tube changing trend (Re-Fig. 58).

Due to the emergence of asymmetric patterns, the structure will tend to assemble into
 asymmetric microstructures that are similar to helix. As the width further increases,
 the obtuse angle position will form a wide-edge boundary condition during etching,
 and rolling will occur at this position, thus forming a structure between helix and tube.

**Re-Fig. 59.** SEM images of self-assembled parallelogram strips. (A) Panorama SEM
 images of self-assembled parallelogram strips. (B-I) SEM images of parallelogram
 strips with $\alpha_t = 10^\circ - 80^\circ$ (step: 10°). (J-Q) FEM results of parallelogram strips with $\alpha_t =$

10° - 80° (step: 10°).

For patterns with the same length-to-width ratio but changed angles (Re-Fig. 59), they
show ring-helix changes. At this time, the difference in small angle will further affect
the asymmetry of the etching. The rolling along its long diagonal sides will create a
more obvious helix structure. When the tilt angle gradually increases, this effect will
decay progressively and become similar to etching in a rectangular pattern.

**Our modification:**

1. Re-Fig. 52-59 has been added as Supplementary Fig. 7-14.

2. The answer has been added as "4. FEM modeling for different patterns" in
Supporting Information.

3. "In addition to the Si/Cr rectangular patterns, the multilevel FEM model has been
designed for various types of patterns such as semicircular, triangular, and
parallelogram patterns (Fig. 2d, Supplementary Fig. 7-14, and Movie S1, S2)" has
been added in manuscript.

4. Fig. 2d-h of manuscript has been revised and added as follow:

**Comment:**

4: How to determine whether the assembled 3D structures reached a final stable
stage? And with the quasistatic condition, could the local size of 3D structures such

as local curvatures of taper or radius of tube during the etching process be predicted?

**Our response:**

Thank you very much for your question. Our definition of stable state mainly starts
from two aspects: simulation and experiment. First, in multilevel FEM model, our
confirmation of the stable state primarily depends on the stability of the elastic energy.
Complete stabilization will greatly extend the simulation time since there is usually a
stress stabilization process in large deformation models with small meshes. Therefore,
we typically use the bulk viscosity coefficient of the elastic mechanics engine as
oscillation damping. When the elastic energy oscillation gradually decreases to an
acceptable range, the current step is considered to be a stable state, and the next step
of quasistatic analysis can be started. Usually, we consider that the amplitude of the
elastic energy oscillation before analysis is A_i . When A_i is less than 5% of the step
elastic performance difference ΔE_i , we consider the analysis step enters a stable state,
as shown in Re-Fig. 60.

**Re-Fig. 60.** Schematic diagram of elastic energy steady state criterion in the
multilevel design model. When $A_i \leq 5\% E_i$ ($i = 1, 2, \dots, n - 1$), the system is
considered to be in a stable state.

In experiments, our confirmation of whether a 3D structure reaches a final stable state
is based on the observation of the sample with a series of etching times. Here, we
provide videos of different patterns being etched in the etchant (Movie S3). It can be
seen that after the Si/Cr bilayer is completely released, the shape does not change with
the extension of etching time. At this time, we will consider the micro-nano structure
to be in a stable state. Our quasistatic conditions can theoretically predict the local
curvature of the model of micro-nano structures. It is difficult for us to measure the

biaxial curvature of micro-nano structures (such as cone structures), so we statistically
 verify the uniaxial curvature of the simulated tubular structure with experiments,
 which proves that the model is feasible for the prediction of uniaxial curvature. We
 designed Si/Cr nanomembrane systems with different thicknesses (Re-Table 4) for
 FEM prediction and experimental verification.

**Re-Table 4.** Si/Cr nanomembrane sample numbers and corresponding parameters

Sample number	Si thickness (nm)	Cr thickness (nm)
60	10
60	20
60	60
15	40
30	40
90	40
Standard	60	40

**Re-Fig. 61.** Diameter and strain distribution of Si/Cr self-rolled microtubes with
 different nanomembrane thicknesses. (A) FEM simulation, theoretical, and
 experimental results of the relationship between Si/Cr microtube radius and Cr
 nanomembrane thickness. (B) FEM simulation, theoretical, and experimental results
 of the relationship between Si/Cr microtube radius and Si nanomembrane thickness.
 The error bars are the standard deviation of radii in experiments.

Among them, the samples all show that the tube radius gradually decreases as the
 thickness increases, and some samples exhibit an increased radius when the thickness
 further increases. When $t_{Si} = 60 \text{ nm}$, while the Cr thickness continues to increase, the
 radii of the assembled tubes are $13.21 \pm 0.55 \mu\text{m}$ (sample #1), $8.13 \pm 0.34 \mu\text{m}$ (sample

#2), $7.48\pm 0.28 \mu\text{m}$ (standard sample), and $6.22\pm 0.59 \mu\text{m}$ (sample #3). (Re-Fig. 61A).
 When $t_{Cr} = 40 \text{ nm}$, while the Si thickness continues to increase, the radii of the
 assembled tubes are $6.95\pm 0.93 \mu\text{m}$ (sample #4), $3.67\pm 0.21 \mu\text{m}$ (sample #5), 7.48 ± 0.28
 1878 μm (standard sample), and $7.06\pm 1.93 \mu\text{m}$ (sample #6) (Re-Fig. 61B). As can be seen
 from the Re-Fig. 61A, B, our simulation results present results that are relatively close
 to the experiment, confirming its ability to finely predict geometric features of
 nanomembranes with different thicknesses. In addition, we also use the force-moment
 balance formula to estimate the radius:

$$1883 \quad R = \frac{(E_{Si})^2 t_{Si}^4 + (E_{Cr})^2 t_{Cr}^4 + 2E_{Si}E_{Cr}t_{Si}t_{Cr}(2t_{Si}^2 + 2t_{Cr}^2 + 3t_{Si}t_{Cr})}{6E_{Si}E_{Cr}t_{Si}t_{Cr}(t_{Si} + t_{Cr})\Delta\varepsilon}.$$

Here, the tube radii are fitted and analyzed, and therefore the Cr layer prestrain is
 obtained to be $\varepsilon_{Cr} \sim 0.65\%$, while the pre-strain of the Si layer is approximately
 $\varepsilon_{Si} \sim -0.55\%$. The corresponding stress gradient is $\Delta\sigma = \varepsilon_{Cr}E_{Cr} - \varepsilon_{Si}E_{Si} \approx$
 2600 MPa , showing good consistency with the preset values in the multilevel model.

**Re-Fig. 62.** SEM images and multilayer design model FEM results of Si (60 nm)/Cr
 (10 nm) double-layer rectangular nanomembrane (sample #1). (A-B) Panorama SEM
 images of self-assembled bilayer rectangular nanomembranes. SEM images of
 double-layer rectangular nanomembranes with sizes of (C) $4\times 40 \mu\text{m}^2$, (D) $10\times 40 \mu\text{m}^2$,
 (E) $20\times 40 \mu\text{m}^2$, (F) $40\times 40 \mu\text{m}^2$, (G) $60\times 40 \mu\text{m}^2$. FEM simulation results of double-
 layer rectangular nanomembranes with sizes of (H) $4\times 40 \mu\text{m}^2$, (I) $10\times 40 \mu\text{m}^2$, (J)
 $20\times 40 \mu\text{m}^2$, (K) $40\times 40 \mu\text{m}^2$, (L) $60\times 40 \mu\text{m}^2$.

images of self-assembled bilayer rectangular nanomembranes. SEM images of
double-layer rectangular nanomembranes with sizes of (C) $4 \times 40 \mu\text{m}^2$, (D) $10 \times 40 \mu\text{m}^2$,
(E) $20 \times 40 \mu\text{m}^2$, (F) $40 \times 40 \mu\text{m}^2$, (G) $60 \times 40 \mu\text{m}^2$. FEM simulation results of double-
layer rectangular nanomembranes with sizes of (H) $4 \times 40 \mu\text{m}^2$, (I) $10 \times 40 \mu\text{m}^2$, (J)
$20 \times 40 \mu\text{m}^2$, (K) $40 \times 40 \mu\text{m}^2$, (L) $60 \times 40 \mu\text{m}^2$.

**Re-Fig. 65.** SEM images and multilayer design model FEM results of 65 Si (15
1915 nm)/Cr (40 nm) double-layer rectangular nanomembrane (sample #4). (A-B)
Panorama SEM images of self-assembled bilayer rectangular nanomembranes. SEM
images of double-layer rectangular nanomembranes with sizes of (C) $4 \times 40 \mu\text{m}^2$, (D)
$10 \times 40 \mu\text{m}^2$, (E) $20 \times 40 \mu\text{m}^2$, (F) $40 \times 40 \mu\text{m}^2$, (G) $60 \times 40 \mu\text{m}^2$. FEM simulation results
of double-layer rectangular nanomembranes with sizes of (H) $4 \times 40 \mu\text{m}^2$, (I) 10×40
1920 μm^2 , (J) $20 \times 40 \mu\text{m}^2$, (K) $40 \times 40 \mu\text{m}^2$, (L) $60 \times 40 \mu\text{m}^2$.

Re-Fig. 66. SEM images and multilayer design model FEM results of Si (30 nm)/Cr (40 nm) double-layer rectangular nanomembrane (sample #5). (A-B) Panorama SEM images of self-assembled bilayer rectangular nanomembranes. SEM images of double-layer rectangular nanomembranes with sizes of (C) $4 \times 40 \mu\text{m}^2$, (D) $10 \times 40 \mu\text{m}^2$, (E) $20 \times 40 \mu\text{m}^2$, (F) $40 \times 40 \mu\text{m}^2$, (G) $60 \times 40 \mu\text{m}^2$. FEM simulation results of double-layer rectangular nanomembranes with sizes of (H) $4 \times 40 \mu\text{m}^2$, (I) $10 \times 40 \mu\text{m}^2$, (J) $20 \times 40 \mu\text{m}^2$, (K) $40 \times 40 \mu\text{m}^2$, (L) $60 \times 40 \mu\text{m}^2$.

Re-Fig. 67. SEM images and multilayer design model FEM results of Si (90 nm)/Cr

(40 nm) double-layer rectangular nanomembrane (sample #6). (A-B) Panorama SEM
images of self-assembled bilayer rectangular nanomembranes. SEM images of
double-layer rectangular nanomembranes with sizes of (C) $4 \times 40 \mu\text{m}^2$, (D) $10 \times 40 \mu\text{m}^2$,
(E) $20 \times 40 \mu\text{m}^2$, (F) $40 \times 40 \mu\text{m}^2$, (G) $60 \times 40 \mu\text{m}^2$. FEM simulation results of double-
layer rectangular nanomembranes with sizes of (H) $4 \times 40 \mu\text{m}^2$, (I) $10 \times 40 \mu\text{m}^2$, (J)
$20 \times 40 \mu\text{m}^2$, (K) $40 \times 40 \mu\text{m}^2$, (L) $60 \times 40 \mu\text{m}^2$.

Moreover, this model can also predict the morphology of the abovementioned
samples with biaxial or uniaxial curvature structures (Re-Fig. 62-67).

In summary, our multilevel design model has been verified with theory and
experiments, which illustrates the potential of this model in quantitative analysis of
geometric structures. In the future, we will also study the sampling and analysis of
biaxial curvature in multi-layer models to broaden the application field of this method.
We also hope to perform local strain control and precursor design on the model to
achieve higher accuracy and multi-layered design model with more detailed
information.

**Our modification:**

- 1. Re-Table 4 has been added as Supplementary Table 1.
- 2. Re-Fig. 61 have been added as Fig. 3 **d, e**.
- 3. Re-Fig. 62-67 have been added as Supplementary Fig. 24-30.
- 4. Answer of local curvature estimation has been added as "8. FEM modeling for
different Si/Cr thickness" in Supporting Information.
- 5. "Moreover, we designed Si/Cr systems containing Si nanomembranes (15/30/90 nm)
and Cr nanomembranes (10/20/60 nm) of different thicknesses to analyze the initial
strain and average strain of Si and Cr nanomembranes after release (Supplementary
Table 1 and Supplementary Fig. 24-30). As shown in Fig. 3**d, e**, we collected the radii
of tubular structures with the same Si thickness ($t_{Si} = 60 \text{ nm}$) and different Cr
thicknesses, and the same Cr thickness ($t_{Cr} = 40 \text{ nm}$) and different Si thicknesses.
With the increase of the thickness of Cr layer, the radii of assembled tubes with $t_{Si} =$
60 nm are $13.21 \pm 0.55 \mu\text{m}$ ($t_{Cr} = 10 \text{ nm}$), $8.13 \pm 0.34 \mu\text{m}$ ($t_{Cr} = 20 \text{ nm}$),
$7.48 \pm 0.28 \mu\text{m}$ ($t_{Cr} = 40 \text{ nm}$), and $6.22 \pm 0.59 \mu\text{m}$ ($t_{Cr} = 60 \text{ nm}$), respectively.
Meanwhile, with the increase of the thickness of Si layer, the radii of assembled tubes

with $t_{Cr} = 60$ nm are 6.95 ± 0.93 μm ($t_{Si} = 15$ nm), 3.67 ± 0.21 μm ($t_{Si} = 30$ nm),
 7.48 ± 0.28 μm ($t_{Si} = 60$ nm), and 7.06 ± 1.93 μm ($t_{Si} = 90$ nm), respectively. We
 used the above experiments and the force-moment balance formula to estimate the
 strain magnitude in the Si/Cr layer, that is

$$1970 \quad R = \frac{(E_{Si})^2 t_{Si}^4 + (E_{Cr})^2 t_{Cr}^4 + 2E_{Si}E_{Cr}t_{Si}t_{Cr}(2t_{Si}^2 + 2t_{Cr}^2 + 3t_{Si}t_{Cr})}{6E_{Si}E_{Cr}t_{Si}t_{Cr}(t_{Si} + t_{Cr})\Delta\varepsilon},$$

where E is Young's modules, t is thickness of nanomembrane, and $\Delta\varepsilon = \varepsilon_{Cr} - \varepsilon_{Si}$ is
 the initial strain difference between Si and Cr layer. We substitute the thicknesses of
 nanomembranes and tube radius (Fig. 31A, B) into this formula for fitting, and the
 prestrain of the Cr layer obtained by fitting is $\varepsilon_{Cr} \sim 0.65\%$, while the pre-strain of the
 Si layer is determined to be $\varepsilon_{Si} \sim -0.55\%$. The corresponding stress gradient is
 approximately $\Delta\sigma = \varepsilon_{Cr}E_{Cr} - \varepsilon_{Si}E_{Si} \approx 2600$ MPa, which is consistent with the
 preset condition in our simulation ($\Delta\sigma = 2000$ MPa), providing an important
 guarantee for high-accuracy structural design. In addition, we used the thickness and
 pre-strain of nanomembranes as parameters to calculate the strain relationship in the
 Si/Cr nanomembrane. The average strain of Si and Cr nanomembranes are also in
 good agreement with the experiment, providing an effective method for strain
 characterization of multilayer nanomembrane systems (Supplementary Fig. 31)." has
 been added in manuscript.

**----- End of Responses to comments -----**

----- List of Changes (manuscript)-----

1. Tianjun Cai is moved to the fourth position in the author list due to the conduction
of experiments. Mingze Ma and Xing Li are added into the author list due to the
conduction of experiments.

2. In Page 2, Lines 23, the word “releasing” have been revised to “release”.

3. In Page 2, Lines 24, the word “nanomembrane” have been revised to
“nanomembranes”.

4. In Page 2, Lines 24-26, the sentence “Here, we propose a quasistatic multilevel
finite element modeling (FEM) to assemble 3D structures from 2D nanomembrane
and offer verification results by Si/Cr bilayer nanomembranes.” have been revised to
“Here, we propose a quasistatic multilevel finite element modeling (FEM) to
assemble 3D structures from 2D nanomembranes and offer verification results by
various bilayer nanomembranes.”.

5. In Page 2, Lines 26-29, the sentence “We confirm that the 3D structural formation
is governed by both the minimum energy state and the geometric constraints imposed
by the edges of sacrificial layer.” have been revised to “Take Si/Cr nanomembrane as
an example, we confirm that the 3D structural formation is governed by both the
minimum energy state and the geometric constraints imposed by the edges of the
sacrificial layer.”.

6. In Page 3, Lines 42, the word “and/or” have been revised to “and”.

7. In Page 3, Lines 61, the word “considered as preference” have been revised to
“regarded as a preference”.

8. In Page 3, Lines 62, the word “dependence” have been revised to “the dependence”.

9. In Page 3, Lines 64, the word “analysis on” have been revised to “analysis in”.

10. In Page 4, Lines 68, the word “geometry” have been revised to “the geometry”.

11. In Page 4, Lines 71, the word “sacrificial layer” have been revised to “the
sacrificial layer”.

12. In Page 4, Lines 71-74, the sentences “The multilevel FEM method has been
successfully applied in a wide range of material systems, nanomembrane thicknesses,
pattern types, and pattern sizes, demonstrating excellent generalizability. We took
Si/Cr rectangular nanomembrane as an example to study the mechanism and
application of the self-assembly process in detail.” have been added.

13. In Page 4, Lines 80, the words “high uniformity” have been revised to “**high-**
 **uniformity**”.

14. Fig. 2d-h of manuscript has been revised and added as follow:

15. In Page 4, Lines 92, the word “substrate” has been revised to “**the** substrate”.

16. In Page 4, Lines 94, the words “as gradual movement” have been revised to “as **a**
 **gradual movement**”.

17. In Page 5, Lines 97, the words “at solid-liquid interface” have been revised to “at
 **the** solid-liquid interface”.

18. In Page 5, Lines 102, the words “with increase in aspect ratio” have been revised
 to “with **an** increase in aspect **ratios**”.

19. In Page 5, Lines 104, the words “etching direction” have been revised to “**the**
 **etching direction**”.

20. In Page 5, Lines 105, the words “both side” have been revised to “both **sides**”.

21. In Page 5, Lines 123, the words “prior rolling direction” have been revised to “**the**
 **prior rolling direction**”.

22. In Page 6, Lines 131, the words “etching system” have been revised to “**the**
 **etching system**”.

23. In Page 6, Lines 132, the words “FEM model” have been revised to “**the** FEM
 **model**”.

24. In Page 7, Lines 153, the words “of nanomembrane” have been revised to “of **the**

**nanomembranes**".

25. In Page 7, Lines 154, the words "Si/Cr bilayer" have been revised to "**the bilayer**

**nanomembranes**".

26. In Page 7, Lines 156, the word "quasistatic" has been revised to "**Quasistatic**".

27. In Page 7, Lines 158, the word "key" has been revised to "**crucial**".

28. In Page 7, Lines 158-168, the sentences "**In addition to the Si/Cr rectangular**

**patterns, the multilevel FEM model has been designed for various types of patterns**

**such as semicircular, triangular, and parallelogram patterns (Fig. 2d, Supplementary**

**Fig. 7-14, and Movie S1, S2), as well as in the high-frequency/low-frequency SiN_x**

**(LF/HF SiN_x) nanomembrane, NiTi nanomembrane, and VO₂/Cr nanomembrane**

**systems, which demonstrate the potential for a wide range of applications (Fig. 2e-h,**

**Supplementary Fig. 15-17). Meanwhile, the method demonstrates cross-scale**

**compatibility for models from the hundred nanometer scale to the hundred**

**micrometer scale (Supplementary Fig. 18-22). In this study, we choose the rectangular**

**Si/Cr rectangular nanomembrane for detailed analysis and discussion of the self-**

**assembly behavior due to their moderate parameter complexity and high geometrical**

**symmetry.**" have been added.

29. In Page 7, Lines 175, the words "one end" have been revised to "**one-end**".

30. In Page 8, Lines 178, the word "is" have been revised to "**are**".

31. In Page 8, Lines 194-195, the sentence "As can be seen in Fig. 3c, the probability

distribution of five types of morphology and pattern sizes is discussed from the

statistical view." has been revised to "**As shown in Fig. 3c, the probability distribution**

**of five types of morphology and pattern sizes is discussed from a statistical point of**

**view.**".

32. In Page 8, Lines 200-201, the sentence "The forming morphology of

nanomembranes also follows the order of ring-arch-helix-taper-tube with increasing

aspect ratio, and they are corresponding to the structure distribution in Fig. 1d and Fig.

2d." has been revised to "The forming morphology of nanomembranes also follows

the order of ring-arch-helix-taper-tube with increasing aspect ratio, **corresponding to**

**the structure distribution** in Fig. 1d and Fig. 2d.".

6. In Page 8-9, Lines 201-226, the sentences "**Moreover, we designed Si/Cr systems**

**containing Si nanomembranes (15/30/90 nm) and Cr nanomembranes (10/20/60 nm)**

**of different thicknesses to analyze the initial strain and average strain of Si and Cr**

nanomembranes after release (Supplementary Table 1 and Supplementary Fig. 24-30).
 As shown in Fig. 3d, e, we collected the radii of tubular structures with the same Si
 thickness ($t_{Si} = 60$ nm) and different Cr thicknesses, and the same Cr thickness
 ($t_{Cr} = 40$ nm) and different Si thicknesses. With the increase of the thickness of Cr
 layer, the radii of assembled tubes with $t_{Si} = 60$ nm are 13.21 ± 0.55 μm ($t_{Cr} =$
 10 nm), 8.13 ± 0.34 μm ($t_{Cr} = 20$ nm), 7.48 ± 0.28 μm ($t_{Cr} = 40$ nm), and
 6.22 ± 0.59 μm ($t_{Cr} = 60$ nm), respectively. Meanwhile, with the increase of the
 thickness of Si layer, the radii of assembled tubes with $t_{Cr} = 60$ nm are 6.95 ± 0.93 μm
 ($t_{Si} = 15$ nm), 3.67 ± 0.21 μm ($t_{Si} = 30$ nm), 7.48 ± 0.28 μm ($t_{Si} = 60$ nm), and
 7.06 ± 1.93 μm ($t_{Si} = 90$ nm), respectively. We used the above experiments and the
 force-moment balance formula to estimate the strain magnitude in the Si/Cr layer, that
 is

$$2085 \quad R = \frac{(E_{Si})^2 t_{Si}^4 + (E_{Cr})^2 t_{Cr}^4 + 2E_{Si}E_{Cr}t_{Si}t_{Cr}(2t_{Si}^2 + 2t_{Cr}^2 + 3t_{Si}t_{Cr})}{6E_{Si}E_{Cr}t_{Si}t_{Cr}(t_{Si} + t_{Cr})\Delta\varepsilon},$$

where E is Young's modules, t is thickness of nanomembrane, and $\Delta\varepsilon =$
 $\varepsilon_{Cr} - \varepsilon_{Si}$ is the initial strain difference between Si and Cr layer. We substitute the
 thicknesses of nanomembranes and tube radius (Fig. 31A, B) into this formula for
 fitting, and the prestrain of the Cr layer obtained by fitting is $\varepsilon_{Cr} \sim 0.65\%$, while the
 pre-strain of the Si layer is determined to be $\varepsilon_{Si} \sim -0.55\%$. The corresponding stress
 gradient is approximately $\Delta\sigma = \varepsilon_{Cr}E_{Cr} - \varepsilon_{Si}E_{Si} \approx 2600$ MPa, which is consistent
 with the preset condition in our simulation ($\Delta\sigma = 2000$ MPa), providing an
 important guarantee for high-accuracy structural design. In addition, we used the
 thickness and pre-strain of nanomembranes as parameters to calculate the strain
 relationship in the Si/Cr nanomembrane. The average strain of Si and Cr
 nanomembranes are also in good agreement with the experiment, providing an
 effective method for strain characterization of multilayer nanomembrane systems
 (Supplementary Fig. 31).” have been added.

33. In Page 9, Lines 228, the words “, however” have been revised to “. However”.

34. In Page 10, Lines 232, the words “combine effect” have been revised to “combine
 the effect”.

35. In Page 10, Lines 233, the words “with length” have been revised to “with a
 length”.

36. In Page 10, Lines 235, the words “release model” have been revised to “release
**models**”.

37. Fig. 3d, e of manuscript has been added as follow:

38. In Page 10, Lines 236-238, the words “region” have been revised to “**Region**”.

39. In Page 10, Lines 240-241, the words “results of simulation” have been revised to
“**the results of simulations**”.

40. In Page 10, Lines 244, the words “bilateral-released model” have been revised to
“**the bilateral-released model**”.

41. In Page 10, Lines 246, the words “opposite edge” have been revised to “**the**
**opposite edge**”.

42. In Page 10, Lines 247, the words “classic model” have been revised to “**the classic**
**model**”.

43. In Page 10, Lines 248, the words “after width” have been revised to “**after the**
**width**”.

44. In Page 10, Lines 251, the words “into ring” have been revised to “**into a ring**”.

45. In Page 10, Lines 258, the words “3D structures” have been revised to “**the 3D**
**structures**”.

46. In Page 11, Lines 261, the words “is corresponding” have been revised to
“**corresponds**”.

47. In Page 11, Lines 264, the words “to roll” have been revised to “**of rolling**”.

48. In Page 11, Lines 265-277, the sentences “**As shown in Fig. 4d, the quasistatic**
**FEM model simulates the change of α with etching time during the assembly process,**

and generally restores the relationship between α and etching time during the
 sacrificial layer etching process in the experiment (Fig. 4e, Movie S3). When the
 etching finished, the distribution of α are as follows: ring ($25^\circ - 35^\circ$), arch ($70^\circ - 80^\circ$),
 helix ($50^\circ - 65^\circ$), taper ($60^\circ - 75^\circ$), and tube ($40^\circ - 55^\circ$). The α angle obtained at the
 end of etching between model calculations and experimental results is highly
 consistent with the calculated α_{min} corresponding to each structure in Fig. 4c,
 providing an observation scheme for tracking intermediate states in the assembly
 process. We noticed that there is an obvious difference of the α between the multilevel
 model and the experiments in the early stage of etching, which is mainly due to the
 non-ideal release of sacrificial layer and local non-uniform etching caused by
 insufficient accuracy in micro-nano processing (Supplementary Fig. 41).” have been
 added.

49. In Page 11, Lines 280, the words “of elastic energy” have been revised to “of the
 elastic energy”.

50. In Page 12, Lines 295, the words “is contact with etchant” have been revised to “is
 in contact with the etchant”.

51. Fig. 4d, e of manuscript has been added as follow:

52. In Page 12, Lines 299, the words “boundary” have been revised to “the boundary”.

53. In Page 12, Lines 300, the word “forming” has been revised to “formation”.

54. In Page 12, Lines 304-307, the sentences “Compared to models in normal etching
 process, the movement of opposite edge in top etch is larger at the beginning, and

more area is released correspondingly in mechanical FEM modeling, leading to a
larger α and $\frac{v_W}{v_L}$ in local area (Supplementary Fig. 19).” have been revised to
“Compared to models in the normal etching process, the movement of the opposite
edge in the top etch is larger at the beginning. More area will be released
correspondingly in mechanical FEM modeling, leading to a larger α and $\frac{v_W}{v_L}$ in the
local area (Supplementary Fig. 44).”.

55. In Page 12-13, Lines 312-327, the sentences “With proper control of surface
tension, reconfiguration of different structure can be achieved (Movie S7). The local
etching method is also applicable to the polymorphic design of other patterns
(Supplementary Fig. 45-57). As shown in Fig. 5f, in parallelogram patterns, there is an
additional parameter tilt angle that define the geometric feature of patterns. In such
patterns, the bottom etching will result in the free-end of the patterns being released at
the end of the etch, when the structure will tend to assemble along the shorter
diagonal or the free edge of width. Conversely, when the top etching start at the free-
end first, and the localized etching will allow the assembly to develop along a longer
diagonal section to satisfy the lowest elastic energy¹⁶. We have successfully utilized
inclined etching to assemble the same parallelogram patterns along different diagonals,
thus forming two types of structures with significant differences (Fig. 5f,
Supplementary Fig. 53-57). It is worth noting that the premise for the preparation of
polymorphic microstructures is that there are multi-stable strain energy states when
the pattern is assembled, and there is an enough elastic energy barrier between the
stable states. Otherwise, the pattern will only assemble into one type of structures
(Supplementary Fig. 46-52).” have been added.

56. Fig. 5f of manuscript has been added as follow:

		Unrelease	Bottom etch	Simulation	Top etch	Simulation
L = 40 μm	W = 4 μm	i	i	i	i	i
	W = 10 μm	ii	ii	ii	ii	ii
	W = 20 μm	iii	iii	iii	iii	iii
	W = 40 μm	iv	iv	iv	iv	iv
	W = 60 μm	v	v	v	v	v

Mechanism	Bottom etch	Simulation	Top etch	Simulation
Bottom etch	 $60 \times 40 \mu\text{m}^2$ $\alpha_t = 45^\circ$		 $60 \times 40 \mu\text{m}^2$ $\alpha_t = 45^\circ$	
	$20 \times 40 \mu\text{m}^2$ $\alpha_t = 40^\circ$		$20 \times 40 \mu\text{m}^2$ $\alpha_t = 40^\circ$	
Top etch	$20 \times 40 \mu\text{m}^2$ $\alpha_t = 70^\circ$		$20 \times 40 \mu\text{m}^2$ $\alpha_t = 70^\circ$	

57. In Page 13, Lines 335, the words “while Cr layer” have been revised to “, while
 the Cr layer”.

58. Fig. 6 of manuscript has been revised as follow:

59. In Page 13, Lines 337-340, the sentences “Si/Cr photodetectors exhibit a
 maximum responsivity of 60 mA/W, response time of 100~700 μ s, and external
 quantum efficiency of 7~12%, which can effectively respond to 520nm incident light
 to achieve photodetection (Supplementary Fig. 60-62).” have been added.

60. In Page 13-14, Lines 347-350, the sentences “The prepared Si/Cr photodetector
 will be placed on a platform at the same height as the bottom of the incident light
 controller. Then the controller will be calibrated to ensure that the projected
 coordinate on YZ plane of (90°, 0°) input laser port is aligned with the center of
 photodetectors (Fig. 6a-b, Supplementary Fig. 63-64).” have been added.

61. In Page 14, Lines 350-354, the sentences “After traversal measurements, the
 photocurrent of each coordinate will be normalized and classified by structure types.
 In order to facilitate visual comparison of data, the normalization photocurrent of each
 structure will be displayed through the hemispherical top view perspective (i.e., XY
 plane).” has been revised to “After measurements of photoresponse of light incident
 from different coordinates, the photocurrent of each coordinate will be normalized
 and classified by structure types. To facilitate visual data comparison, the

normalization photocurrent of each structure will be displayed through **the projection**
**of light controller on YZ plane.**”.

62. In Page 14, Lines 375, the words “contains” have been revised to “**containing**”.

63. In Page 14, Lines 376, the words “the training of” have been revised to “**training**”.

64. In Page 14, Lines 378, the words “a accuracy” have been revised to “**an accuracy**”.

65. In Page 14, Lines 383 the words “corresponding angle” have been revised to “**the**
**corresponding angle**”.

66. In Page 15, Lines 392, the words “quasistatic” have been revised to “**Quasistatic**”.

67. In Page 15, Lines 394-397, the sentences “**This FEM method shows good**
**applicability in a variety of material systems, precursor pattern types, pattern sizes,**
**and nanomembrane thicknesses, which can also conduct preliminary geometric**
**feature analysis of the self-assembly process and results.**” have been added.

68. In Page 15, Lines 397, the words “Si/Cr bilayer nanomembrane” have been
revised to “Si/Cr bilayer **rectangular** nanomembrane”.

69. In Page 15, Lines 399, the words “etching process” have been revised to “**the**
**etching process**”.

70. In Page 15, Lines 399, the word “nanomembrane” has been revised to
“**nanomembranes**”.

71. In Page 15, Lines 403, the word “assistant” has been revised to “**assistance**”.

72. In Page 16, Lines 410, the words “sacrificial layer” have been revised to “**the**
**sacrificial layer**”.

73. In Page 16, Lines 415, the words “simulation” have been revised to “**the**
**simulation**”.

74. In Page 16, Lines 423, the word “released” has been revised to “**removed**”.

75. In Page 16, Lines 424, the word “Si and Cr layer” has been revised to “**the Si and**
**Cr layers**”.

76. In Page 16, Lines 430-432, the sentences “**For model with complex deformation**
**and significant volume overlap, contact module is applied to ensure the accuracy of**
**FEM simulations (Supplementary Fig. 67).**” have been added.

77. In Page 17, Lines 435, the words “direct laser writing” have been revised to “**the**
**direct laser writing**”.

78. In Page 17, Lines 441, the words “Ge sacrificial layer” have been revised to “**the**
**Ge sacrificial layer**”.

79. In Page 17, Lines 443, the words “Si/Cr bilayer” have been revised to “**the** Si/Cr
bilayer”.

80. In Page 17, Lines 446, the words “released Si/Cr bilayer” have been revised to
“**the** released Si/Cr bilayer”.

81. In Page 17, Lines 446, the words “released Si/Cr bilayer” have been revised to
“**the** released Si/Cr bilayer”.

82. In Page 17, Lines 452-454, the sentences “**TEM characterization and**
**crystallographic analysis of the Si/Cr nanomembrane is performed by JEOL**
**ARM200F (Supplementary Fig. 68-69).**” have been added.

83. In Page 18, Lines 460-462, the sentence “A multichannel data set contains 275
photocurrent each channel is collected from traversing measurement of various
structures and introduced into a 7-layer hidden layer.” has been revised to A
multichannel data set contains 275 photocurrent **data in** each channel, **which** is
collected from traversing **measurements** of various structures and introduced into a 7-
layer hidden layer.”.

84. Renumbering the index numbers of supplementary figures and movies.

85. In Page 23, Lines 622-624, the sentence “ Z.Z. and B.W. designed and conducted
the experiments and data analysis;” has been revised to “**Z.Z. and B.W. designed and**
**conducted and data analysis; Z.Z., B.W., T.C., M.M., and X.L. conducted the**
**experiments;**”.

86. In Page 24, Lines 642, the sentences “Elastic energy difference of nanomembranes
releasing from adjacent edge and opposite edge for different aspect ratio.” have been
revised to “Elastic energy difference of nanomembranes releasing from adjacent **and**
**opposite edge for different aspect ratios.**”.

87. In Page 25-26, Lines 647-657, the sentences “**quasistatic multilevel model FEM**
**simulation of bilayer pre-strain nanomembrane release.** Modeling process of
multilevel quasistatic FEM: **a** simulating sacrificial layer boundary movement of
release process, **b** collecting coordinate of boundary in discrete time points, **c**
importing coordinate into dynamic simulation as boundary condition. **d** Modeling and
result of quasistatic multilevel FEM modeling, in which moving boundaries are
derived from coordinates in the reaction-diffusion model.” have been revised to
“**Quasistatic multilevel model FEM simulation of bilayer pre-strain**
**nanomembrane release.** The modeling process of multilevel quasistatic FEM: a

simulating sacrificial layer boundary movement of the release process, **b** collecting
coordinates of boundary in discrete time points, **c** importing coordinates into the
dynamic simulation as a boundary condition. **d** Modeling and the result of quasistatic
multilevel FEM modeling, in which moving boundaries are derived from coordinates
in the reaction-diffusion model. Quasistatic multilevel FEM simulation and
experimental results of **e** 60 nm Si/40nm Cr triangle pattern, **f** 25 nm LF SiN_x/25 nm
HF SiN_x rectangle pattern, **g** 15 nm NiTi (0.2 Å/s)/ 15 nm NiTi (1.5 Å/s) rectangle
pattern, and **h** 120 nm VO₂/30 nm Cr rectangle patterns.”.

88. In Page 28, Lines 663-670, the sentences “ **c** Statistical table of the probability of
morphological features in pattern with different sizes of L from 20 to 80 μm with step
of 5 μm , and W from 2 to 80 μm with step of 2 μm , where the order number represent
corresponding structures in Fig. 3**b**.” have been revised and add as “**c** Statistical table
of the probability of morphological features in pattern with different sizes of L from
20 to 80 μm with **the** step of 5 μm , and W from 2 to 80 μm with **the** step of 2 μm ,
where the order numbers represent corresponding structures in Fig. 3**b**. **d**
Experimental result, theory calculation, and multilevel FEM simulation of radius of
tubes with fixed Si layer thickness ($t_{\text{Si}} = 60$ nm) and varied Cr layer thickness. **e**
Experimental result, theory calculation, and multilevel FEM simulation of radius of
tubes with fixed Cr layer thickness ($t_{\text{Cr}} = 40$ nm) and varied Si layer thickness. The
error bars in Fig. 3**d, e** are the standard deviation of radii in experiments.”.

89. In Page 29-30, Lines 676-682, the sentences “**Elastic energy and distribution of**
**self-assembly nanomembrane. a** Diagram of idealized bilateral release models that
consist two regions released in chronological order. **b** Relative elastic energy of
idealized bilateral release models with different direction angles and pattern widths
via FEM simulation. **c** Relationship between direction angle, distribution of various
morphology, and relative strain energy.” have been revised to “**Assembly mechanism**
**and elastic energy of self-assembly nanomembrane. a** Diagram of idealized
bilateral release models that consist **of two regions released chronologically. b**
Relative elastic energy of idealized bilateral release models with different direction
angles and pattern widths via FEM simulation. **c** Relationship between direction angle,
distribution of various morphology, and relative strain energy. **d** Multilevel FEM
simulations and **e** experimental results of relationship between direction angle α and
etching time.”.

90. In Page 32, Lines 689-692, the sentences “**f Mechanism, experimental results, and**
 **quasistatic FEM simulation results of multimorphic assembly in Si/Cr bilayer**
 **parallelogram patterns.**” have been added.

91. In Page 34, Lines 693-703, the sentences “**a Schematic of DNN-assisted analysis**
 **of the incident angle. Schematic of omnidirectional photoresponse measurement setup.**
 **b-g XY plane projection of the photoresponse through the b planar, c ring, d arch, e**
 **helix, f taper, and g tube. h The epoch-dependent recognition accuracy of training set**
 **in longitudinal and latitudinal direction using DNN. i Angle identification of incident**
 **angle in longitudinal ($\varphi = 0^\circ$) and latitudinal ($\theta = 40^\circ$) direction. j Weight mapping of**
 **output angles and input angles of longitudinal direction ($\varphi = 0^\circ$).” have been revised**
 **to “a Schematic of the omnidirectional photoresponse measurement setup. A**
 **hemispherical PMMA light controller is aligned and placed on the photodetectors. b**
 **Si/Cr photodetectors at the spherical center of omnidirectional light controller. c**
 **Schematic of DNN-assisted analysis of the incident angle. d Relationship between**
 **responsivity and power of different photodetectors. e Current-Time relationship of**
 **different structures illuminated by 520 nm laser. f-k XY plane projection of the**
 **photoresponse through the f planar, g ring, h arch, i helix, j taper, and k tube. l The**
 **epoch-dependent recognition accuracy of training set in longitudinal and latitudinal**
 **direction using DNN. m Angle identification of incident angle in longitudinal ($\varphi = 0^\circ$)**
 **and latitudinal ($\theta = 40^\circ$) direction. n Weight mapping of output angles and input**
 **angles of longitudinal direction ($\varphi = 0^\circ$).”.**

----- **List of Changes (Supporting Information)**-----

1. The content list of Supporting Information is updated from 13 parts and 3 movies to
 **22 parts and 7 movies, and the index number of supplementary figures are**
 **renumbered.**

**List of Contents**

1. Reaction-diffusion model for H ₂ O ₂ /Ge system	3
2. Model selection for sacrificial layer	12
3. FEM modeling for unidirectional rolling	14
4. FEM modeling for different patterns	15
5. FEM modeling for various material systems	25

6. FEM applicability for different sizes	30
7. Fabrication of Si/Cr bilayer	35
8. FEM modeling and strain analysis for different Si/Cr thickness	37
9. Calculation model and verification for elastic energy	48
10. Elastic energy calculation for unilateral rolling	51
11. SEM images for standard sample of Si/Cr bilayer	53
12. Relationship between occurrence possibility of bilayer nanomembrane and
relative strain energy	56
13. Tracking of self-assembly process of Si/Cr nanomembrane	58
14. Formation and distribution of 3D structure array	60
15. Reaction-diffusion model and FEM in bottom etching	61
16. Multilevel design model in FEM in top etching	63
17. Design and fabrication of multimorphic structures via controlled local etching	64
18. SEM images and 2D angular photodetection for Si/Cr photodetectors	77
19. Photodetection mechanism and performance of Si/Cr photodetectors	79
20. Setup of incident light directional detection	87
21. FEM parameters, results, and experiments of steady/unsteady reaction-diffusion
model	89
22. Contact module in FEM model	91
23. TEM characterization and crystallographic analysis of the Si/Cr nanomembrane	94
Legends for Movie S1 to S7	98
SI References	99

2. The references of SI are updated and renumbered.

3. **Response to Comment #3 of Reviewer #3 and Re-Fig. 52-59 (Page 78-87 of this**
**file)** have been added as **“4. FEM modeling for different patterns”** in Supporting
Information as follow:

**4.FEM modeling for different patterns**

When the geometry of the nanomembrane changes, its boundary conditions
will affect the concentration distribution of the etchant, but the effect on the flow
velocity and concentration will be minimal, because the concentration distribution of

the etchant in the hydrostatic state mainly depends on its diffusion coefficient and
 concentration layer. Moreover, the direction of the etchant flux will always follow the
 change of the concentration gradient, which also reflects the broad applicability of
 multilevel FEM.

 **Supplementary Fig. 7.** Effect of etching pattern on concentration and etchant flux
 distribution. (A) Schematic diagram of semicircular pattern parameters. (B)
 Schematic diagram of triangle pattern parameters. (C) Schematic diagram of
 parallelogram strip parameters. Concentration and etchant flux vector distribution
 during the etching process of (D) triangle, (E) semicircle, (F) parallelogram, and (G)
 rectangular strip.

First, we studied the concentration distribution and etchant flow direction of
 different forms of sacrificial layers. As shown in the Supplementary Fig. 7, the

concentration gradient and etchant flow direction during etching of triangles,
 semicircles, parallelogram strips and rectangles are all perpendicular to the sacrificial
 layer boundary, and the etchant concentration decreases as the distance between the
 sacrificial layer interface decreases. It shows that mass transfer mainly occurs at the
 solid-liquid interface.

 **Supplementary Fig. 8** Effect of pattern geometry on etching concentration and
 morphology. (A) Initial setting of etching pattern. (B) The etching morphology
 changes and concentration distribution when the pattern size is reduced to 0.1 times.
 (C) The etching morphology changes and concentration distribution for the original
 pattern. (D) The etching morphology changes and concentration distribution when the

pattern size is reduced to 0.1 times when the pattern size is expanded 10 times. The
morphological changes and concentration distribution of the sacrificial layer at the
same absolute time (E) when the pattern size is reduced to 0.1 times, (F) the original
pattern, and (G) the pattern size is expanded 10 times.

Subsequently, we analyzed the etchant concentration at the solid-liquid
interface, as shown in the Supplementary Fig. 8, 9. It can be seen that for various
sacrificial layer patterns and different sizes, the concentration of the etchant at the
solid-liquid interface is extremely low compared with the concentration of the etchant
at the initial condition (10 mol/L) (Supplementary Fig. 9D), indicating that etching
near the interface The liquid has reacted completely. This data also shows that the
chemical reaction rate in the $\text{H}_2\text{O}_2/\text{Ge}$ system is greater than the diffusion rate, and it
is a diffusion-controlled solid-liquid reaction system. For various sacrificial layer
patterns, the area where the concentration gradient of the etchant exists is about 50-70
2399 μm (Supplementary Fig. 7,8). This diffusion distance is only affected by the ratio of
2400 chemical reaction rate and diffusion rate. When the reaction rate is slow, the amount
of material consumed will be less than the diffusion rate, resulting in a shorter
concentration gradient area, as shown in Supplementary Fig. 4A. In summary,
although the shapes of the sacrificial layer will affect the local distribution of
concentration, it will always follow the principle of maximizing the flux of the etchant
along the normal direction of the solid-liquid interface. Its overall distribution only
depends on the ratio of chemical reaction to diffusion rate, proving broad applicability
of this multi-layer FEM model.

Supplementary Fig. 9. Flow rate and concentration distribution of sacrificial layer etching during the actual etching process. (A) Schematic diagram of the flow velocity field near the bubble during its rise. (B) Flow velocity changes at the measurement point in an ideal stationary fluid environment. (C) Changes in flow velocity at the measure point in the presence of bubble disturbance. (D) Concentration-time relationship of measure points during the etching process.

**Supplementary Fig. 10.** The flow velocity distribution of (A) triangle, (B) semicircle,
 (C) parallelogram, (D) rectangular strip and corresponding coordinates of measure
 points during the etching and bubble rising process in Supplementary Fig. 9.

Secondly, we have used FEM to demonstrate the impact on etching when
 there is a high-speed external flow field, which will not occur in the usual wet etching
 process (Supplementary Fig. 3A). We first analyzed the flow velocity field during
 etching in a completely stationary liquid, where the reference point is shown as the
 red dot in Supplementary Fig. 10. FEM results show that the total flow rate is
 negligible during completely stationary etching (Supplementary Fig. 9B). However, in
 the actual process, the generation and disappearance of bubbles and other disturbances
 will make this situation difficult to be achieved. Therefore, we established a transient
 model based on the changes in the liquid flow velocity field when the bubbles rise to
 observe the parts that have the most obvious impact on the model in actual
 experiments. Taking the rising rate of water bubbles in a water tank as an example,
 we established a piecewise function to simulate the influence of the flow field near the
 nanomembrane during the rising process of bubbles (Supplementary Fig. 9A).
 Therefore, the flow velocity field defined is as shown in the function, and the field is
 200 μm from the reference point boundary^{1,2} :

$$v_{\text{flow}} = \begin{cases} 0.2\vec{x} + 0.2\vec{y} \text{ m/s}, & 0s \leq t \leq 0.02s \\ 0 & 0.02s \leq t \leq 1s \end{cases}$$

Where \vec{x} and \vec{y} are unit vector in x and y directions. The first section of the function is used to simulate the flow velocity field when the bubble rises, and the second section simulates the flow velocity field when the bubble disappears. At this time, we can see that when the bubbles rise, the flow velocity near the sacrificial layer is only 1 cm/s. As the bubbles disappear, the flow velocity near the sacrificial layer decreases rapidly. The flow velocity only takes 0.05-0.1 s to drop to the corresponding level, which will only cause minor influence on self-assembly. Therefore, the multilevel FEM model is applicable to sacrificial layers of various shapes (Supplementary Fig. 9C). The etching process of the sacrificial layer usually lasts for tens of minutes, and the number of oxygen bubbles generated is small, so the impact of the flow rate on the self-assembly will be negligible.

A

B

C

D

E

F

G

H

I

J

Supplementary Fig. 11. SEM images of self-assembled triangle nanomembranes and FEM results of the multilevel design model. (A-B) Panorama SEM images of self-

2452 assembled triangle nanomembranes. SEM images of self-assembled triangle
 nanomembranes with sizes of (C) $D = 24 \mu\text{m}$, (D) $D = 52 \mu\text{m}$, (E) $D = 100 \mu\text{m}$, and (F)
 $D = 110 \mu\text{m}$. FEM simulation results of triangle nanomembranes with sizes of (G) D
 $= 24 \mu\text{m}$, (H) $D = 52 \mu\text{m}$, (I) $D = 100 \mu\text{m}$, and (J) $D = 110 \mu\text{m}$.

 **Supplementary Fig. 12.** SEM images of the self-assembled semicircle
 nanomembranes and FEM results of the multilevel design model. (A) Panorama SEM
 images of a self-assembled semicircle. SEM images of self-assembled semicircle
 nanomembranes with sizes of (B) $D = 20 \mu\text{m}$, (C) $D = 28 \mu\text{m}$, (D) $D = 54 \mu\text{m}$, (E) $D =$
 $70 \mu\text{m}$. FEM results of self-assembled semicircle nanomembranes with sizes of (F) D
 $= 20 \mu\text{m}$, (G) $D = 28 \mu\text{m}$, (H) $D = 54 \mu\text{m}$, (I) $D = 70 \mu\text{m}$ of self-assembled
 semicircular FEM simulation results.

Supplementary Fig. 13. SEM images of self-assembled parallelogram strips with different widths and FEM results of the multilevel design model. (A-B) Panorama SEM images of self-assembled parallelogram strip nanomembranes with different widths. SEM images of self-assembled parallelogram strip nanomembranes with sizes of (C) $4 \times 40 \mu\text{m}^2$, $\alpha_t = 45^\circ$, (D) $16 \times 40 \mu\text{m}^2$, $\alpha_t = 45^\circ$, (E) $30 \times 40 \mu\text{m}^2$, $\alpha_t = 45^\circ$, and (F) $80 \times 40 \mu\text{m}^2$, $\alpha_t = 45^\circ$. FEM simulation results of self-assembled parallelogram strip nanomembranes with sizes of (G) $4 \times 40 \mu\text{m}^2$, $\alpha_t = 45^\circ$, (H) $16 \times 40 \mu\text{m}^2$, $\alpha_t = 45^\circ$, (I) $30 \times 40 \mu\text{m}^2$, $\alpha_t = 45^\circ$, and (J) $80 \times 40 \mu\text{m}^2$, $\alpha_t = 45^\circ$.

**Supplementary Fig. 14.** SEM images of self-assembled parallelogram strips. (A)
 Panorama SEM images of self-assembled parallelogram strips. (B-I) SEM images of
 parallelogram strips with $\alpha_t = 10^\circ$ - 80° (step: 10°). (J-Q) FEM results of parallelogram
 strips with $\alpha_t = 10^\circ$ - 80° (step: 10°).

For different pattern designs, we tried triangles, semicircles, and various
 parallelograms (Supplementary Fig. 10-14). In triangles and semicircles, the
 multilevel model only gets tubular structures. In parallelograms with different angles
 and aspect ratios, the model is also designed by designing corresponding etching
 paths, in which the asymmetry in pattern leads to the assembly of helix structure, and
 the experimental results once again verified the feasibility of this design and process.

4. A part of response to Comment #4 of Reviewer #1 and Re-Fig. 7-9 (Page 16-18 of
 this file) have been added as “5. FEM modeling for various material systems” in
 Supporting Information as follow:

**5.FEM modeling for various material systems**

For the multilevel FEM model, its universality to different material systems will be
important for the future development of electronic devices and microrobots. Therefore,
we selected three other representative material systems in micro-nano devices for
feasibility verification: low-frequency/high-frequency silicon nitride (HF/LF SiN_x) for
optical resonators and capacitors^{3,4}, NiTi nanomembranes for microrobots⁵, and
VO₂/Cr nanomembrane for bolometers and actuators^{6,7}. These three systems include
several on-chip and off-chip applications realized by micro-nano 3D structures, and
there are also great differences in strain states. The experimental results are also
consistent with the simulation of the FEM model.

**Supplementary Fig. 15.** SEM images of self-assembled SiN_x nanomembrane and
FEM results of the multilevel design model. (A-B) Panorama SEM images of self-
assembled silicon nitride nanomembranes. (C) 4×20 μm², (D) 10×20 μm², (E) 20×20

2505 μm^2 , (F) $24 \times 20 \mu\text{m}^2$, (G) $36 \times 20 \mu\text{m}^2$, (H) $60 \times 20 \mu\text{m}^2$, SEM images of self-assembled
SiN_x nanomembrane. FEM simulation results of self-assembled SiN_x nanomembranes
with sizes of (I) $4 \times 20 \mu\text{m}^2$, (J) $10 \times 20 \mu\text{m}^2$, (K) $20 \times 20 \mu\text{m}^2$, (L) $24 \times 20 \mu\text{m}^2$, (M)
$36 \times 20 \mu\text{m}^2$, (N) $60 \times 20 \mu\text{m}^2$.

First, we will demonstrate self-assembly based on HF/LF SiN_x nanomembranes
(Supplementary Fig. 15). We take this system into the multilevel FEM model for
analysis, and the result shows a transition of ring-taper-tube structure. As can be seen
from the SEM images, the system based on dual-frequency silicon nitride exhibits a
ring-helix-taper-tube structure with size from $2 \times 20 \mu\text{m}^2$ to $80 \times 20 \mu\text{m}^2$ (2-20/22-
28/30-44/46 -80 μm) and stress gradient of 1500 MPa⁸. Among them, the structural
characteristics of the helix are mainly reflected in the asymmetry of the rolling process.
We obtained experiment results that are highly consistent with the FEM
model, verifying that the structure of self-assembled HF/LF SiN_x is controlled by the
boundary and size of the sacrificial layer at the same time. The multilevel model
provides an effective means for high-precision design and potential performance
control of microstructure inductors and optical microcavity devices.

**Supplementary Fig. 16.** SEM images of self-assembled NiTi alloy nanomembrane
and FEM results of multilayer design model. (A) Panorama SEM image of self-
assembled NiTi nanomembranes. SEM images of self-assembled nanomembranes

with sizes of (B) $30 \times 40 \mu\text{m}^2$, (C) $38 \times 40 \mu\text{m}^2$, (D) $42 \times 40 \mu\text{m}^2$, (E) $60 \times 40 \mu\text{m}^2$, and (F)
$72 \times 40 \mu\text{m}^2$. FEM simulation results of self-assembled nanomembranes with sizes of
(G) $4 \times 40 \mu\text{m}^2$, (H) $10 \times 40 \mu\text{m}^2$, (I) $20 \times 40 \mu\text{m}^2$, (J) $40 \times 40 \mu\text{m}^2$, and (K) $60 \times 40 \mu\text{m}^2$.

Subsequently, we demonstrated the preparation of self-assembled structures of NiTi
alloy nanomembranes (Supplementary Fig. 16). The usual preparation method of NiTi
alloy microstructure is based on oblique deposition of photoresist sacrificial layer.
Among them, grazing angle deposition can provide a fixed end and a free end for the
photoresist through the shadow effect⁹. We use the difference in deposition rate
during electron beam evaporation of NiTi nanomembranes to introduce strain
differences and use the interlayer strain gradient to achieve self-assembly of the
nanomembranes. Since the etching rate of photoresist in acetone is extremely fast and
it is an etching model dominated by the diffusion process, the boundary conditions of
the sacrificial layer will be similar to the Si/Cr system. In the FEM model, it is worth
noting that under large strains ($2 \sim 3\%$ ⁵, 2.5% for FEM model), even if the boundary
conditions cause obvious long-side rolling at the lateral sides. When the patterns are
fully released, they will still roll unidirectionally from the short end due to the
deformation limit of the fixed end, resulting in most of the self-assembly, and the
results are all unidirectional tubes. In the experiment, the structures we obtained are
all ring-tube structures, and few helix-type structures existed, which verified the
applicability of this model to situations where one end is dominant under diffusion-
reaction equilibrium. Introducing this multilevel design method will facilitate the
mass production of microrobots and the refined design of actuators.

**Supplementary Fig. 17.** SEM images of self-assembled VO₂/Cr nanomembrane and
 FEM results of the multilayer design model. (A-B) SEM images of 40×400 μm² self-
 assembled VO₂/Cr nanomembranes. (C) SEM image of 200×200 μm² self-assembled
 VO₂/Cr nanomembrane. FEM simulation results of self-assembled nanomembranes
 with sizes of (D-E) 40×400 μm² and (F) 200×200 μm²

We used VO₂, a type of heat-sensitive functional material, combined with the
 Cr layer for verification (Supplementary Fig. 17). We studied the self-assembled 120
 2558 nm VO₂/30 nm Cr system and conducted structural design and prediction of the self-
 2559 assembled VO₂ nanomembrane patterns of 200×200 μm² and 50×500 μm. Due to the
 2560 large thickness of vanadium oxide, the radii of its ring and tube will increase
 accordingly. However, patterns with an aspect ratio = 1:1 still exhibit a tube structure.
 This is because the VO₂ strain is relatively large (0.6~0.8%⁷, 0.8% for FEM), which
 can still cause a significant strain gradient even when the Cr layer is thin, allowing the
 taper to overcome both ends. The competitive energy barriers of multidirectional
 rolling in the etching process are finally realized to assemble the tubular structure.
 Among them, the 200×200 μm² self-assembled nanomembrane exhibits the shape of
 taper and tube, most of which are tube-type structures.

5. Response to Comment #5 of Reviewer #2 and Re-Fig. 42-46 (Page 60-65 of this
 file) have been added as “6. FEM applicability for different sizes” in Supporting
 Information as follow:

**6.FEM applicability for different sizes**

For 2D nanomembranes, the size effects involved in shrinking and
 amplification are quite different. For the two-dimensional patterns involved in this
 manuscript with a size of 2 to 80 μm , we will discuss the two cases separately. When
 the overall size is reduced by 10 times, the minimum size will be close to 200 nm,
 while the thickness of the nanomembrane system is about 100 nm, which means that
 the self-rolling theory and the plane strain/plane stress theory in the simulation will no
 longer apply. In this condition, the strain perpendicular to the thickness direction will
 not be negligible¹⁰, that is,

$$\varepsilon_z = \frac{1}{E}(\sigma_z - \mu(\sigma_x + \sigma_y)) \neq 0$$

When the width is equal to the thickness ($W = 0.4 \mu\text{m}$, $t_{Si} + t_{Cr} = 0.1 \mu\text{m}$),
 which will lead to a significant increase in the radius of curvature (Supplementary Fig.
 18, 19). At the same time, the length of the cantilever beam will be significantly
 shortened, and its large deformation behavior will also be significantly weakened. It
 can be seen from the FEM that the patterns with aspect ratio $\frac{W}{L}$ from 0.1 to 2 all
 exhibit uniaxial bending, and their small deformation and a small distance between
 the fixed end make it difficult to form multi-directional rolling during the release
 process.

 **Supplementary Fig. 18.** FEM simulation results of double-layer rectangular
 nanomembranes with pattern sizes of (A) $0.4 \times 4 \mu\text{m}^2$, (B) $1 \times 4 \mu\text{m}^2$, (C) $2 \times 4 \mu\text{m}^2$, (D)

$4 \times 4 \mu\text{m}^2$, and (E) $6 \times 4 \mu\text{m}^2$.

**Supplementary Fig. 19.** Tube radius ratio between the size of $(0.4-6) \times 4 \mu\text{m}^2$ (R) and
$60 \times 40 \mu\text{m}^2$ (R_0).

At the same time, the surface tension and liquid flow in the nano-scale
structure will be significantly enhanced, resulting in a significant deterioration in the
forming stability and difficulty to control the rolling direction¹¹, which will also
introduce additional considerations for the simulation. Moreover, the sharp reduction
in sizes makes the structure preparation a new problem because the conventional
photolithography process will make it difficult to process sub-micron-scale patterns,
and other processes (such as EBL) will further increase the process time and cost.

A

B

C

D

E

F

G

H

I

J

K

**Supplementary Fig. 20.** SEM images and multilayer design model FEM results of a

double-layer rectangular nanomembrane with a size factor of 0.5 times. (A-B)
 Panorama SEM images of self-assembled bilayer rectangular nanomembranes. SEM
 images double-layer rectangular nanomembranes with sizes of (C) $2 \times 20 \mu\text{m}^2$, (D)
 $5 \times 20 \mu\text{m}^2$, (E) $10 \times 20 \mu\text{m}^2$, (F) $20 \times 20 \mu\text{m}^2$, and (G) $30 \times 20 \mu\text{m}^2$. FEM simulation
 results of double-layer rectangular nanomembranes with sizes of (H) $2 \times 20 \mu\text{m}^2$, (I)
 $5 \times 20 \mu\text{m}^2$, (J) $10 \times 20 \mu\text{m}^2$, (K) $20 \times 20 \mu\text{m}^2$, and (L) $30 \times 20 \mu\text{m}^2$.

When the size is scaled to 0.5 times, their pattern sizes are considerably
 increased compared to 0.1 times, but the length of the released cantilever beam is still
 short, which cannot make the self-assembled Si/Cr nanomembrane roll in a stable
 state from multiple directions. Therefore, it will be challenging to fabricate a taper-
 type structure at this scale (Supplementary Fig. 20).

**Supplementary Fig. 21.** SEM images and multilayer design model FEM results of a
 double-layer rectangular nanomembrane with a size factor of 5 times. (A-B)
 Panorama SEM images of self-assembled bilayer rectangular nanomembranes. SEM
 images of double-layer rectangular nanomembranes with sizes of (C) $20 \times 200 \mu\text{m}^2$, (D)
 $50 \times 200 \mu\text{m}^2$, (E) $100 \times 200 \mu\text{m}^2$, (F) $200 \times 200 \mu\text{m}^2$, and (G) $300 \times 200 \mu\text{m}^2$. FEM
 simulation results of double-layer rectangular nanomembranes with sizes of (H)

$20 \times 200 \mu\text{m}^2$, (I) $50 \times 200 \mu\text{m}^2$, (J) $100 \times 200 \mu\text{m}^2$, (K) $200 \times 200 \mu\text{m}^2$, and (L) 300×200
2626 μm^2

**Supplementary Fig. 22.** SEM images and multilayer design model FEM results of
double-layer rectangular nanomembranes with a size factor of 10 times. (A-F) SEM
images of double-layer rectangular nanomembranes with a size factor of 10 times.
FEM simulation results of double-layer rectangular nanomembranes with sizes of (G)
$40 \times 400 \mu\text{m}^2$, (H) $100 \times 400 \mu\text{m}^2$, (I) $200 \times 400 \mu\text{m}^2$, (J) $400 \times 400 \mu\text{m}^2$, and (K) 600×400
2633 μm^2 .

According to the simulation of the multilevel design model, when the pattern
size is enlarged 5-10 times, the patterns will tend to roll unidirectionally in multiple
directions from each edge, eventually forming a complex 3D structure composed of
various tubular structures and local bending (Supplementary Fig. 21-22). For the FEM
simulation of large-size patterns, we did not use the contact module. When
considering the contact module, the mesh needs to be as small as possible to avoid the
problem of the nanomembrane being unable to roll due to collisions of self-rolling
meshes, which will significantly extend the calculation time to decades of days.

Moreover, we only used the incomplete etching time points in the FEM simulation to
show the corresponding insufficient etching structure in the SEM. Only when the
width is small (20~40 μm), the nanomembrane tends to form a multi-turn rolled
structure. Since the cantilever beams are too long, the multi-turn rolled structure will
be difficult to maintain a tight structure, thus forming a structure between helix and
ring (Supplementary Fig. 21A). In the experiment, we observed multi-turn rolled
structures and long-side unidirectional rolled structures in the smaller width area,
which also means that the disturbance during the release process of the sacrificial
layer at large sizes will have a more significant impact on the self-assembly process.
At the same time, for broader nanomembrane patterns, the larger bending moment
caused by the long edge rolling during the rolling process will cause the Si/Cr
nanomembrane and the edge of the Si nanomembrane to tear, thus further affecting
the yield.

In summary, the size effect will significantly affect the self-assembly process
of pre-strained nanomembranes. At smaller sizes, we need to consider the failure of
plane elasticity theory and the weakening of large deformation effects, limiting the
diversity of nanomembrane structure assembly. When the size is larger, we need to
consider the occurrence of local rolling in multiple directions and possible problems
such as tearing and breakage of nanomembrane during experiments, which usually
result in lower yields. Although the distribution of microstructures will be affected by
size effects, our multilevel model still performs good applicability for structural
design and prediction in the scale range of hundreds of nanometers to hundreds of
micrometers.

6. Response to Comment #3 of Reviewer #2, Re-Fig. 33-40, and Re-Table 2 (Page 45-
57 of this file) have been added as “8. FEM modeling and strain analysis for different
Si/Cr thickness” in Supporting Information as follow:

**8.FEM modeling and strain analysis for different Si/Cr thickness**

For the bilayer nanomembrane system, the self-rolling moment is contributed
by the strain distribution in the thickness direction and the integral of the thickness
between the nanomembranes, so the thickness of Si and Cr will affect the radius
structure after the moment balance of the system. Therefore, we designed a series of
Si/Cr nanomembrane models with different thickness combinations and calculated the

radius distribution under different thicknesses of Si and Cr layers with the same pre-
 strain using Nikishkov's curvature formula¹². The Si/Cr systems we designed with
 different parameters are Si nanomembranes (15/30/90 nm) and Cr nanomembranes
 (10/20/60 nm) of different thicknesses. The above samples are shown in
 Supplementary Table 1.

**Supplementary Table 1.** Si/Cr nanomembrane sample numbers and corresponding
 parameters.

Sample number	Si thickness (nm)	Cr thickness (nm)
60	10
60	20
60	60
15	40
30	40
90	40
Standard	60	40

We then conducted self-assembly experiments on samples from the above
 parameters and studied the relationship between morphology-size distribution and
 tube radius-thickness distribution.

Supplementary Fig. 24. SEM images and FEM results of Si (60 nm)/Cr (10 nm) double-layer rectangular nanomembrane (sample #1). (A-B) Panorama SEM images of self-assembled bilayer rectangular nanomembranes. SEM images of double-layer rectangular nanomembranes with different sizes of (C) $4 \times 40 \mu\text{m}^2$, (D) $10 \times 40 \mu\text{m}^2$, (E) $20 \times 40 \mu\text{m}^2$, (F) $40 \times 40 \mu\text{m}^2$, and (G) $60 \times 40 \mu\text{m}^2$. FEM simulation results of double-layer rectangular nanomembranes with different sizes of (H) $4 \times 40 \mu\text{m}^2$, (I) $10 \times 40 \mu\text{m}^2$, (J) $20 \times 40 \mu\text{m}^2$, (K) $40 \times 40 \mu\text{m}^2$, and (L) $60 \times 40 \mu\text{m}^2$

Supplementary Fig. 25. SEM images and multilayer design model FEM results of Si (60 nm)/Cr (20 nm) double-layer rectangular nanomembrane (sample #2). (A-B) Panorama SEM images of self-assembled bilayer rectangular nanomembranes. SEM images of double-layer rectangular nanomembranes with different sizes of (C) $4 \times 40 \mu\text{m}^2$, (D) $10 \times 40 \mu\text{m}^2$, (E) $20 \times 40 \mu\text{m}^2$, (F) $40 \times 40 \mu\text{m}^2$, and (G) $60 \times 40 \mu\text{m}^2$. FEM simulation results of double-layer rectangular nanomembranes with different sizes of (H) $4 \times 40 \mu\text{m}^2$, (I) $10 \times 40 \mu\text{m}^2$, (J) $20 \times 40 \mu\text{m}^2$, (K) $40 \times 40 \mu\text{m}^2$, and (L) $60 \times 40 \mu\text{m}^2$

Supplementary Fig. 26. SEM images and multilayer design model FEM results of Si (60 nm)/Cr (60 nm) double-layer rectangular nanomembrane (sample #3). (A-B) Panorama SEM images of self-assembled bilayer rectangular nanomembranes. SEM images of double-layer rectangular nanomembranes with different sizes of (C) $4 \times 40 \mu\text{m}^2$, (D) $10 \times 40 \mu\text{m}^2$, (E) $20 \times 40 \mu\text{m}^2$, (F) $40 \times 40 \mu\text{m}^2$, and (G) $60 \times 40 \mu\text{m}^2$. FEM simulation results of double-layer rectangular nanomembranes with different sizes of (H) $4 \times 40 \mu\text{m}^2$, (I) $10 \times 40 \mu\text{m}^2$, (J) $20 \times 40 \mu\text{m}^2$, (K) $40 \times 40 \mu\text{m}^2$, and (L) $60 \times 40 \mu\text{m}^2$.

 **Supplementary Fig. 27.** SEM images and multilayer design model FEM results of Si
 (15 nm)/Cr (40 nm) double-layer rectangular nanomembrane (sample #4). (A-B)
 Panorama SEM images of self-assembled bilayer rectangular nanomembranes. SEM
 images of double-layer rectangular nanomembranes with different sizes of (C) 4×40
 μm², (D) 10×40 μm², (E) 20×40 μm², (F) 40×40 μm², and (G) 60×40 μm². FEM
 simulation results of double-layer rectangular nanomembranes with different sizes of
 (H) 4×40 μm², (I) 10×40 μm², (J) 20×40 μm², (K) 40×40 μm², and (L) 60×40 μm²

Supplementary Fig. 28. SEM images and multilayer design model FEM results of Si (30 nm)/Cr (40 nm) double-layer rectangular nanomembrane (sample #5). (A-B) Panorama SEM images of self-assembled bilayer rectangular nanomembranes. SEM images of double-layer rectangular nanomembranes with different sizes of (C) $4 \times 40 \mu\text{m}^2$, (D) $10 \times 40 \mu\text{m}^2$, (E) $20 \times 40 \mu\text{m}^2$, (F) $40 \times 40 \mu\text{m}^2$, and (G) $60 \times 40 \mu\text{m}^2$. FEM simulation results of double-layer rectangular nanomembranes with different sizes of (H) $4 \times 40 \mu\text{m}^2$, (I) $10 \times 40 \mu\text{m}^2$, (J) $20 \times 40 \mu\text{m}^2$, (K) $40 \times 40 \mu\text{m}^2$, (L) $60 \times 40 \mu\text{m}^2$.

Supplementary Fig. 29. SEM images and multilayer design model FEM results of Si

(90 nm)/Cr (40 nm) double-layer rectangular nanomembrane (sample #6). (A-B)
 Panorama SEM images of self-assembled bilayer rectangular nanomembranes. SEM
 images of double-layer rectangular nanomembranes with different sizes of (C) 4×40
 2736 μm^2 , (D) 10×40 μm^2 , (E) 20×40 μm^2 , (F) 40×40 μm^2 , and (G) 60×40 μm^2 . FEM
 simulation results of double-layer rectangular nanomembranes with different sizes of
 (H) 4×40 μm^2 , (I) 10×40 μm^2 , (J) 20×40 μm^2 , (K) 40×40 μm^2 , (L) 60×40 μm^2 .

**Supplementary Fig. 30.** Relationship between structures and widths of the samples.

First, we observed the ring-arch-helix-taper-tube structural transition in
 various types of self-assembled Si/Cr nanomembranes by multilevel models and
 experiments (Supplementary Fig. 24-29). However, according to the structure-width
 relationship diagram (Supplementary Fig. 30), it can be seen that there are obvious
 differences in the transition regions of each structure. For example, in sample #3, the
 taper-tube transition occurs at a narrower width $W = 36\sim 38 \mu\text{m}$, and in sample #4,
 there is almost no tube structure. As the thickness of the Si nanomembrane continues
 to increase, the Si/Cr nanomembrane will be prone to forming helix and arch
 structures (samples #4-#6). It indicates that the structural distribution is significantly
 related to the system strain during the self-assembly of nanomembranes. Therefore,
 we started with the radius of various tubular structures to study the strain state of the
 Si/Cr bilayer nanomembrane.

**Supplementary Fig. 31.** Strain distribution of Si/Cr self-rolling microtubes under
 different nanomembrane thicknesses. (A) Dependence of the average strain of the Cr
 layer on the pre-strain of the Cr layer and the thickness of the Cr layer in Si/Cr
 microtubes. (B) Dependence of the average strain of the Si layer on the pre-strain of
 the Si layer and the thickness of the Si layer in Si/Cr microtubes.

We used the above experiments and the force-moment balance^{12,13} formula to
 estimate the strain magnitude in the Si/Cr layer, that is

$$R = \frac{(E_{Si})^2 t_{Si}^4 + (E_{Cr})^2 t_{Cr}^4 + 2E_{Si}E_{Cr}t_{Si}t_{Cr}(2t_{Si}^2 + 2t_{Cr}^2 + 3t_{Si}t_{Cr})}{6E_{Si}E_{Cr}t_{Si}t_{Cr}(t_{Si} + t_{Cr})\Delta\varepsilon},$$

where E is Young's modules, t is thickness of nanomembrane, and $\Delta\varepsilon = \varepsilon_{Cr} - \varepsilon_{Si}$ is the initial strain difference between Si and Cr layer. We substitute the set thickness and tube radius distributions into this formula for fitting. The relationship between the tube radius and the thickness of the nanomembrane we obtained is shown in the Supplementary Fig. 31A, B, in which the prestrain of the Cr layer obtained by fitting is $\varepsilon_{Cr} \sim 0.65\%$, the pre-strain of the Si layer is approximately $\varepsilon_{Si} \sim -0.55\%$. The corresponding stress gradient is approximately $\Delta\sigma = \varepsilon_{Cr}E_{Cr} - \varepsilon_{Si}E_{Si} \approx 2600 \text{ MPa}$, which is consistent with the preset conditions in our simulation ($\Delta\sigma = 2000 \text{ MPa}$) is highly close, providing an important guarantee for high-accuracy structural design. We can see that as the thicknesses of Si and Cr nanomembranes continue to increase, the Si/Cr tube radius shows a decreasing-increasing trend. As shown in Supplementary Fig. 31A, when $t_{Si} = 60 \text{ nm}$, the Cr thickness continues to increase, the radii of the assembled tubes are $13.21 \pm 0.55 \mu\text{m}$ (sample #1), 8.13 ± 0.34

2776 μm (sample #2), $7.48\pm 0.28 \mu\text{m}$ (standard sample), and $6.22\pm 0.59 \mu\text{m}$ (sample #3). As
 shown in Supplementary Fig. 31B, when $t_{Cr} = 40 \text{ nm}$ As the Si thickness continues to
 increase, the radii of the assembled tubes are $6.95\pm 0.93 \mu\text{m}$ (sample #4), 3.67 ± 0.21
 2779 μm (sample #5), $7.48\pm 0.28 \mu\text{m}$ (standard sample), and $7.06\pm 1.93 \mu\text{m}$ (sample #6).
 With the increase in thickness, maintaining the same strain layer requires a more
 significant strain difference between the two materials. Therefore, when the strain
 difference between the two materials is constant, a large layer thickness will always
 lead to an increase in the tube radius. We also obtained the strain distribution in the
 thickness direction by simulating the same prestrained Si/Cr system with different
 thicknesses.

 In addition, the radius-thickness relationship of each sample obtained by the
 finite element design model also shows good consistency with experiments and
 formula calculations. For tubular models of various thicknesses, the multilayer design
 models all show larger tube radii than the theoretical models. This is because the
 elastic mechanics model used in the modeling process is a plane stress model, and for
 wider strip self-assembly, its mechanical behavior will be closer to the plane strain
 model, and the model can be modified as

$$R = \frac{(E'_{Si})^2 t_{Si}^4 + (E'_{Cr})^2 t_{Cr}^4 + 2E'_{Si}E'_{Cr}t_{Si}t_{Cr}(2t_{Si}^2 + 2t_{Cr}^2 + 3t_{Si}t_{Cr})}{6E'_{Si}E'_{Cr}t_{Si}t_{Cr}(t_{Si} + t_{Cr})\Delta\varepsilon'}$$

, where $E'_i = E_i/(1 - \nu_i^2)$, $\Delta\varepsilon' = (1 + \nu_i)\Delta\varepsilon$. The correction of Young's modulus and
 strain gradient will improve the accuracy of the model in a wider range of situations.

We studied the influence of the thickness and initial strain of the other layer on
 the average strain in the nanomembrane layer when the parameters of the Si layer and
 the Cr layer are fixed, respectively. Among them, the average strain is calculated by
 the following formula:

$$\underline{\varepsilon} = \frac{\int_{t_{i-1}}^{t_i} \varepsilon(z) dz}{\sum_{i=1}^2 t_i}$$

It is used to characterize the strain state and strain size of the entire
 nanomembrane. First, we chose $t_{Si} = 60 \text{ nm}$, $\varepsilon_{Si} = -0.55\%$ to quantify the effect of
 t_{Cr} and ε_{Cr} (Supplementary Fig. 31C). It can be seen from the scanning image that
 when the top layer is Cr, its overall strain state mainly manifests as tensile strain,
 which is consistent with the mechanical behavior observed in the experiment. When

the Si/Cr nanomembrane is released from the substrate, the tensile strain The Cr
nanomembrane in the stretched state will be released and will be subject to relative
compressive strain. In contrast, the Si nanomembrane will be subject to more tensile
strain. As the thickness of the Cr nanomembrane continues to increase, its own strain
will dominate the overall strain state of the nanomembrane. For each sample in our
experiment, as the Cr thickness increases from 10 nm to 60 nm, the average strain
shows a small attenuation from 0.9% to 0.5% and maintains the tensile strain state. It
is worth noting that the increase in average strain relative to prestrain mainly
originates from the neutral axis position $y_b = \frac{\sum_{i=1}^n E_i t_i (y_i + y_{i-1})}{2 \sum_{i=1}^n E_i t_i}$, $y_i = \sum_{k=1}^i t_k$ during
rolling. The change in the overall strain will also lead to a gradual increase in the tube
radius as the nanomembrane thickness further increases.

Subsequently, we chose $t_{Cr} = 40 \text{ nm}$, $\varepsilon_{Cr} = 0.65\%$ to quantify the effect of
t_{Si} and ε_{Si} (Supplementary Fig. 31D). The average strain relationship will show
multiple rising and falling trends due to the interaction between the tube radius and
the neutral axis, the neutral axis, and the thickness of each layer. When $t_{Si} =$
$40 \sim 100 \text{ nm}$, The abnormal prestrain-average strain region that appears mainly comes
from the fact that when the neutral axis is located near the Si layer, the bending
moments in the layer cancel each other out. When the thickness of the Si layer further
increases, the strain within the Si layer is still dominated by its pre-strained state. In
the experiment, as the Si nanomembrane thickness increased from 15 to 90 nm, the
average strain of each sample remained at a compressive strain state of $-0.5\% \sim -0.6\%$,
which is highly consistent with the experimental verification and model design of
Si/Cr double-layer nanomembranes.

7. Revising the title of the part 11 from “SEM images for Si/Cr bilayer with different
sizes” to “SEM images for standard sample of Si/Cr bilayer”.

8. A part of response to Comment #6 of Reviewer #2 and Re-Fig. 47C-H (Page 65-68
of this file) have been added as “13. Tracking of self-assembly process of Si/Cr
nanomembrane” in Supporting Information as follow:

**13. Tracking of self-assembly process of Si/Cr nanomembrane**

Supplementary Fig. 41. Non-ideal situations in FEM and experiments. (A-C) Ideal etching boundary as envisioned in the FEM model. (D-F) The local anisotropic release process and optical images exist in the experiment.

As shown in the Fig. 4D, we obtained the trend of α changing with etching time for various structures in the FEM. They all exhibit a trend of gradually increasing with the etching process. At the end of etching, the obtained α are $\sim 35^\circ$ (4 μm , ring), $\sim 78^\circ$ (10 μm , arch), $\sim 64^\circ$ (20 μm , helix), $\sim 63^\circ$ (40 μm , taper), $\sim 53^\circ$ (60 μm , tube), which is also close to the α angle corresponding to the lowest elastic energy we obtained from the energy calculation, which are $30^\circ\sim 40^\circ$ (ring), $50^\circ\sim 80^\circ$ (arch), $65^\circ\sim 75^\circ$ (helix), $50^\circ\sim 65^\circ$ (taper), $40^\circ\sim 55^\circ$ (tube). We noticed that at $W = 20 \mu\text{m}$, the angle-time relationship fluctuated in the early stages of etching, which is caused by a sudden change in the rolling direction when the helix structure switched from an initial bidirectional rolling to a unidirectional rolling. Through the sacrificial layer etching video (Movie), we also obtained the α -time curves of various structures above. The α obtained at the end of the etching are $\sim 26^\circ$ (4 μm , ring), $\sim 71^\circ$ (10 μm , arch), $\sim 56^\circ$ (20 μm , helix), $\sim 64^\circ$ (40 μm , taper), and $\sim 40^\circ$ (60 μm , tube), which are also in the corresponding lowest energy α region (Supplementary Fig. 4E). It is worth noting that although the α angle at the end of etching is consistent with that predicted by the multilevel model, the α -time relationship during the process is different from the model in the initial stage of etching, which is mainly composed of two reasons. First of all, in the experiments, the etched area is not always released uniformly with the etching time. For example, the assembly of the 40 μm area into the taper shape will mainly occur at the end of etching (Movie S). At the same time, the multilevel model stipulates the time required for each quasistatic step is the same (Supplementary Fig. 41C-E). Secondly, in the experiment, we observed the phenomenon of α first increasing and then decreasing during the initial etching of the $W = 60 \mu\text{m}$ pattern. This is due to the preferential local rolling on both sides of the rectangle during the actual experiment (Supplementary Fig. 41F-H). It will lead to a change in the dominant rolling direction during initial etching^{14,15}. As etching continued,

experiments and multilevel designs finally gave similar results, but it also shows that
there is room for improvement in the prediction accuracy of the model.

9. Response to Comment #2 of Reviewer #2 and Re-Fig. 16-32 (Page 30-44 of this
file) have been added as “17. Design and fabrication of multimorphic structures via
controlled local etching” in Supporting Information as follow:

**17.Design and fabrication of multimorphic structures via controlled local etching**

The rectangles we used in the manuscript as a demonstration are due to their
moderate symmetry and variable control complexity, so the relationship between the
elastic energy of the system and the assembled structure can be obtained more directly
in the multilevel design method. For circular or semicircular patterns, the parameters
are mainly controlled by the radius. The relationship between the self-assembly
behavior lacks the aspect ratio as a parameter to diversify the structure type. Therefore,
it is difficult to obtain enough information to analyze the forming mechanism. For the
tilted parallelogram strip, we will introduce length L , width W , and tilt angle α_t to
analyze the assembly process. The tilt angle as an additional parameter will increase
the data magnitude by one dimension, which may be helpful for detailed analysis of
the relationship between the self-assembly and the result, but it will increase the cost
of comprehensive simulation and analysis. Here, we exhibit some multilevel design
examples and experimental results with patterns of semicircle, triangle, and
parallelogram strips to verify the universality of the multilevel model.

Since the circular pattern has a short connection area with the fixed end, it is
easy to tear and fall off from the Si fixed end during the etching process. Therefore,
we used a semicircular pattern with a longer connection area for research.

Supplementary Fig. 45. Schematic diagram of the etching process of different shapes. The red dotted line in the figure is the moving boundary of the sacrificial layer, and the black arrow is the etching direction. Geometric parameters of (A) semicircle, (B) isosceles right triangle, (C) parallelogram strip. Schematic diagram of the sacrificial layer boundary changes of (D) semicircle, (E) isosceles right triangle, and (F) parallelogram strip that are completely immersed in the etchant. Schematic diagram of the boundary changes of the sacrificial layer during top etching of (G) semicircle, (H) isosceles right triangle, and (I) parallelogram strip. Schematic diagram of the boundary changes of the sacrificial layer during bottom etching of (J) semicircle, (K) isosceles right triangle, and (L) parallelogram strip.

As shown in Supplementary Fig. 45, we have set models for semicircles with different radii R , isosceles right triangles with different diagonal lengths D , parallelogram strips with the same included angle α_t and different aspect ratios $\frac{W}{L}$, and parallelogram strips with the same length-to-width ratio at different included angles. FEM analysis and experiments are conducted, confirming the broad applicability of the model (an example of the sacrificial layer etching animation of the above pattern

is attached to the Movie S1 and S2). For the top etching (Supplementary Fig. 45G-I)
 and bottom etching (Supplementary Fig. 45J-L) processes, we used the same method
 as modeling rectangular patterns, that is, adding a preliminary release area for the top
 etching and stepwise etching from both sides to the top for the bottom etching.

 **Supplementary Fig. 46.** SEM images and FEM results of the bottom-etched self-
 assembled semicircle of the multilevel design model. (A) Panorama SEM images of
 the bottom-etched self-assembled semicircle. SEM images of bottom-etched self-
 assembled semicircles with sizes of (B) $D = 20 \mu\text{m}$, (C) $D = 28 \mu\text{m}$, (D) $D = 54 \mu\text{m}$,
 and (E) $D = 70 \mu\text{m}$. FEM results of bottom-etched self-assembled semicircles with
 sizes of (F) $D = 20 \mu\text{m}$, (G) $D = 28 \mu\text{m}$, (H) $D = 54 \mu\text{m}$, and (I) $D = 70 \mu\text{m}$.

 **Supplementary Fig. 47.** SEM images and FEM results of the top-etched self-
 assembled semicircle of the multilevel design model. (A) Panorama SEM images of
 the top-etched self-assembled semicircle. SEM images of top-etched self-assembled

semicircles with sizes of (B) $D = 20 \mu\text{m}$, (C) $D = 28 \mu\text{m}$, (D) $D = 54 \mu\text{m}$, and (E) $D =$
$70 \mu\text{m}$. TEM results of top-etched self-assembled semicircles with sizes of (F) $D = 20$
2925 μm , (G) $D = 28 \mu\text{m}$, (H) $D = 54 \mu\text{m}$, and (I) $D = 70 \mu\text{m}$.

**Supplementary Fig. 48.** SEM images and FEM results of the bottom-etched self-
assembled triangle. (A-B) Panorama SEM images of bottom-etched self-assembled
triangles. SEM images of bottom-etched self-assembled triangles with sizes of (C) D
$= 24 \mu\text{m}$, (D) $D = 52 \mu\text{m}$, (E) $D = 100 \mu\text{m}$, and (F) $D = 110 \mu\text{m}$. FEM simulation
results of the bottom-etched triangle with sizes of (G) $D = 24 \mu\text{m}$, (H) $D = 52 \mu\text{m}$, (I)
$D = 100 \mu\text{m}$, and (J) $D = 110 \mu\text{m}$.

Supplementary Fig. 49. SEM images and FEM results of the top-etched self-assembled triangle. (A) Panorama SEM images of the top-etched self-assembled triangle. SEM images of top-etched self-assembled triangles with sizes of (B) $D = 24 \mu\text{m}$, (C) $D = 52 \mu\text{m}$, (D) $D = 100 \mu\text{m}$, and (E) $D = 110 \mu\text{m}$. FEM simulation results of the top-etched triangle with sizes of (F) $D = 24 \mu\text{m}$, (G) $D = 52 \mu\text{m}$, (H) $D = 100 \mu\text{m}$, and (I) $D = 110 \mu\text{m}$.

Supplementary Fig. 50. SEM images and FEM results of immersive-etched self-assembled triangles of different heights. (A) Panorama SEM images of immersive-etched self-assembled triangles of different heights. (B) SEM images and (C) FEM simulation results of immersive-etched self-assembled triangles with $D = 20 \mu\text{m}$ and triangle bottom height $H = 80 \mu\text{m}$.

Supplementary Fig. 51. SEM images and FEM results of bottom-etched self-assembled triangles with different heights. (A) Panorama SEM images of self-assembled triangles with different heights etched on the bottom. (B) SEM images and (C) FEM simulation results of bottom-etched self-assembled triangles with $D = 20 \mu\text{m}$ and $H = 80 \mu\text{m}$.

Supplementary Fig. 52. SEM images of self-assembled triangles with different heights etched on the top and FEM results of the multilevel design model. (A) Panorama SEM images of self-assembled triangles with different heights etched on the top. (B) SEM images and (C) FEM simulation results of top-etched self-assembled triangles with $D = 20 \mu\text{m}$ and $H = 80 \mu\text{m}$.

For semicircles (Supplementary Fig. 12,46,47) and isosceles right triangles
(Supplementary Fig. 11,48,49), we only obtained tubular structures under immersive-
etched conditions in both FEM simulations and experiments, because the
nanomembranes tended to roll in one direction. It is related to the fact that the etching
path always tends to form local long-end rolls at the free end. It is difficult to control
multiple etching directions during the etching process, and thus, it is hard for
semicircles and triangles to form multi-stable microstructures. For acute-angled
isosceles triangles, various types of etching also show similar results (Supplementary
Fig. 50-52). Among them, the helix morphology formed under immersive etching
exhibits more asymmetry due to its asymmetry in all directions etching. An unstable
local state is more likely to exist in the isotropic etching path, causing asymmetric
rolling in the initial etching process.

**Supplementary Fig. 53.** Schematic diagram of the identification of multi-stable states
during the etching process. (A) Schematic diagram of the etching direction in a
parallelogram. (B) The strain energy-etching direction relationship diagram shows a
strain energy barrier between the two lowest energy points.

**Supplementary Fig. 54.** SEM images and FEM results of immersive-etched
 parallelogram strips with different widths. (A-B) Panorama SEM images of
 immersive-etched self-assembled parallelogram strip nanomembranes with different
 widths. SEM images of self-assembled parallelogram strip immersive-etched
 nanomembrane with sizes of (C) $4 \times 40 \mu\text{m}^2$, $\alpha_t = 45^\circ$, (D) $16 \times 40 \mu\text{m}^2$, $\alpha_t = 45^\circ$, (E)
 $30 \times 40 \mu\text{m}^2$, $\alpha_t = 45^\circ$, and (F) $80 \times 40 \mu\text{m}^2$, $\alpha_t = 45^\circ$, $\alpha_t = 45^\circ$. FEM simulation results of
 self-assembled immersive-etched parallelogram strip nanomembrane with sizes of (G)
 $4 \times 40 \mu\text{m}^2$, $\alpha_t = 45^\circ$, (H) $16 \times 40 \mu\text{m}^2$, $\alpha_t = 45^\circ$, (I) $30 \times 40 \mu\text{m}^2$, $\alpha_t = 45^\circ$, and (J) 80×40
 2987 μm^2 , $\alpha_t = 45^\circ$.

**Supplementary Fig. 55.** SEM images and FEM results of bottom-etched
 parallelogram strips with different widths. (A) Panorama SEM images of bottom-
 etched self-assembled parallelogram strip nanomembranes with different widths.
 SEM images of self-assembled parallelogram strip bottom-etched nanomembrane
 with sizes of (C) $4 \times 40 \mu\text{m}^2$, $\alpha_t = 45^\circ$, (D) $16 \times 40 \mu\text{m}^2$, $\alpha_t = 45^\circ$, (E) $30 \times 40 \mu\text{m}^2$, $\alpha_t =$
 45° and (F) $80 \times 40 \mu\text{m}^2$, $\alpha_t = 45^\circ$, $\alpha_t = 45^\circ$. FEM simulation results of self-assembled
 bottom-etched parallelogram strip nanomembrane with sizes of (G) $4 \times 40 \mu\text{m}^2$, $\alpha_t =$
 45° , (H) $16 \times 40 \mu\text{m}^2$, $\alpha_t = 45^\circ$, (I) $30 \times 40 \mu\text{m}^2$, $\alpha_t = 45^\circ$, and (J) $80 \times 40 \mu\text{m}^2$, $\alpha_t = 45^\circ$.

We believe that polymorphic assembly can also be realized in other forms, but
 the premise is that there are at least two low-strain elastic energy states in the
 assembly structures to form a polymorphic state¹⁶. Meanwhile, there should be a large
 enough energy barrier for switching between to ensure that the states are stable^{17,18}.

Taking a parallelogram pattern as an example, we assume the ideal etching direction
is in different etching directions β . When passing through a fixed point in the strip, if
an assembling structure wants to switch from the longer diagonal side rolling to the
shorter diagonal side rolling, a higher strain energy state will exist when the rolling
direction is changed, hindering the transition between the two stable states
(Supplementary Fig. 53). Therefore, only patterns with multi-stable state and a
sufficiently large strain energy barrier (such as parallelogram strips and rectangles
with moderate tilt angles) can realize multiple structural designs under the same
precursor. For semicircles and isosceles right triangles, the length of the free end is
short within the characteristic size of 2 to 80 μm , making it difficult to form a
potential barrier that effectively separates the two forms, so there is no multi-stable
structure.

A parallelogram with the same angle but a changed length-to-width ratio will
roll along different diagonals due to the difference in spatial order during oblique
etching (Supplementary Fig. 13,54,55). When etching occurs from the bottom, the
wide edge at the top will be fixed during the initial etching. When finally released, it
will tend to exhibit a unidirectional roll perpendicular to the shorter diagonal, thus to
form various types of tubes and rings (Supplementary Fig. 54). When etching occurs
from the top, the rapid expansion of the initial etching boundary in the obtuse angle
area at the top will cause self-assembly to occur along the shorter diagonal direction,
thus forming a large number of helix structures (Supplementary Fig. 55). The
switching between the two forms will produce high elastic energy states in other
rolling directions as potential barriers, thereby achieving two types of stable structures
with apparent differences. We noticed that some Cr nanomembrane areas are affected
by photolithography accuracy and local penetration of chromium etchant during
micro-nano processing. Therefore, the ends of the parallelogram strips will be
inconsistent with the design model. At the same time, some simulations of
parallelogram strips have problems that the nanomembrane moves behind the
substrate that should have existed in the experiments (Supplementary Fig. 54I).
Although their morphology is consistent with the experiment, this means that for
more accurate model predictions, the contact between the self-assembled
nanomembrane and the substrate plane needs to be considered. When applying an

improved model in the future, the multilevel FEM method can be further refined to
improve the feasibility of the application.

**Supplementary Fig. 56.** SEM images and FEM results of the bottom-etched self-
assembled parallelogram strips with different tilt angles. (A) Panorama SEM images
of immersion etching of self-assembled parallelogram strips at different tilt angles.
SEM images of bottom-etched self-assembled parallelogram strips with angles of (B-I)
$\alpha_t = 10^\circ$ - 80° . FEM results of bottom-etched self-assembled parallelogram strips with
angles of (J-Q) $\alpha_t = 10^\circ$ - 80° .

Supplementary Fig. 57. SEM images and FEM results of top-etched self-assembled parallelogram strips with different tilt angles. (A) Panorama SEM images of immersion etching of self-assembled parallelogram strips at different tilt angles. SEM images of top-etched self-assembled parallelogram strips with angles of (B-I) $\alpha_t = 10^\circ$ - 80° . FEM results of top-etched self-assembled parallelogram strips with angles of (J-Q) $\alpha_t = 10^\circ$ - 80° .

For parallelogram strips with the same width-length ratio and different tilt angles, the structural differences will mainly exist in the area with a tilt angle of 30° - 60° (Supplementary Fig. 14, 56, 57). When the tilt angle is too small ($10^\circ \leq \alpha_t < 30^\circ$), the pattern shows a strong dependence on long-side rolling, making it difficult to form various patterns, so both types of etching methods exhibit helix structures. When the tilt angle is moderate ($30^\circ \leq \alpha_t < 60^\circ$), the rolling in different diagonal directions form an effective strain energy barrier, thereby regulating the self-assembly process of the nanomembranes. At this time, the parallelogram strips etched on the bottom end roll along the longer diagonal and form a tube structure. In contrast, the etched tops roll wide edges from the shorter diagonal direction, mainly forming a helix structure. When the tilt angle is large ($60^\circ \leq \alpha_t < 80^\circ$), the length difference between the two

diagonals of the parallelogram gradually reduces, and the difference in elastic energy
release caused by the two etching methods is not high enough to form different types
of structures. It gradually approaches the helix structures in the released rectangular
shapes. However, it is worth noting that the difference in etching methods still affects
the rolling direction in the initial phase, causing the pattern to form a helix structure
with opposite chirality.

10. Revising the title of the part 18 from “SEM images and photoresponse for Si/Cr
photodetectors” to “SEM images and 2D angular photodetection for Si/Cr
photodetectors”.

11. Moved Supplementary Fig. 59A into Fig. 6e in manuscript:

12. Replaced Supplementary Fig. 22 in origin Supporting Information in “18. SEM
images and 2D angular photodetection for Si/Cr photodetectors” with a new part “20.
Setup of incident light directional detection”:

**Supplementary Fig. 22.** Image of hemispherical omnidirectional incident light
controller. (Origin Supporting Information)

13. A part of response to Comment #2 and Comment #3 of Reviewer #1, Re-Table 1,
and Re-Fig. 2-4 (Page 5-14 of this file) have been added as “19. Photodetection
mechanism and performance of Si/Cr photodetectors” in Supporting Information as
follow:

**Supplementary Fig. 60.** Electrical properties and photoelectric responsivity of Si/Cr3089 photodetector. (A) I-V curve and (B) $\log(I)$ - $\log(V)$ curve of arch-type photodetector.3090 (C) $\ln(I/V)$ - \sqrt{V} relationship curve of arch-type photodetector in the poole-frenkel

emission region. (D) Photoresponse of channel area and metal-semiconductor junction
area of arch-type photodetector. (E) I-V curves of arch-type photodetectors under 520
3093 nm laser irradiation with different powers. (F) Responsivity-power relationship
diagram of photodetectors with different structures. (G) Responsivity-wavelength
relationship of arch-type photodetector. (H) Simulated absorbance of Ge/Si/Cr in
planar SiO₂ (35 nm)/Ge (50 nm)/Si (60 nm)/Cr (40 nm) multilayer.

We characterized the photodetection properties of the prepared Si/Cr
photodetector, as shown in Supplementary Fig. 60. We conducted an I-V test on the
arch-type Si/Cr photodetector, which showed Schottky contact-dominated carrier
transport behavior in the atmospheric environment (Supplementary Fig. 60A).
According to the log(I)~log(V) curve, it can be seen that the device mainly exhibits
three types of transmission behavior at 0~10 V, with slopes of 0.61, 0.99 and 1.02
respectively (Supplementary Fig. 60B). Among them, the slope is 0.61 in the region
of 0.05~0.35 V. This is because when the voltage is small, carriers will be easily
trapped by surface defects before being transported to the electrode, causing the
$\frac{d \log(I)}{d \log(V)}$ less than 1. In the region of 0.35~1.35 V, the $\frac{d \log(I)}{d \log(V)}$ is 0.99, and the current-
voltage shows a linear relationship, but it does not indicate that Si/Cr exhibits ohmic
contact. This is because the device exhibits the typical Poole-Frenkel (P-F) emission
effect in Region III (1.35-10 V), which is only present in Schottky contacts. The
effect mainly originates from the external electric field emitting the trapped electrons
and holes to the continuous state related to the intrinsic dislocation of the Schottky
barrier layer, and further directional movement of carrier generates current. When
carriers in P-F emission enter the semiconductor material side from the Schottky
contact metal material layer, they need to pass through the trap energy level in the
potential barrier rather than through direct transition. At the same time, P-F emission
mainly occurs inside the semiconductor material rather than at the metal-
semiconductor interface. Its current density can be expressed as:

$$J = CE_b \exp \left[- \frac{q(\phi_B - \sqrt{qE_b/\pi\epsilon_0\epsilon_s})}{k_B T} \right]$$

in which ϕ_B is the voltage barrier that electrons must cross when moving from
one atom to another in a material in a zero electric field, k_B is Boltzmann's constant,
ϵ_s is the relative dielectric constant of the barrier layer, T is the temperature, E_b is the

external electric field strength, which is a constant. Taking the natural logarithm of
 both sides of this formula gives:

$$3124 \quad \ln (J/E_b) = \frac{q}{kT} \sqrt{\frac{qE_b}{\pi\epsilon_0\epsilon_s}} - \frac{q\phi_B}{kT} + \ln C.$$

There is

$$\ln (J/E_b) \propto \sqrt{E_b},$$

and combine the linear relationship of

$$J = \sigma E.$$

Multiply both ends by the cross-sectional area of the device A , we have

$$3128 \quad I = JA = \sigma AE = \sigma A \frac{V}{L} = k\sigma V,$$

where k is a constant. The current-voltage relationship and the current density-
 electric field intensity relationship can be linearly converted, so we can observe the
 relationship between $\ln (I/V) - \sqrt{V}$ to analyze the P-F emission in the device¹⁹. As
 shown in Supplementary Fig. 60C, the $\ln (I/V) - \sqrt{V}$ curve in Region III shows
 excellent linearity, confirming that the device is a Schottky contact. Therefore, the
 linear region in Region II is mainly related to non-ideal factors in the device. Due to
 the large series resistance, the exponential current characteristics have not yet been
 demonstrated in this voltage region. At this time, the Schottky junction I-V
 characteristic equation will be written as²⁰:

$$I = I_0 \left[\exp \left(\frac{q(V - IR_s)}{nkT} \right) - 1 \right]$$

$$I_0 = AA^*T^2 \exp (-q\phi_b/kT)$$

, when $V \sim IR_s$ exists, the exponential term of the above formula can be
 expanded as:

$$I \sim I_0 \cdot \left(\frac{q(V - IR_s)}{nkT} + 1 \right) - 1 = I_0 \cdot \frac{q(V - IR_s)}{nkT}$$

, then $I = \frac{V}{\frac{nkT}{I_0q} + IR_s} = c_0V$, where c_0 is a constant. Taking the logarithm of both

sides gives $\log(I) = \log(V) + \log(c_0)$, in which the slope is $\frac{d \log(I)}{d \log(V)} = 1$. Therefore,
 the larger series resistance may be the reason for the Region II slope of 1. Due to the
 lack of C-V testing and variable temperature testing platforms, we are currently
 unable to measure the barrier height of the Si-Cr Schottky contact. We hope that
 corresponding characterization methods can be introduced in subsequent work for

further analysis. Since the voltage in Region I is small, a large number of carriers
cannot be moved into the electrode area for collection, and the voltage in Region III is
large, which will bring a large dark current and noise that affects photodetection.
Therefore, we chose the operating voltage of the photodetector to be 1 V.

Then, we analyzed the mechanism of the photodetector. The photodetection of
this device is mainly based on the resistance change caused by the generation of
carriers in the silicon nanomembrane under light rather than the built-in electric
potential field of the Schottky junction. We conducted photoresponse tests on the
channel area and the Si/Cr Schottky contact area. It can be seen that only the silicon
channel area shows an obvious response to light, while the Si-Cr Schottky contact
area exhibits no response to light, which means that the Schottky junction is not
mainly involved in the photodetection process, and photo carriers mainly originate
from the silicon nanomembrane channel (Supplementary Fig. 60D). Although
commercial photodetectors are mainly photovoltaic devices based on p-n junctions,
photoconductive photodetectors are also widely used in the direction of photosensitive
elements. In the follow-up work, we aim to further optimize device design, develop
CMOS technology, and prepare microstructure photodetectors with higher
performance.

The thickness of our photodetector is at the nanometer scale, which will
exhibit a weak response in mid-infrared wavelength. Therefore, we focused on
studying the responsivity-wavelength relationship in the visible spectrum. We tested
the responsivity-wavelength relationship of the arch-type Si/Cr photodetector, as
shown in Supplementary Fig. 60G. It can be seen that the detector can effectively
detect the visible light range, and its responsivity is 20~60 mA/W in the wavelength
range of 400-750 nm. The responsivity shows multiple peaks, which is different from
bulk Si devices. The reflection at the silicon oxide/silicon substrate interface will
cause periodic fluctuations in the absorption of the Si/Cr detector. We simulated the
SiO₂ (35 nm)/Ge (50 nm)/Si (60 nm)/Cr (40 nm) system and found that the absorption
rate of Ge/Si/Cr is similar to our device responsivity wavelength relationship,
confirming that the nanomembrane system has a certain phase length interference
effect on specific wavelength incident light (Supplementary Fig. 60H). Due to the
local rolling of the arch structure, its response band will differ from that of the planar
multilayer nanomembrane system.

**Supplementary Fig. 61.** Response time of (A) ring, (B) arch, (C) helix, (D) taper, (E)

tube, and (F) planar Si/Cr photodetectors.

For various 3D structure Si/Cr photodetectors, the relationship between the responsivity and power is as shown in the Supplementary Fig. 60F. The Si/Cr 3D photodetector can detect incident light under weaker light, thanks to their 3D structures. The responsivity of photodetectors can reach up to 40 mA/W, which is slightly lower than commercial silicon-based photodetectors (Supplementary Fig. 60E). However, various types of Si/Cr photodetectors, including planar structures, exhibit a decrease in responsivity as the power density of the incident light source

increases. This may be related to the recombination of photogenerated charge carriers
or the enhanced scattering rate as the carrier concentration increases at higher optical
power densities. We tested the responsivity-wavelength relationship of the arch-type
Si/Cr photodetector, as shown in the Supplementary Fig. 60F. It can be seen that the
detector can effectively detect the visible light range, and its responsivity will
gradually increase as the wavelength increases.

At the same time, we tested the responsivity of the Si/Cr photodetector, as
shown in the Supplementary Fig. 62. The ring, arch, helix, taper, tube, and planar
structure photodetectors showed rise times of 120, 138, 389, 758, and 192 μs
respectively and decay times of 10, 688, 180, 624, and 197 μs , respectively. The
response time of photodetectors is slower than commercial silicon-based detectors and
is primarily limited by the lower carrier mobility and presence of impurities in
amorphous silicon. It is worth noting that in nanomembranes, the thickness of the
semiconductor material is very thin so that the surface defects will dominate the
performance of the device, and the nanomembrane released from the sacrificial layer
will be contaminated by impurities on both the front and back sides. The above
disadvantages will further affect the response time of the photodetectors. Therefore,
surface engineering of self-assembled structures will become an important research
direction in our subsequent work to improve the performance of 3D photodetectors.

For the photodetectors, their photoconversion efficiency is usually
characterized by external quantum efficiency (EQE). We obtained the EQE of the
photodetectors through the responsivity calculation. The calculation formula is as
follows:

$$EQE = R_{\lambda} \cdot \frac{h\nu}{e} \times 100\%$$

Supplementary Fig. 62. EQE-wavelength relationship of the arch-type photodetector.

It fluctuates in the range of 7~12% (Supplementary Fig. 62), which can
 effectively detect incident light, but there is still room for improvement. The low
 responsivity and quantum efficiency of photodetectors are mainly caused by the
 following reasons. First, the silicon nanomembrane that constitutes the photodetector
 is thinner and absorbs fewer carriers generated by incident light, resulting in a smaller
 photocurrent. Secondly, the silicon nanomembrane grown by electron beam
 evaporation is amorphous, and wet etching during the self-assembly process will
 introduce additional contamination at the nanomembrane interface. The amorphous
 structure and a large number of surface states will significantly affect the migration
 rate of photogenerated carriers from the channel to the electrode and their
 recombination before they are collected by the electrode, thus reducing the
 responsivity and prolonging the response time. Our research on this type of 3D
 structured photodetector mainly focuses on its angle detection and identification, and
 there is a lot of room for improvement in performance optimization. We have
 compiled some performance of visible light band detector devices reported in
 commercial and previous studies (Supplementary Table 2).

**Supplementary Table 2.** Comparison of photodetection performance with other
 photodetectors in this work.

Detector material	Detecti on mechanism	Detec tion range	Responsiv ity	Photocon version efficiency	Response time/response frequency	Source
Si (60 nm)	Photoc onducti ve	400- 750 nm	52.5 mA/W @ 530 nm	12.2%@ 530 nm (EQE)	138μs (rise)/68μs(de cay)	This work
Si(bulk)	Photov oltaic	350- 1100 nm	640 mA/W @ 980 nm	82.2%@ 980 nm (EQE)	10 ns (rise)/10 ns (decay)	THORLABS FDS100-CAL
Si/Graph ene	Photov oltaic	1300- 1600 nm	30 mA/W @ 1550 nm	10%@15 50 nm (IQE)	~18 GHz	Ref. 21

Si (bulk)	Photoc onducti ve	400- 700 nm	~4 @540 nm	A/W	~900% @540 nm	30 ms	JCHL GL48569 photoresistor
Silicon (300 nm)	Photov oltaic	400- 800 nm	80 mA/W@6 00 nm	-	-	9 GHz	Ref. 22
Si(bulk): Ag	Photoc onducti ve	400- 1200 nm	1.71 @800 nm	A/W	266% @800 nm (EQE)	12.5 (rise)/15.9 μ s (decay)	Ref. 23
CdTe nanopart icles	Photoc onducti ve	365 nm, 500- 800 nm	4 (visible wavelengt h)	mA/W	-	134 μ s	Ref. 24
β -Ga ₂ O ₃	Photoc onducti ve	200- 300 nm	61.3 @254 nm	A/W	30000% @254 nm	3 ms (rise)/35 ms (decay)	Ref. 25

In the follow-up research work, we also hope to improve the detection
performance of the 3D photodetector further through dry etching, passivation layer
coating, and high-quality thin nanomembrane epitaxy. In addition, the free-standing
self-rolling microstructure has excellent potential in long-wave infrared detection. The
freestanding structure can be realized using a simple one-step self-rolling process,
thereby avoiding the complicated process flow in the traditional bolometer
manufacturing process²⁶.

14. A part of response Comment #3 of Reviewer #1 and Re-Fig. 5-6 (Page 14-16 of
this file) have been added as “20. Setup of incident light directional detection” in
Supporting Information as follow:

**20.Setup of incident light directional detection**

In order to facilitate readers' understanding, we have supplemented the process
of setting up the test platform. As shown in the Supplementary Fig. 63, the laser is
driven by DC power supply 1, in which the signal generator generates a square wave

signal to modulate the laser to achieve the emission of optical signals of a specified
 frequency. Currently, the Si/Cr photodetector driven by another DC power source
 detects the optical signal emitted by the laser and generates the current signal. The
 current signal will be input into the lock-in amplifier and the oscilloscope respectively
 through the preamplifier, where the lock-in amplifier is used to record the
 photocurrent intensity, and the oscilloscope is used to record the photoresponse
 waveform and measure the response time. First, we fix the 3D photodetector to be
 tested on the optical platform, which has been connected to the PCB circuit board
 through the bonding machine, and the PCB circuit board is connected to the
 semiconductor test unit (Supplementary Fig. 64A-C). The optical image of the
 omnidirectional light controller is shown in Supplementary Fig. 64D. It consists of
 fixed grooves distributed according to spherical coordinates and a detachable SMA
 optical fiber interface. The fixed grooves are distributed at intervals of 10° according
 to longitude and latitude, and they interface with the SMA optical fiber. After the
 combination, the laser can be connected to perform photoelectric testing on the
 sample in the center of the spherical shell.

Then, we place the incident light controller on the optical platform and
 perform XY plane alignment and Z-axis alignment, respectively, to ensure that the
 incident light ($90^\circ, 0^\circ$) orientation is located in the center of the photodetectors on the
 PCB board and align the height of the z-axis at the bottom of the controller with the
 photodetectors. Then, connect the 520 nm laser to the required SMA port and use a
 semiconductor measurement unit to provide voltage to the sample for photodetection
 of the incident light.

Supplementary Fig. 63. Schematic diagram of 3D structure Si/Cr photodetector testing equipment.

**Supplementary Fig. 64.** Testing Setup for 3D structured photodetector. (A)
Positioning of device and controller support platform. (B) Connection of devices, test
boxes, and test units. (C) Test unit connection and schematic diagram. The (D) XY
plane of the incident light direction controller is aligned with the (e-f) Z axis. (G)
Schematic diagram of photodetection testing process.

15. Response Comment #4 of Reviewer #2 and Re-Fig. 41 (Page 57-60 of this file)
have been added as “22. Contact module in FEM model” in Supporting Information
as follow:

**22.Contact module in FEM model**

**Supplementary Fig. 67.** Schematic diagram of contact module in the multilevel
 design model. (A), (C) FEM simulation results of the multilevel model in the case of
 single-turn rolling without the contact module. (B), (D) FEM simulation results of the
 multilevel model in the case of single-turn rolling when applying the contact module.
 (E) FEM simulation results of the multilevel model in the case of multi-turn rolling
 without applying the contact module. (F) FEM simulation results of the multilevel
 model in the case of multi-turn rolling when applying the contact module. (G) SEM
 images of multi-turn rolling Si/Cr nanomembrane.

Here, we demonstrate some examples of the contact module in the multilevel
 FEM model adopted in this study. In the unidirectional rolling process of the

3301 traditional FEM model, when the contact module is not applied to the model, the self-
3302 assembled nanomembrane will partially overlap (Supplementary Fig. 67A, C). This
phenomenon has less impact on self-assembly models with a single turn or less than a
single turn because the self-contact area of such models is smaller. However, the lack
of a contact module will cause extremely inaccurate boundary condition results for the
analysis of the multi-turn rolling model and further affect the subsequent stress-strain
analysis that may be required. The large-scale Si/Cr nanomembrane self-assembly
model contains a large number of model self-contacts during the deformation process.
As shown in the Supplementary Fig. 67E, when the contact module is not introduced
into the model, it will cause serious distortion of the model, with multiple tubes of
self-assembly in different directions, and it is difficult to stabilize. Here, we apply the
general contact module to the same model and define the tangential contact
mechanism as a penalty function whose form is

$$3314 \quad \tau = \mu\sigma,$$

where τ is the tangential friction force, σ is the normal stress, and the friction
coefficient is defined as the dynamic friction coefficient of the Cr-Cr layer $\mu = 0.34$.
The normal contact mechanism is hard contact.

As a result, it can be seen that there is no volume overlap of the
nanomembranes in the calculation results (Supplementary Fig. 67B, D). At the same
time, its multi-turn self-assembled structure will quickly stabilize due to the existence
of the contact volume, reflecting more realistic experimental results. After applying
the contact module, we found that the self-contact problem of the model is
significantly improved (Supplementary Fig. 67F), but due to the need to calculate the
contact between a large number of small-sized meshes during the quasistatic release
of the component, a large amount of computing power is consumed, which will
significantly extend the simulation time. Based on the above situation, when the self-
assembly process is free from self-contact or only has single-axis curvature single-
turn rolling, we mainly use non-contact module operations. When the self-assembly
process contains complex deformation or multi-turn rolling, we will consider the
contact module to ensure the accuracy of the results.

16. Response Comment #1 of Reviewer #3 and Re-Fig. 48-49 (Page 69-74 of this file)
have been added as “23. TEM characterization and crystallographic analysis of the

Si/Cr nanomembrane” in Supporting Information as follow:

**23.TEM characterization and crystallographic analysis of the Si/Cr**
**nanomembrane**

We analyzed the samples by transmission electron microscope (TEM), and it
can be seen that there is no coherent interface at the Si/Cr interface, but an obvious
non-coherent phase boundary. The above evidence exhibits that the prestrain of Si and
Cr bilayer nanomembranes is mainly caused by the temperature difference during
deposition and the difference in thermal expansion coefficient of the bilayer
nanomembranes after cooling, rather than lattice mismatch.

In order to confirm that the strain gradient between Si/Cr double-layer
nanomembranes does not originate from lattice mismatch, we performed focused ion
beam (FIB) cutting on the sample and used high-angle annular dark field scanning
transmission electron microscopy (HAADF-STEM) and TEM to examine the cross
section of the sample. Morphological characterization is performed. As can be seen
from Supplementary Fig. 68A, the sample is composed of Si/SiO₂/Ge
nanomembrane/Si nanomembrane/Cr nanomembrane from bottom to top. The carbon
protective layer and Pt protective layer are sprayed on the Cr nanomembrane to
protect the cross section of FIB cutting samples. To verify the elemental composition
of each material, energy dispersive X-ray spectroscopy is used to conduct elemental
analysis of the interface. The distribution diagram of each element is shown in
Supplementary Fig. 68B-G, in which each element shows high signal intensity in the
corresponding single-element layer, confirming the elemental composition of each
layer in the cross-section. From the analysis of Supplementary Fig. 68B and C, it can
be seen that there is a silicon oxide layer with a thickness of approximately 35 nm
above the silicon substrate, which is obtained by thermal oxidation of the silicon
wafer substrate. In addition, Supplementary Fig. 68C shows stronger intensity in the
Cr nanomembrane layer. This is because the higher temperature during electron beam
evaporation will oxidize the evaporated Cr and the remaining oxygen in the cavity.
But in general, chromium nanomembranes have less oxygen adsorption content,
which can be seen from optical images and composition analysis. Since silicon and
germanium nanomembranes are also deposited by electron beam evaporation, their
EDX images show a small amount of oxygen distribution. As shown in
Supplementary Fig. 68G, the entire nanomembrane system shows a small amount of

carbon element distribution. This is due to the slight accumulation of carbon-
containing substances and carbon protective layers in the sample during processing
and electron beam bombardment. The samples are plasma cleaned, but contamination
by carbonaceous species is not completely avoided. This cross-sectional analysis
shows the clear interface between the Ge/Si/Cr three-layer nanomembranes, which
can be further analyzed by selected area electron diffraction (SAED) techniques to
analyze the interface and crystallization state of each material.

**Supplementary Fig. 68.** Characterization of the cross-sectional structure and element
distribution of the nanomembrane sample. (A) HAADF-STEM image of the cross-
section of the sample. EDX images of the distribution of (B) Si, (C) O, (D) Pt, (E) Cr,
(F) Ge, and (G) C elements in the cross section of the sample.

As shown in Supplementary Fig. 69A-D, we used high-resolution transmission
electron microscopy (HRTEM) to characterize the morphology of each layer and
interfaces. It can be seen that there are obvious interfaces between each layer. In
addition, we performed SAED analysis on each layer. Supplementary Fig. 69A shows
the interface diagram between the silicon substrate and silicon oxide. The images
show the orderly arrangement of atoms in the silicon substrate without grain boundary.
At the same time, Supplementary Fig. 69E shows the SAED image of the silicon
substrate. It can be seen that clear diffraction spots in the direction of the [011] zone
axis, in which the silicon (200) and (11 $\bar{1}$) crystal plane proves that the substrate is
single crystal silicon. In Supplementary Fig. 69B and Supplementary Fig. 69C, the
HRTEM images of the germanium nanomembrane and silicon nanomembrane do not
show regular atomic arrangement. In the SAED images of Supplementary Fig. 69F

and Supplementary Fig. 69G, there are no sharp diffraction rings corresponding to
polycrystalline materials or discrete diffraction spots corresponding to single crystal
materials, but diffused diffraction rings, confirming that the germanium
nanomembrane and silicon nanomembrane deposited by electron beam evaporation
are amorphous nanomembranes. Previous studies have reported growing
polycrystalline silicon and polycrystalline germanium thin nanomembranes through
electron beams, but it is worth noting that the substrate usually needs to be heated to
400-600 °C^{27,28}, which will provide sufficient energy for the movement of atoms to
the liquid-solid interface during the growth of silicon and germanium crystals. The
electron beam evaporation in this study is deposited on the substrate at room
temperature, which causes the atoms to cool immediately after nucleation on the
substrate and become unable to continue growing, eventually forming amorphous
materials. Supplementary Fig. 69D shows the HRTEM image of the Si/Cr interface,
in which the Si nanomembrane still shows a disordered atomic arrangement, and the
atoms in the Cr nanomembrane show obvious grain boundary structures and grains.
From the SAED characterization of Supplementary Fig. 69H, it can be seen that the
Cr nanomembrane layer exists in a polycrystalline form, which exhibits obvious
diffraction rings, and the diffraction ring distance in the inverted space is highly
consistent with the theoretical value²⁹. Compared with Si and Ge, which are
semiconductor materials, the growth of Cr grains faces a relatively small
thermodynamic energy barrier, and their thermal conductivity is also good, which will
facilitate the growth during electron beam evaporation and ultimately form a
polycrystalline Cr nanomembrane.

In summary, we can conclude that the strain gradient between Si/Cr
nanomembranes is not due to lattice mismatch because the Si nanomembranes are
amorphous nanomembranes, which do not have an ordered lattice structure, and the
Cr nanomembranes are polycrystalline. The lattice constants of the two materials are
very different, and they cannot achieve coherent crystal planes. Therefore, we believe
that the strain gradient of Si/Cr nanomembranes originates from nanomembrane
cooling after electron beam evaporation and the difference in thermal strain caused by
different thermal expansion coefficients during cooling process after deposition.

**Supplementary Fig. 69.** Characterization of the interface morphology of the
 nanomembrane sample substrate and analysis of the crystal form of each layer of
 nanomembrane. HRTEM images of (A) silicon substrate/silicon oxide, (B) silicon
 oxide/germanium nanomembrane, (C) germanium nanomembrane/silicon
 nanomembrane, (D) silicon nanomembrane/chromium nanomembrane interface.
 SAED images of (E) silicon substrate, (F) germanium nanomembrane, (G) silicon
 nanomembrane, and (H) chromium nanomembrane (the inset is the TEM images of
 the corresponding SAED area).

17. “Legends to Movies S1 to S7” are sorted and added as follow:

**Movie S1.** Etchant concentration and flux distribution during etching process. This
 movie demonstrates the concentration and flux distribution of etchant in different
 patterns during etching process.

**Movie S2.** Flow velocity during etching process. This movie demonstrates the flow
 velocity of different patterns when a bubble rises up around the pattern.

**Movie S3.** Etching process of pre-strained nanomembranes. This movie demonstrates
 the optical movie of etching process of Si/Cr nanomembrane with different sizes.

**Movie S4.** Selective etching by controlling etchant level by tilting wafer. This movie
 exhibits an etching direction selection via different tilting directions.

**Movie S5.** Structural forming of pre-strained nanomembrane in top etching. This
 movie demonstrates quasistatic releasing process of pre-strained nanomembrane from
 top etching, which finally morph into a taper structure.

**Movie S6.** Structural forming of pre-strained nanomembrane in bottom etching. This

movie demonstrates quasistatic releasing process of pre-strained nanomembrane from
bottom etching. When the nanomembrane is released incompletely, two sides of
nanomembrane buckle up to build an energy barrier, which hinder the formation of
taper structure, and finally turn into tube structure.

**Movie S7.** Reconfiguration of Si/Cr microstructure via surface tension. This movie
exhibits the structure reconfiguration of assembled Si/Cr nanomembrane from ring
structure to planar pattern, and then reassembled into ring from planar pattern.

References in revised SI

- 1. Kosior, D., Wiertel-Pochopien, A., Kowalczyk, P. B., Zawala, J. (2023)
Bubble Formation and Motion in Liquids—A Review. *Minerals*. 13(9).
10.3390/min13091130.
- 2. Samkhaniani, N., Ansari, M. R., Numerical simulation of superheated vapor
bubble rising in stagnant liquid. *Heat and Mass Transf.* **53**, 2885-2899 (2017).
- 3. Huang, W., et al., Three-dimensional radio-frequency transformers based on a
self-rolled-up membrane platform. *Nat. Electron.* **1**, 305-313 (2018).
- 4. Sang, L., et al., Monolithic radio frequency SiN_x self-rolled-up nanomembrane
interdigital capacitor modeling and fabrication. *Nanotechno.* **30**, 364001
(2019).
- 5. Kim, M.-S., Lee, H.-T., Ahn, S.-H., Laser Controlled 65 Micrometer Long
Microrobot Made of Ni-Ti Shape Memory Alloy. *Adv. Mater. Techno.* **4**,
1900583 (2019).
- 6. Wu, B., et al., One-step rolling fabrication of VO₂ tubular bolometers with
polarization-sensitive and omnidirectional detection. *Sci. Adv.* **9**, eadi7805
(2023).
- 7. Li, X., et al., Self-rolling of vanadium dioxide nanomembranes for enhanced
multi-level solar modulation. *Nat. Commun.* **13**, 7819 (2022).
- 8. Huang, W., et al., Precision structural engineering of self-rolled-up 3D
nanomembranes guided by transient quasi-static FEM modeling. *Nano Lett.* **14**,
6293-6297 (2014).
- 9. Xu, B., et al., Stimuli-responsive and on-chip nanomembrane micro-rolls for
enhanced macroscopic visual hydrogen detection. *Sci. Adv.* **4**, eaap8203
(2018).
- 10. Landau, L. D., Lifshitz, E. M., Atkin, R., Fox, N. (2020) The Theory of
Elasticity. *Physics of Continuous Media*, (CRC Press), pp 167-178.
- 11. Mei, Y., et al., Fabrication, Self-Assembly, and Properties of Ultrathin
AlN/GaN Porous Crystalline Nanomembranes: Tubes, Spirals, and Curved
Sheets. *ACS Nano* **3**, 1663-1668 (2009).
- 12. Nikishkov, G. P., Curvature estimation for multilayer hinged structures with
initial strains. *J. Appl. Phys.* **94**, 5333-5336 (2003).
- 13. Hsueh, C.-H., Modeling of elastic deformation of multilayers due to residual
stresses and external bending. *J. Appl. Phys.* **91**, 9652-9656 (2002).
- 14. Chun, I. S., et al., Geometry effect on the strain-induced self-rolling of
semiconductor membranes. *Nano Lett.* **10**, 3927-3932 (2010).
- 15. Alben, S., Balakrishnan, B., Smela, E., Edge effects determine the direction of
bilayer bending. *Nano Lett.* **11**, 2280-2285 (2011).

- 16. Armon, S., Efrati, E., Kupferman, R., Sharon, E., Geometry and Mechanics in
the Opening of Chiral Seed Pods. *Science* **333**, 1726-1730 (2011).
- 17. Guo, Q., Zheng, H., Chen, W., Chen, Z., Modeling Bistable behaviors in
Morphing Structures through Finite Element Simulations. *Biomed. Mater. Eng.*
**24**, 557-562 (2014).
- 18. Seffen, K. A., Guest, S. D., Prestressed Morphing Bistable and Neutrally
Stable Shells. *J. Appl. Mech.* **78**, 011002 (2010).
- 19. Zhou, G., et al., Coexistence of Negative Differential Resistance and Resistive
Switching Memory at Room Temperature in TiOx Modulated by Moisture.
*Adv. Electron. Mater.* **4**, 1700567 (2018).
- 20. Sze, S. M., Li, Y., Ng, K. K. (2021) *Physics of semiconductor devices* (John
wiley & sons).
- 21. Pospischil, A., et al., CMOS-compatible graphene photodetector covering all
optical communication bands. *Nat. Photon.* **7**, 892-896 (2013).
- 22. Xu, H., et al., High-Performance Lateral Avalanche Photodiode Based on
Silicon-on-Insulator Structure. *IEEE Electron Device Lett.* **43**, 1077-1080
(2022).
- 23. Qiu, X., et al., Trap Assisted Bulk Silicon Photodetector with High
Photoconductive Gain, Low Noise, and Fast Response by Ag Hyperdoping.
*Adv. Opt. Mater.* **6**, 1700638 (2018).
- 24. Naje, A. N., Muhammed, G. S., Murad, H. I., Improvement of CdTe
nanoparticles photoconductive detector by adding metal nanoparticles. *J.*
*Optics*, (2023).
- 25. Shen, G., et al., Solar-blind UV communication based on sensitive β -Ga₂O₃
photoconductive detector array. *Appl. Phys. Lett.* **123**, 041103 (2023).
- 26. Rogalski, A. (2000) *Infrared detectors* (CRC press).
- 27. Yun, J., et al., Effect of deposition temperature on electron-beam evaporated
polycrystalline silicon thin-film and crystallized by diode laser. *Appl. Phys.*
*Lett.* **104**, 242102 (2014).
- 28. Michael, A., et al., Investigation of E-Beam Evaporated Silicon Film
Properties for MEMS Applications. *J. Microelectromech. Syst.* **24**, 1951-1959
(2015).
- 29. Haynes, W. M. (2014) *CRC handbook of chemistry and physics* (CRC press).

REVIEWERS' COMMENTS

Reviewer #1 (Remarks to the Author):

The authors addressed previous concerns and improved the quality of the manuscript. Therefore, it is recommended for the publication.

Reviewer #2 (Remarks to the Author):

The authors have provided a detailed and thorough response to the comments and addressed all of my concerns. The revised manuscript is recommended for publication.

Reviewer #3 (Remarks to the Author):

The authors have addressed my questions and improved the manuscript nicely. I recommend the acceptance of this revised manuscript as is.